# Rethinking Compressed Convolutional Neural Networks from a Statistical Perspective

## Abstract

Many designs have recently been proposed to improve the model efficiency of convolutional neural networks (CNNs) at a fixed resource budget, while there is a lack of theoretical analysis to justify them. This paper first formulates CNNs with high-order inputs into statistical models, which have a special "Tucker-like" formulation. This makes it possible to further conduct the sample complexity analysis to CNNs as well as compressed CNNs via tensor decomposition. Tucker and CP decompositions are commonly adopted to compress CNNs in the literature. The low rank assumption is usually imposed on the output channels, which according to our study, may not be beneficial to obtain a computationally efficient model while a similar accuracy can be maintained. Our finding is further supported by ablation studies on CIFAR10, SVNH and UCF101 datasets.

## 1 Introduction

The introduction of AlexNet (Krizhevsky et al., 2012) spurred a line of research in 2D CNNs, which progressively achieve high levels of accuracy in the domain of image recognition (Simonyan & Zisserman, 2015; Szegedy et al., 2015; He et al., 2016; Huang et al., 2017). The current state-of-the-art CNNs leave little room to achieve significant improvement on accuracy in learning still-images, and attention has hence been diverted towards two directions. The first is to deploy deep CNNs on mobile devices by removing redundancy from the over-parametrized network, and some representative models include MobileNetV1 & V2 (Howard et al., 2017; Sandler et al., 2018).

The second direction is to utilize CNNs to learn from higher-order inputs, for instance, video clips (Tran et al., 2018; Hara et al., 2017) or electronic health records (Cheng et al., 2016; Suo et al., 2017). This area has not yet seen a widely-accepted state-of-the-art network. High-order kernel tensors are usually required to account for the multiway dependence of the input. This notoriously leads to heavy computational burden, as the number of parameters to be trained grows exponentially with the dimension of inputs. Subsequently, model compression becomes the critical juncture to guarantee the successful training and deployment of tensor CNNs.

**Tensor methods for compressing CNNs.** Denil et al. (2013) showed that there is huge redundancy in network weights such that the entire network can be approximately recovered with a small fraction of parameters. Tensor decomposition recently has been widely used to compress the weights in a CNN network (Lebedev et al., 2015; Kim et al., 2016; Kossaifi et al., 2020b; Hayashi et al., 2019). Specifically, the weights at each layer are first summarized into a tensor, and then tensor decomposition, CP or Tucker decomposition, can be applied to reduce the number of parameters. Different tensor decomposition to convolution layers will lead to a variety of compressed CNN block designs. For instance, the bottleneck block in ResNet (He et al., 2016) corresponds to the convolution kernel with a special Tucker low-rank structure, and the depthwise separable block in MobileNetV1 (Howard et al., 2017) and the inverted residual block in MobileNetV2 (Sandler et al., 2018) correspond to the convolution kernel with special CP forms. All the above are for 2D CNNs, and Kossaifi et al. (2020b) and Su et al. (2018) considered tensor decomposition to factorize convolution kernels for higher-order tensor inputs.

Tensor decomposition can also be applied to fully-connected layers since they may introduce a large number of parameters (Kossaifi et al., 2017; 2020a); see also the discussions in Section 5. Moreover, Kossaifi et al. (2019) summarized all weights of a network into one single high-order tensor, and then directly imposed a low-rank structure to achieve full network compression. While the idea is

highly motivating, the proposed structure of the high-order tensor is heuristic and can be further improved; see the discussions in Section 2.4.

Parameter efficiency of the above proposed architectures was heuristically justified by methods, such as FLOPs counting, naive parameter counting and/or empirical running time. However, there is still lack of a theoretical study to understand the mechanism of how tensor decomposition can compress CNNs. This paper attempts to fill this gap from statistical perspectives.

**Sample Complexity Analysis.** Du et al. (2018a) first characterized the statistical sample complexity of a CNN; see also Wang et al. (2019) for compact autoregressive nets. Specifically, consider a CNN model, $y = F_{\mathrm{CNN}}(\boldsymbol{x}, \mathcal{W}) + \xi$, where $y$ and $\boldsymbol{x}$ are output and input, respectively, $\mathcal{W}$ contains all weights and $\xi$ is an additive error. Given the trained and true underlying networks $F_{\mathrm{CNN}}(\boldsymbol{x}, \widehat{\mathcal{W}})$ and $F_{\mathrm{CNN}}(\boldsymbol{x}, \mathcal{W}^*)$, the root-mean-square prediction error is defined as

$$\mathcal{E}(\widehat{\mathcal{W}}) = \sqrt{E_{\boldsymbol{x}} |F_{\mathrm{CNN}}(\boldsymbol{x}, \widehat{\mathcal{W}}) - F_{\mathrm{CNN}}(\boldsymbol{x}, \mathcal{W}^*)|^2}, \tag{1}$$

where $\widehat{\mathcal{W}}$ and $\mathcal{W}^*$ are trained and true underlying weights, respectively, and $E_{\boldsymbol{x}}$ is the expectation on $\boldsymbol{x}$. The sample complexity analysis is to investigate how many samples are needed to guarantee a given tolerance on the prediction error. It can also be used to detect the model redundancy. Consider two nested CNNs, where $F_1$ is more compressed than $F_2$. Given the same true underlying networks, when the prediction errors from trained $F_1$ and $F_2$ are comparable, we then can argue that $F_2$ has redundant weights comparing with $F_1$. As a result, conducting sample complexity analysis to CNNs with higher order inputs will shed light on the compressing mechanism of popular compressed CNNs via tensor decomposition.

The study in Du et al. (2018a) is limited to 1-dimensional convolution with a single kernel, followed by weighted summation, and its theoretical analysis cannot be generalized to CNNs with compressed layers. In comparison, our paper presents a more realistic modeling of CNN by introducing a general $N$-dimensional convolution with multiple kernels, followed by an average pooling layer and a fully-connected layer. The convolution kernel and fully-connected weights are in tensor forms, and this allows us to explicitly model compressed CNNs via imposing low-rank assumption on weight tensors. Moreover, we used an alternative technical tool, and a sharper upper bound on the sample complexity can be obtained.

Our paper makes three main contributions:

1. We formulate CNNs with high-order inputs into statistical models, and show that they have an explicit "Tucker-like" form.

2. The sample complexity analysis can then be conducted to CNNs as well as compressed CNNs via tensor decomposition, with weak conditions allowing for time-dependent inputs like video data.

3. From theoretical analysis, we draw an interesting finding that forcing low dimensionality on output channels may introduce unnecessary parameter redundancy to a compressed network.

## 1.1 COMPARISON WITH OTHER EXISTING WORKS

Deep neural networks are usually over-parametrized, yet empirically, they can generalize well. It is an important topic in the literature to theoretically study the generalization ability of deep neural networks, including deep CNNs (Li et al., 2020; Arora et al., 2018). The generalization error, defined as the difference between test and training errors, is commonly used to evaluate such ability, and many techniques have been developed to control its bound; see, for example, the VC dimension (Vapnik, 2013), the Rademacher complexity & covering number (Bartlett & Mendelson, 2002), the norm-based capacity control (Neyshabur et al., 2017; Golowich et al., 2018; Bartlett et al., 2017; Neyshabur et al., 2015) and low-rank compression based methods (Li et al., 2020; Zhou & Feng, 2018; Arora et al., 2018). These works use a model-agnostic framework, and hence relies heavily on explicit regularization such as weight decay, dropout or data augumentation, as well as algorithm-based implicit regularization to remove the redundancy in the network.

We, however, attempt to theoretically explain how much compressibility is achieved in a compressed network architecture. Specifically, we make a comparison between a CNN and its compressed version, and makes theoretically-supported modification to the latter to further increase efficiency.

Subsequently, our analysis requires an explicit formulation for the network architecture, which is provided in Section 2, and the prediction error at (1) is adopted as our evaluation criteria. We notice that Li et al. (2020) also proposes to use the CP Layers to compress the weights in each convolution layer. But their study is still model-agnostic, since the ranks of the underlying CP Layers depend on the trained weights. In details, their proposed approach uses regularization assumptions on the weights and hence, their derived theoretical bound is influenced by training and not suitable to analyze the network design exclusively.

Other existing works, that aim to provide theoretical understanding for the neural networks, include the understanding of parameter recovery with gradient-based algorithms for deep neural networks (Zhong et al., 2017b; Fu et al., 2020; Goel et al., 2018; Zhong et al., 2017a); the development of other provably efficient algorithms (Cao & Gu, 2019; Du & Goel, 2018); and the investigation of convergence in an over-parameterized regime (Allen-Zhu et al., 2019; Li & Liang, 2018; Du et al., 2018b). Our work differs greatly from these works in both target and methodology. We do not consider computational complexity or algorithm convergence. Instead, we focus on the statistical sample complexity to depict the mechanism of compressed block designs for CNNs.

## 2 FORMULATING CNNS WITH HIGHER-ORDER INPUTS

### 2.1 NOTATION

**Tensor notations.** We follow the notations in Kolda & Bader (2009) to denote vectors by lowercase boldface letters, e.g. $\boldsymbol{a}$; matrices by capital boldface letters, e.g. $\boldsymbol{A}$; tensors of order 3 or higher by Euler script boldface letters, e.g. $\boldsymbol{\mathcal{A}}$. For an $N^{\text{th}}$-order tensor $\boldsymbol{\mathcal{A}} \in \mathbb{R}^{l_1 \times \cdots \times l_N}$, denote its elements by $\boldsymbol{\mathcal{A}}(i_1, i_2, \ldots, i_N)$ and the $n$-mode unfolding by $\boldsymbol{\mathcal{A}}_{(n)}$, where the columns of $\boldsymbol{\mathcal{A}}_{(n)}$ are the $n$-mode vectors of $\boldsymbol{\mathcal{A}}$, for $1 \leq n \leq N$. And $\boldsymbol{\mathcal{A}}(:, \cdots, i_n, \cdots, :)$ denotes the subtensor of $\boldsymbol{\mathcal{A}}$, holding only the $n^{\text{th}}$ index fixed. The vectorization operation is denoted by $\text{vec}(\cdot)$. The inner product of two tensors $\boldsymbol{\mathcal{A}}, \boldsymbol{\mathcal{B}} \in \mathbb{R}^{l_1 \times \cdots \times l_N}$ is defined as $\langle \boldsymbol{\mathcal{A}}, \boldsymbol{\mathcal{B}} \rangle = \sum_{i_1} \cdots \sum_{i_N} \boldsymbol{\mathcal{A}}(i_1, \ldots, i_N) \boldsymbol{\mathcal{B}}(i_1, \ldots, i_N)$, and the Frobenius norm is $\|\boldsymbol{\mathcal{A}}\|_{\text{F}} = \sqrt{\langle \boldsymbol{\mathcal{A}}, \boldsymbol{\mathcal{A}} \rangle}$. The mode-$n$ multiplication $\times_n$ of a tensor $\boldsymbol{\mathcal{A}} \in \mathbb{R}^{l_1 \times \cdots \times l_N}$ and a matrix $\boldsymbol{B} \in \mathbb{R}^{p_n \times l_n}$ is defined as $(\boldsymbol{\mathcal{A}} \times_n \boldsymbol{B})(i_1, \ldots, j_n, \ldots, i_N) = \sum_{i_n=1}^{l_n} \boldsymbol{\mathcal{A}}(i_1, \ldots, i_n, \ldots, i_N) \boldsymbol{B}(j_n, i_n)$, for $1 \leq n \leq N$, respectively. The mode-$n$ multiplication $\bar{\times}_N$ of a tensor $\boldsymbol{\mathcal{A}} \in \mathbb{R}^{l_1 \times \cdots \times l_N}$ and a vector $\boldsymbol{b} \in \mathbb{R}^{l_n}$ is defined as $(\boldsymbol{\mathcal{A}} \bar{\times}_n \boldsymbol{b})(i_1, \ldots, i_{n-1}, i_{n+1}, \ldots, i_N) = \sum_{i_n=1}^{l_n} \boldsymbol{\mathcal{A}}(i_1, \ldots, i_n, \ldots, i_N) \boldsymbol{b}(i_n)$, for $1 \leq n \leq N$. The symbol "$\otimes$" is the Kronecker product and "$\circ$" is the outer product. We extend the definition of Khatri-Rao product to tensors: given tensors $\boldsymbol{\mathcal{A}} \in \mathbb{R}^{l_1 \times l_2 \times \cdots l_N \times K}$ and $\boldsymbol{\mathcal{B}} \in \mathbb{R}^{p_1 \times p_2 \times \cdots p_N \times K}$, their Khatri-Rao product is a tensor of size $\mathbb{R}^{l_1 p_1 \times l_2 p_2 \times \cdots l_N p_N \times K}$, denoted by $\boldsymbol{\mathcal{C}} = \boldsymbol{\mathcal{A}} \odot \boldsymbol{\mathcal{B}}$, where $\boldsymbol{\mathcal{C}}(:, \cdots, k) = \boldsymbol{\mathcal{A}}(:, \cdots, k) \otimes \boldsymbol{\mathcal{B}}(:, \cdots, k), 1 \leq k \leq K$.

**CP decomposition.** The Canonical Polyadic (CP) decomposition (Kolda & Bader, 2009) factorizes the tensor $\boldsymbol{\mathcal{A}} \in \mathbb{R}^{l_1 \times \cdots \times l_N}$ into a sum of rank-1 tensors, i.e. $\boldsymbol{\mathcal{A}} = \sum_{r=1}^R \alpha_r \boldsymbol{h}_r^{(1)} \circ \boldsymbol{h}_r^{(2)} \circ \cdots \circ \boldsymbol{h}_r^{(N)}$, where $\boldsymbol{h}_r^{(j)}$ is a unit norm vector of size $\mathbb{R}^{l_j}$ for all $1 \leq j \leq N$. The CP rank is the number of rank-1 tensors $R$.

**Tucker decomposition.** The Tucker ranks of an $N^{\text{th}}$-order tensor $\boldsymbol{\mathcal{A}} \in \mathbb{R}^{l_1 \times \cdots \times l_N}$ are defined as the matrix ranks of the unfoldings of $\boldsymbol{\mathcal{A}}$ along all modes. If the Tucker ranks of $\boldsymbol{\mathcal{A}}$ are $(R_1, \ldots, R_N)$, then there exist a core tensor $\boldsymbol{\mathcal{G}} \in \mathbb{R}^{R_1 \times \cdots \times R_N}$ and matrices $\boldsymbol{H}^{(i)} \in \mathbb{R}^{l_i \times R_i}$, such that $\boldsymbol{\mathcal{A}} = \boldsymbol{\mathcal{G}} \times_1 \boldsymbol{H}^{(1)} \times_2 \boldsymbol{H}^{(2)} \cdots \times_N \boldsymbol{H}^{(N)}$, known as Tucker decomposition (Tucker, 1966),

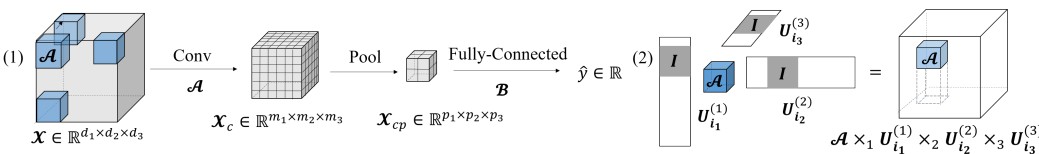

Figure 1: (1) 3-layer CNN for a 3D input $\boldsymbol{\mathcal{X}}$ with one kernel tensor $\boldsymbol{\mathcal{A}}$, average pooling, and fully-connected weights $\boldsymbol{\mathcal{B}}$. (2) $\boldsymbol{U}_{i_j}^{(j)}$'s act as positioning factors. The white spaces indicate zero entries.

## 2.2 BASIC THREE-LAYER CNNS

Consider a three-layer CNN with one convolution, one average pooling and one fully-connected layers, and we assume linear activations for simplicity. Specifically, for a general tensor-structured input $\mathcal{X} \in \mathbb{R}^{d_1 \times d_2 \times \cdots \times d_N}$, we first perform its convolution with an $N^{\text{th}}$-order kernel tensor $\mathcal{A} \in \mathbb{R}^{l_1 \times l_2 \times \cdots \times l_N}$ to get an intermediate output $\mathcal{X}_c \in \mathbb{R}^{m_1 \times m_2 \times \cdots \times m_N}$, and then use average pooling with pooling sizes $(q_1, \cdots, q_N)$ to derive another intermediate output $\mathcal{X}_{cp} \in \mathbb{R}^{p_1 \times p_2 \times \cdots \times p_N}$. Finally, $\mathcal{X}_{cp}$ goes through a fully-connected layer, with weight tensor $\mathcal{B} \in \mathbb{R}^{p_1 \times p_2 \times \cdots \times p_N}$ to produce a scalar output; see Figure 1(1).

We consider the convolution layer with the stride size of $s_c$ along each dimension. Assume that $m_j = (d_j - l_j)/s_c + 1$ are integers for $1 \leq j \leq N$, otherwise zero-padding will be needed. Let

$$\boldsymbol{U}_{i_j}^{(j)} = [\ \underbrace{\boldsymbol{0}}_{(i_j-1)s_c}\ \underbrace{\boldsymbol{I}}_{l_j}\ \underbrace{\boldsymbol{0}}_{d_j-(i_j-1)s_c-l_j}\ ]' \in \mathbb{R}^{d_j \times l_j} \text{ for } 1 \leq i_j \leq m_j,\ 1 \leq j \leq N, \tag{2}$$

which act as positioning factors to stretch the kernel tensor $\mathcal{A}$ into a tensor of the same size as $\mathcal{X}$, while the rest of entries are filled with zeros; see Figure 1(2). As a result, $\mathcal{X}_c$ has entries

$$\mathcal{X}_c(i_1, i_2, \ldots, i_N) = \langle \mathcal{X}, \mathcal{A} \times_1 \boldsymbol{U}_{i_1}^{(1)} \times_2 \boldsymbol{U}_{i_2}^{(2)} \times_3 \cdots \times_N \boldsymbol{U}_{i_N}^{(N)} \rangle.$$

For the pooling layer with the stride size of $s_p$ along each dimension, we assume the pooling sizes $\{q_j\}_{j=1}^N$ satisfy the relationship of $m_j = q_j + (p_j - 1)s_p$, so the sliding windows can be overlapped. But for ease of notation, we can simply take $q_j = m_j/p_j$. The average pooling operation is equivalent to forming $p_1 p_2 p_3 \cdots p_N$ consecutive blocks within $\mathcal{X}_c$, each of size $\mathbb{R}^{q_1 \times q_2 \times \cdots \times q_N}$, and then take the average per block. The resulting tensor $\mathcal{X}_{cp}$ has entries $\mathcal{X}_{cp}(i_1, i_2, \ldots, i_N) = (\prod_j q_j)^{-1} \sum_j \sum_{k_j=(i_j-1)q_j+1}^{i_j q_j} \mathcal{X}_c(k_1, k_2, \ldots, k_N)$, for $1 \leq i_j \leq p_j$ and $1 \leq j \leq N$. If we denote $\boldsymbol{U}_{\mathcal{F},i_j}^{(j)} = q_j^{-1} \sum_{k=(i_j-1)q_j+1}^{i_j q_j} \boldsymbol{U}_k^{(j)}$, then equivalently,

$$\mathcal{X}_{cp}(i_1, i_2, \ldots, i_N) = \langle \mathcal{X}, \mathcal{A} \times_1 \boldsymbol{U}_{\mathcal{F},i_1}^{(1)} \times_2 \boldsymbol{U}_{\mathcal{F},i_2}^{(2)} \times_3 \cdots \times_N \boldsymbol{U}_{\mathcal{F},i_N}^{(N)} \rangle.$$

The fully-connected layer performs a weighted summation over $\mathcal{X}_{cp}$, with weights given by entries of the tensor $\mathcal{B} \in \mathbb{R}^{p_1 \times p_2 \times \cdots \times p_N}$. Denote $\boldsymbol{U}_{\mathcal{F}}^{(j)} = (\boldsymbol{U}_{\mathcal{F},1}^{(j)}, \cdots, \boldsymbol{U}_{\mathcal{F},p_j}^{(j)})$ for $1 \leq j \leq N$, and the predicted output has the form of

$$\widehat{y} = \langle \mathcal{X}_{cp}, \mathcal{B} \rangle = \langle \mathcal{X}, (\mathcal{B} \otimes \mathcal{A}) \times_1 \boldsymbol{U}_{\mathcal{F}}^{(1)} \times_2 \boldsymbol{U}_{\mathcal{F}}^{(2)} \times_3 \cdots \times_N \boldsymbol{U}_{\mathcal{F}}^{(N)} \rangle.$$

Similarly, for a CNN with $K$ kernels, denote $\{\mathcal{A}_k, \mathcal{B}_k\}_{k=1}^K$ to be the set of kernels and fully-connected weights, where $\mathcal{B}_k \in \mathbb{R}^{p_1 \times p_2 \times \cdots \times p_N}$ and $\mathcal{A}_k \in \mathbb{R}^{l_1 \times l_2 \times \cdots \times l_N}$. The model can then be explicitly represented by

$$y^i = \widehat{y}^i + \xi^i = \langle \mathcal{X}^i, \mathcal{W}_X \rangle + \xi^i,\ \ 1 \leq i \leq n, \tag{3}$$

where $\xi^i$ is the additive error, $\mathcal{X}^i \in \mathbb{R}^{d_1 \times d_2 \times \cdots \times d_N}$, and the composite weight tensor

$$\mathcal{W}_X = (\sum_{k=1}^K \mathcal{B}_k \otimes \mathcal{A}_k) \times_1 \boldsymbol{U}_{\mathcal{F}}^{(1)} \times_2 \boldsymbol{U}_{\mathcal{F}}^{(2)} \times_3 \cdots \times_N \boldsymbol{U}_{\mathcal{F}}^{(N)}. \tag{4}$$

The innate weight-sharing compactness of CNN can be equivalently represented as a "Tucker-like" form at (4). The factor matrices $\{\boldsymbol{U}_{\mathcal{F}}^{(j)} \in \mathbb{R}^{d_j \times l_j p_j}\}_{j=1}^N$ are fixed and solely determined by CNN operations on the inputs. They have full column ranks given that $p_j l_j \leq d_j$ always holds. The core tensor is a special Kronecker product that depicts the layer-wise interaction between weights.

## 2.3 SAMPLE COMPLEXITY ANALYSIS

We can now derive the non-asymptotic error bound for the CNN model. Let $\mathcal{Z}^i = \mathcal{X}^i \times_1 \boldsymbol{U}_{\mathcal{F}}^{(1)'} \times_2 \boldsymbol{U}_{\mathcal{F}}^{(2)'} \times_3 \cdots \times_N \boldsymbol{U}_{\mathcal{F}}^{(N)'} \in \mathbb{R}^{l_1 p_1 \times l_2 p_2 \times \cdots \times l_N p_N}$. Model (3) is equivalent to

$$y^i = \langle \mathcal{Z}^i, \mathcal{W} \rangle + \xi^i = \sum_{k=1}^K \langle \mathcal{Z}^i, \mathcal{B}_k \otimes \mathcal{A}_k \rangle + \xi^i, \tag{5}$$

where $\boldsymbol{\mathcal{W}} = \sum_{k=1}^{K} \boldsymbol{\mathcal{B}}_k \otimes \boldsymbol{\mathcal{A}}_k$. The trained weights have the form of $\widehat{\boldsymbol{\mathcal{W}}} = \sum_{k=1}^{K} \widehat{\boldsymbol{\mathcal{B}}}_k \otimes \widehat{\boldsymbol{\mathcal{A}}}_k$, where

$$\{\widehat{\boldsymbol{\mathcal{B}}}_k, \widehat{\boldsymbol{\mathcal{A}}}_k\}_{1 \le k \le K} = \underset{\boldsymbol{\mathcal{B}}_k, \boldsymbol{\mathcal{A}}_k, 1 \le k \le K}{\arg\min} \frac{1}{n} \sum_{i=1}^{n} \left( y^i - \sum_{k=1}^{K} \langle \boldsymbol{\mathcal{Z}}^i, \boldsymbol{\mathcal{B}}_k \otimes \boldsymbol{\mathcal{A}}_k \rangle \right)^2 . \tag{6}$$

Denote $\boldsymbol{x}^i = \text{vec}(\boldsymbol{\mathcal{X}}^i)$ and $\boldsymbol{z}^i = \text{vec}(\boldsymbol{\mathcal{Z}}^i)$, and it can be verified that $\boldsymbol{z}^i = \boldsymbol{U}_G' \boldsymbol{x}^i$, where $\boldsymbol{U}_G = \boldsymbol{U}_{\mathcal{F}}^{(1)} \otimes \{\boldsymbol{U}_{\mathcal{F}}^{(N)} \otimes [\boldsymbol{U}_{\mathcal{F}}^{(N-1)} \otimes \cdots \otimes (\boldsymbol{U}_{\mathcal{F}}^{(3)} \otimes \boldsymbol{U}_{\mathcal{F}}^{(2)})]\}$ represents the CNN operations on the inputs. Let $\bar{\boldsymbol{x}} = (\boldsymbol{x}^{1\prime}, \boldsymbol{x}^{2\prime}, \ldots, \boldsymbol{x}^{n\prime})'$, and we make the following technical assumptions.

**Assumption 1.** *(Time-dependent inputs)* $\bar{\boldsymbol{x}}$ *is normally distributed with mean zero and variance* $\boldsymbol{\Sigma} = \mathbb{E}(\bar{\boldsymbol{x}}\bar{\boldsymbol{x}}')$, *where* $c_x \boldsymbol{I} \le \boldsymbol{\Sigma} \le C_x \boldsymbol{I}$ *for some* $0 < c_x < C_x$.

Denote by $\lambda_{\max}(\boldsymbol{A})$ and $\lambda_{\min}(\boldsymbol{A})$ the maximum and minimum eigenvalues of a symmetric matrix $\boldsymbol{A}$, respectively. When $\{\boldsymbol{x}^i\}$ is a stationary time series with spectral density function $f_X(\theta)$, we can take $c_x = \inf_{-\pi \le \theta \le \pi} \lambda_{\min}(f_X(\theta))$ and $C_x = \sup_{-\pi \le \theta \le \pi} \lambda_{\max}(f_X(\theta))$; see Basu & Michailidis (2015). For independent inputs, $\boldsymbol{\Sigma} = \text{diag}\{\boldsymbol{\Sigma}_{11}, \ldots, \boldsymbol{\Sigma}_{nn}\}$ is block diagonal, and it holds that $c_x = \min_{1 \le j \le n} \lambda_{\min}(\boldsymbol{\Sigma}_{jj})$ and $C_x = \max_{1 \le j \le n} \lambda_{\max}(\boldsymbol{\Sigma}_{jj})$, where $\boldsymbol{\Sigma}_{jj} = \mathbb{E}(\boldsymbol{x}^j \boldsymbol{x}^{j\prime})$.

**Assumption 2.** *(Sub-Gaussian errors)* $\{\xi^i\}$ *are independent* $\sigma^2$*-sub-Gaussian random variables with mean zero, and is independent of* $\{\boldsymbol{\mathcal{X}}^j, 1 \le j \le i\}$ *for all* $1 \le i \le n$.

**Assumption 3.** *(Restricted isometry property)* $c_u \boldsymbol{I} \le \boldsymbol{U}_G' \boldsymbol{U}_G \le C_u \boldsymbol{I}$ *for some* $0 < c_u < C_u$.

Denote $\kappa_U = C_x C_u$ and $\kappa_L = c_x c_u$. Let $\boldsymbol{\mathcal{W}}^* = \sum_{k=1}^{K} \boldsymbol{\mathcal{B}}_k^* \otimes \boldsymbol{\mathcal{A}}_k^*$ be the true weight. The model complexity of CNN at (5) is $d_{\mathcal{M}} = K(P + L + 1)$, where $P = p_1 p_2 \cdots p_N$ and $L = l_1 l_2 \cdots l_N$.

**Theorem 1** (CNN). *Suppose that Assumptions 1-3 hold and* $n \gtrsim (\kappa_U / \kappa_L)^2 d_{\mathcal{M}}$. *Then,*

$$\|\widehat{\boldsymbol{\mathcal{W}}} - \boldsymbol{\mathcal{W}}^*\|_{\text{F}} \le \frac{16 \sigma \sqrt{\alpha_{\text{RSM}}}}{\alpha_{\text{RSC}}} \left( \sqrt{\frac{d_{\mathcal{M}}}{n}} + \sqrt{\frac{\delta}{n}} \right) \quad and, \quad \mathcal{E}(\widehat{\boldsymbol{\mathcal{W}}}) \le \sqrt{\kappa_U} \|\widehat{\boldsymbol{\mathcal{W}}} - \boldsymbol{\mathcal{W}}^*\|_{\text{F}}$$

*with probability* $1 - 4 \exp\{-[c_{\text{H}}(0.25 \kappa_L / \kappa_U)^2 n - 9 d_{\mathcal{M}}]\} - \exp\{-d_{\mathcal{M}} - 8\delta\}$, *where* $\delta = O_p(1)$, $c_{\text{H}}$ *is a positive value,* $\alpha_{\text{RSC}} = \kappa_L / 2$, *and* $\alpha_{\text{RSM}} = 3\kappa_U / 2$.

It can be seen that the prediction error $\mathcal{E}(\widehat{\boldsymbol{\mathcal{W}}})$, is $O_p(\sqrt{d_{\mathcal{M}}/n})$, and hence the sample complexity is of order $O(d_{\mathcal{M}}/\varepsilon^2)$, to achieve a prediction error $\varepsilon$. Technical proofs of all theorems and corollary in the paper are deferred to Appendix A.3.

Note that, for simplicity, we assume a simple regression model at (5). The theoretical analysis can indeed be established for classification problems as well. In details, we provide corresponding theorem and corollaries for both binary and multiclass classification problems in Appendix A.4.

## 2.4 CNNs WITH MORE LAYERS

Consider a 5-layer CNN with "convolution $\to$ pooling $\to$ convolution $\to$ pooling $\to$ fully connected" layers, where the settings for the first "convolution $\to$ pooling" layers are the same as those in Section 2.2.

Denote the kernel for the second convolution by $\widetilde{\boldsymbol{\mathcal{A}}} \in \mathbb{R}^{\tilde{l}_1 \times \tilde{l}_2 \times \cdots \times \tilde{l}_N}$, and the fully-connected weight tensor $\boldsymbol{\mathcal{B}}$ is of size $\mathbb{R}^{\tilde{p}_1 \times \tilde{p}_2 \times \cdots \times \tilde{p}_N}$. We similarly define matrices $\widetilde{\boldsymbol{U}}_{i_j}^{(j)}$ and $\widetilde{\boldsymbol{U}}_{\mathcal{F}, i_j}^{(j)}$, which are both of size $\mathbb{R}^{p_j \times \tilde{l}_j}$ with $1 \le i_j \le \tilde{p}_j$ and $1 \le j \le N$, and can be used to represent the second convolution and pooling operations, respectively. Note that the output from the first pooling layer, $\boldsymbol{\mathcal{X}}_{cp}$, is the input to the second convolution layer, and hence the output from the second pooling layer has entries

$$\widetilde{\boldsymbol{\mathcal{X}}}_{cp}(i_1, i_2, \ldots, i_N) = \langle \boldsymbol{\mathcal{X}}_{cp}, \widetilde{\boldsymbol{\mathcal{A}}} \times_1 \widetilde{\boldsymbol{U}}_{\mathcal{F}, i_1}^{(1)} \times_2 \widetilde{\boldsymbol{U}}_{\mathcal{F}, i_2}^{(2)} \times_3 \cdots \times_N \widetilde{\boldsymbol{U}}_{\mathcal{F}, i_N}^{(N)} \rangle.$$

Stack the matrices $\{\boldsymbol{U}_{\mathcal{F}, i_j}^{(j)} \in \mathbb{R}^{d_j \times l_j}\}_{1 \le i_j \le p_j}$ and $\{\widetilde{\boldsymbol{U}}_{\mathcal{F}, i_j}^{(j)} \in \mathbb{R}^{p_j \times \tilde{l}_j}\}_{1 \le i_j \le \tilde{p}_j}$ into 3D tensors $\boldsymbol{\mathcal{U}}^{(j)} \in \mathbb{R}^{d_j \times l_j \times p_j}$ and $\widetilde{\boldsymbol{\mathcal{U}}}^{(j)} \in \mathbb{R}^{p_j \times \tilde{l}_j \times \tilde{p}_j}$, respectively. Let $\boldsymbol{U}_{\mathcal{DF}}^{(j)} = (\boldsymbol{\mathcal{U}}^{(j)} \times_3 \widetilde{\boldsymbol{\mathcal{U}}}^{(j)})_{(1)}$ for $1 \le j \le N$ and, by some algebra in Appendix A.2, we can show that the predicted output has the form of

$$\widehat{y} = \langle \widetilde{\boldsymbol{\mathcal{X}}}_{cp}, \boldsymbol{\mathcal{B}} \rangle = \langle \boldsymbol{\mathcal{X}}, (\boldsymbol{\mathcal{B}} \otimes \widetilde{\boldsymbol{\mathcal{A}}} \otimes \boldsymbol{\mathcal{A}}) \times_1 \boldsymbol{U}_{\mathcal{DF}}^{(1)} \times_2 \boldsymbol{U}_{\mathcal{DF}}^{(2)} \times_3 \cdots \times_N \boldsymbol{U}_{\mathcal{DF}}^{(N)} \rangle, \tag{7}$$

where $\boldsymbol{\mathcal{U}}^{(j)} \times_3 \widetilde{\boldsymbol{\mathcal{U}}}^{(j)}$ yields a tensor of size $\mathbb{R}^{d_j \times l_j \times \widetilde{l}_j \times \widetilde{p}_j}$, with $\sum_{i=1}^{p_j} \boldsymbol{\mathcal{U}}^{(j)}(k_1, k_2, i)\widetilde{\boldsymbol{\mathcal{U}}}^{(j)}(i, k_3, k_4)$ as its $(k_1, k_2, k_3, k_4)$th entry. The case with multiple kernels will have a form similar to (4). Following the same logic, it can indeed be generalized to an even deeper CNN by adding more "convolution-pooling" layers. Since the composite weight tensor always has a "Tucker-like" form, techniques in proving Theorem 1 can be adopted to conduct the sample complexity analysis for deep CNNs.

Note that, at model (7), the kernels at different convolution layers appear in the form of $\widetilde{\boldsymbol{\mathcal{A}}} \otimes \boldsymbol{\mathcal{A}}$, which actually provides a theoretical justification for exploring the layer-wise low-dimensional structure in the literature. On the other hand, we summarize all weights of a network into one single tensor, akin to Kossaifi et al. (2019). But our summarized tensor has an explicit "nested doll" structure, where the weight structure of the previous layer is nested within that of the current layer.

In practice, zero-padding can be used before each convolution layer to preserve the input size. This is compatible with our framework, since we can propagate the padding of zeros to previous layers and all the way back to the input tensor $\boldsymbol{\mathcal{X}}$.

## 3  COMPRESSED CNNS

For a high-order input, a deep CNN with a larger number of kernels may involve heavy computation, which renders its training difficult for portable devices with limited resources. In real applications, many compressed CNN block designs have been proposed to improve the efficiency, and most of them are based on either matrix factorization or tensor decomposition; see Lebedev et al. (2015); Kim et al. (2016); Astrid & Lee (2017); Kossaifi et al. (2020b).

Tucker decomposition can be used to compress CNNs, which is same as introducing a multilayer CNN block; see Figure 3 in Kim et al. (2016). Specifically, we stack the kernels $\boldsymbol{\mathcal{A}}_k \in \mathbb{R}^{l_1 \times l_2 \times \cdots \times l_N}$ with $1 \leq k \leq K$ into a higher order tensor $\boldsymbol{\mathcal{A}}_{\text{stack}} \in \mathbb{R}^{l_1 \times l_2 \times \cdots \times l_N \times K}$, and assume that $\boldsymbol{\mathcal{A}}_{\text{stack}}$ has Tucker ranks of $(R_1, R_2, \cdots, R_N, R_{N+1})$. As a result,

$$\boldsymbol{\mathcal{A}}_{\text{stack}} = \boldsymbol{\mathcal{G}} \times_1 \boldsymbol{H}^{(1)} \times_2 \boldsymbol{H}^{(2)} \cdots \times_{N+1} \boldsymbol{H}^{(N+1)}, \tag{8}$$

where $\boldsymbol{\mathcal{G}} \in \mathbb{R}^{R_1 \times \cdots \times R_N \times R_{N+1}}$ is the core tensor, and $\boldsymbol{H}^{(j)} \in \mathbb{R}^{l_j \times R_j}$ are factor matrices for $1 \leq j \leq N+1$. From Section 2, to train such a compressed CNN block with linear activations is equivalent to search for the least-square estimators at (6) with $\boldsymbol{\mathcal{A}}_{\text{stack}}$ being constrained to have the form of (8). Denote the estimators by $\{\widehat{\boldsymbol{\mathcal{B}}}_k^{\text{TU}}, \widehat{\boldsymbol{\mathcal{A}}}_k^{\text{TU}}\}_{1 \leq k \leq K}$. The trained weights then have a form of $\widehat{\boldsymbol{\mathcal{W}}}_{\text{TU}} = \sum_{k=1}^K \widehat{\boldsymbol{\mathcal{B}}}_k^{\text{TU}} \otimes \widehat{\boldsymbol{\mathcal{A}}}_k^{\text{TU}}$, and the model complexity is $d_{\mathcal{M}}^{\text{TU}} = \prod_{j=1}^{N+1} R_j + \sum_{i=1}^N l_i R_i + R_{N+1}P$.

CP decomposition is more popular in compressing CNNs; see Kossaifi et al. (2020b); Lebedev et al. (2015); Astrid & Lee (2017). Let $\{\widehat{\boldsymbol{\mathcal{B}}}_k^{\text{CP}}, \widehat{\boldsymbol{\mathcal{A}}}_k^{\text{CP}}\}_{1 \leq k \leq K}$ be the least-square estimators at (6) with $\boldsymbol{\mathcal{A}}_{\text{stack}}$ having a CP decomposition with a rank of $R$. Then, the trained weights are $\widehat{\boldsymbol{\mathcal{W}}}_{\text{CP}} = \sum_{k=1}^K \widehat{\boldsymbol{\mathcal{B}}}_k^{\text{CP}} \otimes \widehat{\boldsymbol{\mathcal{A}}}_k^{\text{CP}}$, and the model complexity is $d_{\mathcal{M}}^{\text{CP}} = R^{N+1} + R(\sum_{i=1}^N l_i + P)$.

**Theorem 2** (Compressed CNN). *Let* $(\widehat{\boldsymbol{\mathcal{W}}}, d_{\mathcal{M}})$ *be* $(\widehat{\boldsymbol{\mathcal{W}}}_{\text{TU}}, d_{\mathcal{M}}^{\text{TU}})$ *for Tucker decomposition, or* $(\widehat{\boldsymbol{\mathcal{W}}}_{\text{CP}}, d_{\mathcal{M}}^{\text{CP}})$ *for CP decomposition. Suppose that Assumptions 1-3 hold and* $n \gtrsim (\kappa_U/\kappa_L)^2 c_N d_{\mathcal{M}}$. *Then,*

$$\|\widehat{\boldsymbol{\mathcal{W}}} - \boldsymbol{\mathcal{W}}^*\|_{\text{F}} \leq \frac{16\sigma\sqrt{\alpha_{\text{RSM}}}}{\alpha_{\text{RSC}}} \left(\sqrt{\frac{c_N d_{\mathcal{M}}}{n}} + \sqrt{\frac{\delta}{n}}\right) \ and, \quad \mathcal{E}(\widehat{\boldsymbol{\mathcal{W}}}) \leq \sqrt{\kappa_U}\|\widehat{\boldsymbol{\mathcal{W}}} - \boldsymbol{\mathcal{W}}^*\|_{\text{F}}$$

*with probability* $1 - 4\exp\{-[c_{\text{H}}(0.25\kappa_L/\kappa_U)^2 n - 3c_N d_{\mathcal{M}}]\} - \exp\{-d_{\mathcal{M}} - 8\delta\}$, *where* $\delta = O_p(1)$, $c_N = 2^{N+1}\log(N+2)$, $c_{\text{H}}$ *is a positive value and* $\alpha_{\text{RSC}}, \alpha_{\text{RSM}}$ *are defined in Theorem 1.*

Theorem 2 shows that, as expected, the sample complexity is proportional to the model complexity. Compared to Theorem 1, we see that the sample complexity of the compressed CNN depends on $\sum_{i=1}^N l_i$ instead of $\prod_{i=1}^N l_i$. When the input dimension $N$ is large, compressed CNN can indeed reduce a large number of parameters.

However, we notice that the sample complexity of a compressed CNN depends on the rank $R_{N+1}$ or $R$, rather than the number of kernels $K$. This differs from the naive parameter counting, and we hence provide the rationale behind this counter-intuitive observation in the corollary below.

**Corollary 1.** *(a) If $\mathcal{A}$ has a Tucker decomposition with the ranks of $(R_1, R_2, \cdots, R_{N+1})$, the CNN is equivalent to one with $R_{N+1}$ kernels, and each kernel $\mathcal{A}_r$, where $1 \leq r \leq R_{N+1}$, has a Tucker decomposition with ranks of $(R_1, R_2, \cdots, R_N)$. Moreover, (b) if the stacked kernel tensor $\mathcal{A}$ has a CP decomposition with rank $R$, the corresponding CNN can be reparameterized into one with $R$ kernels, and each kernel $\mathcal{A}_r$, where $1 \leq r \leq R$, has a CP decomposition form with rank $R$.*

Define the $K/R$ ratio to be the value of $K/R_{N+1}$ for Tucker decomposition or $K/R$ for CP decomposition, and in practice, it always holds that $K \geq R$. Corollary 1 essentially states that when $K/R > 1$, there exists model redundancy in the CNN model with linear activations. In other words, we can obtain a more compressed network by setting $K = R$ such that it has the same sample complexity as the previous one. The $K/R$ ratio appears in various block designs; see Figure 2. And we can use this finding to evaluate their model efficiency.

**(T1) Standard bottleneck block.** The basic building block in ResNet (He et al., 2016) can be exactly replicated by a Tucker decomposition on $\mathcal{A}_{\text{stack}} \in \mathbb{R}^{l_1 \times l_2 \times C \times K}$, with ranks of $(l_1, l_2, R, R)$, where $C$ is the number of input channels (Kossaifi et al., 2020b). As shown by the ablation studies in Session 4, when $K \gg R$, this design may not be the most parameter efficient.

**(T2) Funnel block.** As a straightforward revision on a standard bottleneck block, the funnel block maintains an output channel size of $K = R$. This is hence an efficient block design. Also, when $R = C$, we obtain a normalized funnel block, with the same number of input and output channels.

**(C1) Depthwise separable block.** This block is the basic module in MobileNetV1 (Howard et al., 2017) with a pair of depthwise and pointwise separable convolution layers. It is equivalent to assuming a CP decomposition on $\mathcal{A}_{\text{stack}} \in \mathbb{R}^{l_1 \times l_2 \times C \times K}$ with the rank of $C$ (Kossaifi et al., 2020b). The $1 \times 1$ pointwise convolution expands the numbers of channels from $C$ to $K$. When $K \gg C$, this design is parameter inefficient from a computation viewpoint.

**(C2) Inverted residual block.** Sandler et al. (2018) later proposed this design in MobileNetV2. It includes expansive layers between the input and output layers, with the channel size of $x \cdot C (x \geq 1)$, where $x$ represents the expansion factor. As discussed by Kossaifi et al. (2020b), it can heuristically correspond to a CP decomposition on $\mathcal{A}_{stack}$, with CP rank equals to $x \cdot C$. Since, the rank of output channel dimension can be at most $x \cdot C$, as long as $K \leq x \cdot C$ holds, it is theoretically efficient and provides leeway for exploring thicker layers within blocks.

With nonlinear activations, the model redundancy may bring some benefit. Such benefit is often not guaranteed. And when $K \gg R$, it will, without a doubt, greatly hinder computational efficiency. We will show, in our ablation studies in the next section, that under realistic settings with nonlinear activations, our finding on the $K/R$ ratio still applies. Specifically, networks with $K/R = 1$ maintains comparable performance as networks with $K/R > 1$, while using much less parameters.

## 4 EXPERIMENTS

This section first verifies the theoretical results with synthetic datasets, and then conducts ablation studies on the bottleneck blocks in ResNet (He et al., 2016) with different $K/R$ ratios.

### 4.1 NUMERICAL ANALYSIS FOR THEORETICAL RESULTS

We choose four settings to verify the sample complexity in Theorem 1; see Table 1. The parameter tensors $\{\mathcal{A}_k, \mathcal{B}_k\}_{k=1}^K$ are generated to have standard normal entries. We consider two types of

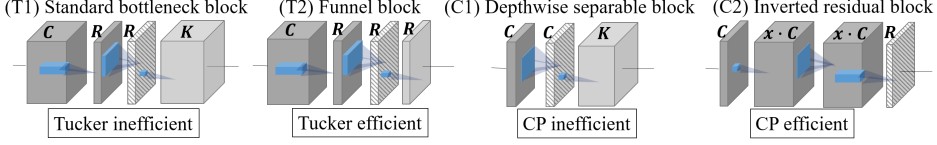

Figure 2: We use thickness of each layer to indicate its relative number of channels. The last layer indicates the output of the block. The dashed layer represents the "bottleneck" of the block.

Table 1: Different settings for verifying Theorem 1 (left), and Theorem 2 (right).

| | Input sizes | Kernel sizes | Pooling sizes | # Kernels | Input sizes | Kernel sizes | Pooling sizes | Tucker ranks | # Kernels |
|---|---|---|---|---|---|---|---|---|---|
| Setting 1 (S1) | (7, 5, 7) | (2, 2, 2) | (3, 2, 3) | 1 | (10, 10, 8, 3) | (5, 5, 3) | (3, 3, 3) | (2, 2, 2, 1) | 2 |
| Setting 2 (S2) | (7, 5, 7) | (2, 2, 2) | (3, 2, 3) | 3 | (10, 10, 8, 3) | (5, 5, 3) | (3, 3, 3) | (2, 2, 2, 1) | 3 |
| Setting 3 (S3) | (8, 8, 3) | (3, 3, 3) | (3, 3, 1) | 1 | (12, 12, 6, 3) | (7, 7, 3) | (3, 3, 2) | (2, 3, 2, 1) | 2 |
| Setting 4 (S4) | (8, 8, 3) | (3, 3, 3) | (3, 3, 1) | 3 | (12, 12, 6, 3) | (7, 7, 3) | (3, 3, 2) | (2, 3, 2, 1) | 3 |

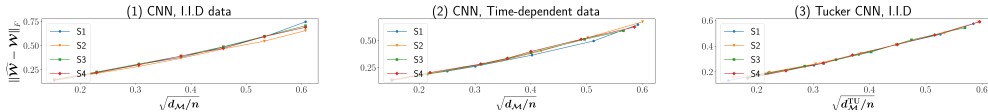

Figure 3: (1)-(2) are the experiment results for Theorem 1 with independent and dependent inputs, respectively. (3) is the experiment results for Tucker CNN in Theorem 2.

inputs: the independent inputs are generated with entries being standard normal random variables, and the time-dependent inputs $\{x^i\}$ are generated from a stationary VAR(1) process. The random additive noises are generated from the standard normal distribution. The number of training samples $n$ varies such that $\sqrt{d_\mathcal{M}/n}$ is equally spaced in the interval $[0.15, 0.60]$, with all other parameters fixed. For each $n$, we generate 200 training sets to calculate the averaged estimation error in Frobenius norm. It can be seen that the estimation error increases linearly with the square root of $d_\mathcal{M}/n$, which is consistent with our finding in Theorem 1.

For Theorem 2, we also adopt four different settings; see Table 1. Here, we consider 4D input tensors. The stacked kernel $\mathcal{A}_{\text{stack}}$ is generated by (8), where the core tensor has standard normal entries and the factor matrices are generated to have orthonormal columns. The number of training samples $n$ is similarly chosen such that $\sqrt{d_\mathcal{M}^{\text{TU}}/n}$ is equally spaced. A linear trend between the estimation error and the square root of $d_\mathcal{M}^{\text{TU}}/n$ can be observed, which verifies Theorem 2.

For details of implementation, we employed the gradient descent method for the optimization with a learning rate of 0.01 and a momentum of 0.9. The procedure is deemed to have reached convergence if the target function drops by less than $10^{-8}$.

### 4.2 ABLATION STUDIES ON COMPUTATIONALLY-EFFICIENT BLOCK DESIGNS

Following the notation in Section 3, we denote by $K$, the number of output channels of a bottleneck block, and by $R$, the Tucker rank of $\mathcal{A}_{\text{stack}}$ along the output channels dimension. Since $K \geq R$, we let $t = K/R$ be the expansive ratio of the block. As $t$ increases, the number of parameters within the block increases. Here, ablation studies are conducted to show that the increase in $t$ does not necessarily lead to the increase in test accuracy. We analyze two image recognition datasets, CIFAR-10 (Krizhevsky et al., 2009) and Street View House Numbers (SVHN) (BNetzer et al., 2011), and an action recognition dataset, UCF101 (Soomro et al., 2012). Standard data pre-processing and augmentation techniques are adopted to all three datasets (He et al., 2016; Hara et al., 2017).

**Network architecture.** Two network architectures are considered in our study; as shown in Figure 4. For image recognition datasets, we adopt a 41-layer Residual Network: it consists of a $3 \times 3$ convolution layer, a max pooling layer, followed by 3 groups of $A + x \times B$ residual blocks with different $R$, and ends with an average pooling and fully-connected layer. Block B represents the bottleneck structure that we are interested in. The standard bottleneck block corresponds to $t = 4$. When $t = 1$, it corresponds to the normalized funnel block in Section 3 (T2). For sake of

Table 2: Test accuracy(%) on CIFAR-10, SVHN and UCF101. For UCF101, we only count the number of parameters and FLOPs for the added blocks.

| $t$ | CIFAR-10 | SVHN | #FLOPs | #Params | UCF101 | #Added FLOPs | #Added Params |
|---|---|---|---|---|---|---|---|
| 1 | 93.53 | 96.30 | 0.17GMac | 5.56M | 79.31 | 0.125GMac | 6.97M |
| 4 | 93.10 | 96.09 | 0.26GMac | 8.55M | 79.36 | 0.145GMac | 9.16M |
| 8 | 93.59 | 96.32 | 0.43GMac | 14.15M | 79.25 | 0.175GMac | 11.11M |
| 16 | 94.03 | 96.31 | 0.87GMac | 29.28M | 79.52 | 0.235GMac | 15.00M |

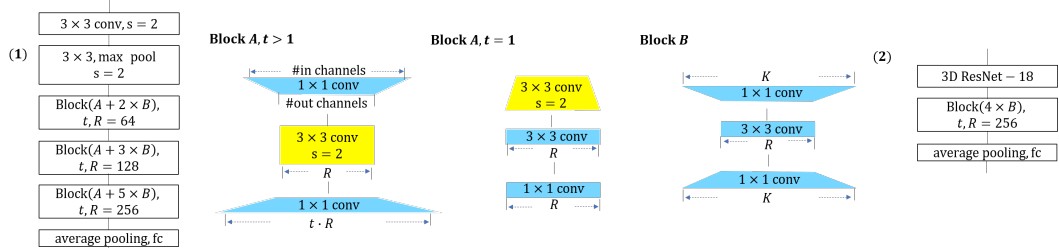

Figure 4: (1) 41-layer Residual Network for CIFAR-10&SVNH. (2) 30-layer 3D Residual Network for UCF101. Here, $t = K/R$ represents the expansion ratio, and takes values of 1,4,8,16.

comparison, we also take $t = 8$ or 16. When $t > 1$, Block A is the standard downsampling block that contains convolution with the stride size of 2. It is inefficient according to our study. So when $t = 1$, we propose minor revisions to simultaneously increase the channels while performing the stride-2 $3 \times 3$ convolution. For UCF101 dataset, we use 3D ResNet-18 as the basic framework and insert 4 stacks of Block B with $R = 256$ before the average pooling and fully-connected layer.

**Results.** The results in Table 2 provide empirical support for our theoretical finding. Though the number of parameters in the network increases with larger $t$, the overall test accuracy remains roughly comparable. For CIFAR-10, it may appear that the test accuracy for $t = 16$ is slightly better than $t = 1$ but it uses 5 times as many parameters.

**Implementation details.** All experiments are conducted in PyTorch and on Tesla V100-DGXS. Following the practice in He et al. (2016), we adopt batch normalization(BN) (Ioffe & Szegedy, 2015) right after each convolution and before the ReLU activation. For network (1), we initialize the weights as in He et al. (2015). We use stochastic gradient descent with weight decay $10^{-4}$, momentum 0.9 and mini-batch size 128. The learning rate starts from 0.1 and is divided by 10 for every 100 epochs. We stop training after 300 epochs, since the training accuracy hardly changes. The training accuracy is approximately 93% for CIFAR-10 and 96% for SVHN. We set seeds 1-5 and report the worst case scenario of the test accuracy in Table 2. For network (2), we follow Hara et al. (2018) and use the Kinetics pretrained 3D ResNet-18. The weights are fixed for the first few layers, and backward propagation is only applied to the added Block B layers and the fully-connected layer. We use stochastic gradient descent with weight decay $10^{-5}$, momentum 0.9 and mini-batch size 64 with 16 frames per clip. The learning rate starts from 0.001 and divided by 10 for every 30 epochs. Training is stopped after 80 epochs and top-1 clip accuracy is around 79% for UCF101.

## 5 CONCLUSION AND DISCUSSION

Our paper proposes a unified theoretical framework that can adopt sample complexity analysis as a tool to study the effective number of parameter in a tensor decomposed CNN block. The main practical takeaways are: (i) For a large kernel tensor $\mathcal{A}$ (Peng et al., 2017), it is always effective to impose low-rank structure on its input dimensions, such as height, width or time-length, etc; and (ii) It is essential to maintain $K/R = 1$ for a CNN block in order to increase accuracy in a more parameter efficient way. In this regard, one can either choose a small $R$ with deeper networks or a larger $R$ with shallower networks. In fact, when $K/R = 1$, the block corresponds exactly to the "straightened" bottleneck in Zagoruyko & Komodakis (2016), who showed in their empirical study that by increasing $R$ in shallow networks, the test accuracy can indeed improve quite significantly.

Since the low-dimensional structure at convolution layers is our focus, tensor decomposition has been applied to the kernel weights $\mathcal{A}_{\text{stack}}$ only in Section 3. Kossaifi et al. (2017) applied tensor decomposition to fully-connected layers since they may contain a large number of parameters as well. Along the line, we can stack the fully-connected weights $\mathcal{B}_k \in \mathbb{R}^{p_1 \times p_2 \times \cdots \times p_N}$ with $1 \leq k \leq K$ into a higher order tensor $\mathcal{B}_{\text{stack}} \in \mathbb{R}^{p_1 \times p_2 \times \cdots \times p_N \times K}$, and further consider a Tensor Contraction Layer (TCL) or Tensor Regression Layer (TRL) as in (Kossaifi et al., 2020a). Similar theoretical analysis can be obtained, and we leave it for future research.

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

# A APPENDIX

This appendix contains five sections. In the first section, we provide details for our mathematical formulation for a high-order 3-layer CNN. In the next section, we discuss the formulation of a 5-layer CNN as an illustrating example of how to add more layers to our framework. A complete version of Theorems 1&2 together with the technical proofs for Theorems 1&2 and Corollary 1 are provided in the third section. In the fourth section, we extend our settings to classification and subsequently establish theorem and corollaries for binary and multiclass classification problem. More implementation details and results for additional experiments are presented in the last section.

## A.1 CNN FORMULATION

For a tensor input $\mathcal{X} \in \mathbb{R}^{d_1 \times \cdots \times d_N}$, it first convolutes with an $N$-dimensional kernel tensor $\mathcal{A} \in \mathbb{R}^{l_1 \times \cdots \times l_N}$ with stride sizes equal to $s_c$, and then performs an average pooling with pooling sizes equal to $(q_1, \cdots, q_N)$. It ends with a fully-connected layer with the weight tensor $\mathcal{B}$ and produces a scalar output. We assume that $m_j = (d_j - l_j)/s_c + 1$ are integers for $1 \leq j \leq N$, otherwise zero-padding will be needed. For ease of notation, we take the pooling sizes $\{q_j\}_{j=1}^N$ to satisfy the relationship $m_j = p_j q_j$.

To duplicate the operation of the convolution layer using a simple mathematical expression, we first need to define a set of matrices $\{\boldsymbol{U}_{i_j}^{(j)} \in \mathbb{R}^{d_j \times l_j}\}$, where

$$\boldsymbol{U}_{i_j}^{(j)} = [\ \underbrace{\boldsymbol{0}}_{(i_j-1)s_c}\ \underbrace{\boldsymbol{I}}_{l_j}\ \underbrace{\boldsymbol{0}}_{d_j-(i_j-1)s_c-l_j}\ ]' \in \mathbb{R}^{d_j \times l_j} \text{ for } 1 \leq i_j \leq m_j,\ 1 \leq j \leq N. \tag{9}$$

$\boldsymbol{U}_{i_j}^{(j)}$ acts as a positioning factor to transform the kernel tensor $\mathcal{A}$ into a tensor of same size as the input $\mathcal{X}$, with the rest of the entries equal to zero.

We are ready to construct our main formulation for a 3-layer tensor CNN. To begin with, we first illustrate the process using a vector input $\boldsymbol{x} \in \mathbb{R}^d$, with a kernel vector $\boldsymbol{a} \in \mathbb{R}^l$; see Fig 5. Using the $i^{\text{th}}$ positioning matrix $\boldsymbol{U}_i^{(1)}$, we can propagate the small kernel vector $\boldsymbol{a}$, into a vector $\boldsymbol{U}_i^{(1)}\boldsymbol{a}$ of size $\mathbb{R}^d$, by filling the rest of the entries with zeros. The intermediate output vector has entries given by $\boldsymbol{x}_c(i) = \langle \boldsymbol{x}, \boldsymbol{U}_i^{(1)}\boldsymbol{a}\rangle$, for $1 \leq i \leq m$. The average pooling operation is equivalent to forming $p$ consecutive vectors within $\boldsymbol{x}_c$, each of size $\mathbb{R}^q$, and then take the average. This results in a vector $\boldsymbol{x}_{cp}$ of size $\mathbb{R}^p$, with $i$th entry equal to $\boldsymbol{x}_{cp}(i) = q^{-1}\sum_{k=(i-1)q+1}^{iq} \boldsymbol{x}_c(i) = \langle \boldsymbol{x}, q^{-1}\sum_{k=(i-1)q+1}^{iq} \boldsymbol{U}_k^{(1)}\boldsymbol{a}\rangle$. The fully-connected layer performs a weighted summation over the $p$ vectors, with weights given by entries of the vector $\boldsymbol{b} \in \mathbb{R}^p$. This gives us the predicted output $\widehat{y} = \langle \boldsymbol{b}, \boldsymbol{x}_{cp}\rangle = \langle \boldsymbol{x}, \boldsymbol{w}_X\rangle$, where

$$\boldsymbol{w}_X = \sum_{i=1}^p b_i \frac{1}{q} \sum_{k=(i-1)q+1}^{iq} \boldsymbol{U}_k^{(1)}\boldsymbol{a} = \boldsymbol{U}_{\mathcal{F}}^{(1)}(\boldsymbol{b} \otimes \boldsymbol{a}) = (\boldsymbol{b} \otimes \boldsymbol{a}) \times_1 \boldsymbol{U}_{\mathcal{F}}^{(1)},$$

and $\boldsymbol{U}_{\mathcal{F}}^{(1)} = q^{-1}(\sum_{k=1}^q \boldsymbol{U}_k^{(1)}, \cdots, \sum_{k=m-q+1}^m \boldsymbol{U}_k^{(1)})$, and "$\times_1$" represents the mode-1 product.

For matrix input $\boldsymbol{X}$ with matrix kernel $\boldsymbol{A}$, however, we need 2 sets of positioning matrices, $\{\boldsymbol{U}_{i_1}^{(1)}\}_{i_1=1}^{m_1}$ and $\{\boldsymbol{U}_{i_2}^{(2)}\}_{i_2=1}^{m_2}$, one for the height dimension and the other for width. Then, the intermediate output from convolution has entries given by $\boldsymbol{X}_c(i_1, i_2) = \langle \boldsymbol{X}, \boldsymbol{U}_{i_1}^{(1)}\boldsymbol{A}\boldsymbol{U}_{i_2}^{(2)\prime}\rangle$, for $1 \leq i_1 \leq p_1$, $1 \leq i_2 \leq p_2$. For the average pooling, we form $p_1 p_2$ consecutive matrices from $\boldsymbol{X}_c$, each of size $\mathbb{R}^{q_1 \times q_2}$ and take the average. This results in $\boldsymbol{X}_{cp} \in \mathbb{R}^{p_1 \times p_2}$, with $\boldsymbol{X}_{cp}(i_1, i_2) = \langle \boldsymbol{X}, (q_1^{-1}\sum_{k_1=(i_1-1)q_1+1}^{i_1 q_1} \boldsymbol{U}_{k_1}^{(1)})\boldsymbol{A}(q_2^{-1}\sum_{k_2=(i_2-1)q_2+1}^{i_2 q_2} \boldsymbol{U}_{k_2}^{(2)})'\rangle$. The output $\boldsymbol{X}_{cp}$ goes through fully-connected layer with weight matrix $\boldsymbol{B}$ and gives the predicted output $\widehat{y} = \langle \boldsymbol{X}, \boldsymbol{W}_X\rangle$, where

$$\boldsymbol{W}_X = \sum_{i_2=1}^{p_2} \sum_{i_1=1}^{p_1} b_{i_1} b_{i_2} \frac{1}{q_1 q_2}(\sum_{k_1=(i_1-1)q_1+1}^{i_1 q_1} \boldsymbol{U}_{k_1}^{(1)})\boldsymbol{A}(\sum_{k_2=(i_2-1)q_2+1}^{i_2 q_2} \boldsymbol{U}_{k_2}^{(2)\prime}) = (\boldsymbol{B} \otimes \boldsymbol{A}) \times_1 \boldsymbol{U}_{\mathcal{F}}^{(1)} \times_2 \boldsymbol{U}_{\mathcal{F}}^{(2)},$$

with $\boldsymbol{U}_{\mathcal{F}}^{(j)} = q_j^{-1}(\sum_{k=1}^{q_j} \boldsymbol{U}_k^{(j)}, \cdots, \sum_{k=m_j-q_j+1}^{m_j} \boldsymbol{U}_k^{(j)})$ for $j = 1$ or 2. And "$\times_1$", "$\times_2$" represent the mode-1 and mode-2 product.

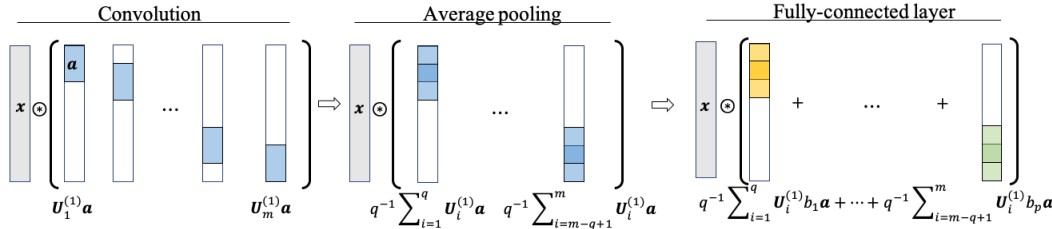

Figure 5: Formulating the process with an input vector $x$. Note that we combine the convolution, pooling and fully connected layers into the composite weight vector $w_X$, where $w_X = q^{-1} \sum_{i=1}^{q} U_i^{(1)} a + \cdots + q^{-1} \sum_{i=m-q+1}^{q} U_i^{(1)} a$. The $\circledast$ represents the convolution operation.

From here, together with the case of high-order tensor input discussed in Session 2.2, we can then derive the form of predicted outcome as $\hat{y} = \langle \mathcal{X}, \mathcal{W}_X^{\text{single}} \rangle$, where

$$\mathcal{W}_X^{\text{single}} = (\mathcal{B} \otimes \mathcal{A}) \times_1 U_{\mathcal{F}}^{(1)} \times_2 U_{\mathcal{F}}^{(2)} \times \cdots \times U_{\mathcal{F}}^{(N)},$$

with $U_{\mathcal{F}}^{(j)} = q_j^{-1}(\sum_{k=1}^{q_j} U_k^{(j)}, \cdots, \sum_{k=m_j-q_j+1}^{m_j} U_k^{(j)})$, and "$\times_j$" represents the mode-$j$ product for $1 \leq j \leq N$.

With $K$ kernels, we denote the set of kernels and the corresponding fully-connected weight tensors as $\{\mathcal{A}_k, \mathcal{B}_k\}_{k=1}^{K}$. Since the convolution and pooling operations are identical across kernels, we can use a summation over kernels to derive the weight tensor for multiple kernels, which is

$$\mathcal{W}_X = (\sum_{k=1}^{K} \mathcal{B}_k \otimes \mathcal{A}_k) \times_1 U_{\mathcal{F}}^{(1)} \times_2 U_{\mathcal{F}}^{(2)} \times \cdots \times U_{\mathcal{F}}^{(N)},$$

and we arrive at the formulation for 3-layer tensor CNN.

## A.2 FIVE-LAYER CNN FORMULATION

Consider a 5-layer CNN with "convolution→pooling→convolution→pooling→fully connected" layers and a 3D tensor input $\mathcal{X} \in \mathbb{R}^{d_1 \times d_2 \times d_3}$. Here, we denote the intermediate output from the first convolution by $\mathcal{X}_c \in \mathbb{R}^{m_1 \times m_2 \times m_3}$, from the second convolution by $\widetilde{\mathcal{X}}_c \in \mathbb{R}^{\tilde{m}_1 \times \tilde{m}_2 \times \tilde{m}_3}$, and the output from the first pooling by $\mathcal{X}_{cp} \in \mathbb{R}^{p_1 \times p_2 \times p_3}$ and from the second pooling by $\widetilde{\mathcal{X}}_{cp} \in \mathbb{R}^{\tilde{p}_1 \times \tilde{p}_2 \times \tilde{p}_3}$. We can first see that the predicted output from the 5-layer CNN is the same as directly feeding $\mathcal{X}_{cp}$ to the second convolution layer, followed by an average pooling layer and a fully-connected layer.

Denote the first convolution kernel tensor by $\mathcal{A} \in \mathbb{R}^{l_1 \times l_2 \times l_3}$, the second convolution kernel tensor by $\mathcal{A} \in \mathbb{R}^{\tilde{l}_1 \times \tilde{l}_2 \times \tilde{l}_3}$, and the fully-connected weight tensor by $\mathcal{B} \in \mathbb{R}^{\tilde{p}_1 \times \tilde{p}_2 \times \tilde{p}_3}$. Define the set of matrices $\{U_{i_j}^{(j)} \in \mathbb{R}^{d_j \times l_j}\}$, for $1 \leq i_j \leq m_j, 1 \leq j \leq N$ as in (2). And similarly, define $\{\widetilde{U}_{i_j}^{(j)} \in \mathbb{R}^{p_j \times \tilde{l}_j}\}$, with

$$\widetilde{U}_{i_j}^{(j)} = [\underbrace{0}_{(i_j-1)\tilde{s}_c} \quad \underbrace{I}_{\tilde{l}_j} \quad \underbrace{0}_{p_j-(i_j-1)\tilde{s}_c-\tilde{l}_j}]', \tag{10}$$

where $\tilde{s}_c$ is the stride size for the second convolution. Let $\widetilde{U}_{\mathcal{F}}^{(j)} = (\widetilde{U}_{\mathcal{F},1}^{(j)}, \cdots, \widetilde{U}_{\mathcal{F},\tilde{p}_j}^{(j)}) = (\tilde{q}_j^{-1} \sum_{k=1}^{\tilde{q}_j} \widetilde{U}_k^{(j)}, \cdots, \tilde{q}_j^{-1} \sum_{k=\tilde{m}_j-\tilde{q}_j+1}^{\tilde{m}_j} \widetilde{U}_k^{(j)})$. We further stack the matrices $\{U_{\mathcal{F},i_j}^{(j)} \in \mathbb{R}^{d_j \times l_j}\}_{1 \leq i_j \leq p_j}$ and $\{\widetilde{U}_{\mathcal{F},i_j}^{(j)} \in \mathbb{R}^{p_j \times \tilde{l}_j}\}_{1 \leq i_j \leq \tilde{p}_j}$ into 3D tensors $\mathcal{U}^{(j)} \in \mathbb{R}^{d_j \times l_j \times p_j}$ and $\widetilde{\mathcal{U}}^{(j)} \in \mathbb{R}^{p_j \times \tilde{l}_j \times \tilde{p}_j}$.

We have that the predicted output

$$
\begin{aligned}
\widehat{y} &= \langle \boldsymbol{\mathcal{B}} \otimes \widetilde{\boldsymbol{\mathcal{A}}}, \boldsymbol{\mathcal{X}}_{cp} \times_1 \widetilde{\boldsymbol{U}}_{\mathcal{F}}^{(1)\prime} \times_2 \widetilde{\boldsymbol{U}}_{\mathcal{F}}^{(2)\prime} \times_3 \widetilde{\boldsymbol{U}}_{\mathcal{F}}^{(3)\prime} \rangle \\
&= \left\langle \boldsymbol{\mathcal{B}} \otimes \widetilde{\boldsymbol{\mathcal{A}}}, \sum_{i_1=1}^{p_1} \sum_{i_2=1}^{p_2} \sum_{i_3=1}^{p_3} \langle \boldsymbol{\mathcal{A}}, \boldsymbol{\mathcal{X}} \times_1 \boldsymbol{U}_{\mathcal{F},i_1}^{(1)\prime} \times_2 \boldsymbol{U}_{\mathcal{F},i_2}^{(2)\prime} \times_3 \boldsymbol{U}_{\mathcal{F},i_3}^{(3)\prime} \rangle \widetilde{\boldsymbol{u}}_{i_1} \circ \widetilde{\boldsymbol{u}}_{i_2} \circ \widetilde{\boldsymbol{u}}_{i_3} \right\rangle \\
&= \left\langle \boldsymbol{\mathcal{B}} \otimes \widetilde{\boldsymbol{\mathcal{A}}} \otimes \boldsymbol{\mathcal{A}}, \sum_{i_1=1}^{p_1} \sum_{i_2=1}^{p_2} \sum_{i_3=1}^{p_3} \boldsymbol{\mathcal{X}} \times_1 (\widetilde{\boldsymbol{u}}_{i_1} \otimes \boldsymbol{U}_{\mathcal{F},i_1}^{(1)\prime}) \times_2 (\widetilde{\boldsymbol{u}}_{i_2} \otimes \boldsymbol{U}_{\mathcal{F},i_2}^{(2)\prime}) \times_3 (\widetilde{\boldsymbol{u}}_{i_3} \otimes \boldsymbol{U}_{\mathcal{F},i_3}^{(3)\prime}) \right\rangle \\
&= \left\langle \boldsymbol{\mathcal{B}} \otimes \widetilde{\boldsymbol{\mathcal{A}}} \otimes \boldsymbol{\mathcal{A}}, \boldsymbol{\mathcal{X}} \times_1 (\boldsymbol{\mathcal{U}}^{(1)} \times_3 \widetilde{\boldsymbol{\mathcal{U}}}^{(1)})_{(1)}' \times_2 (\boldsymbol{\mathcal{U}}^{(2)} \times_3 \widetilde{\boldsymbol{\mathcal{U}}}^{(2)})_{(1)}' \times_3 (\boldsymbol{\mathcal{U}}^{(3)} \times_3 \widetilde{\boldsymbol{\mathcal{U}}}^{(3)})_{(1)}' \right\rangle \\
&= \left\langle \boldsymbol{\mathcal{X}}, (\boldsymbol{\mathcal{B}} \otimes \widetilde{\boldsymbol{\mathcal{A}}} \otimes \boldsymbol{\mathcal{A}}) \times_1 (\boldsymbol{\mathcal{U}}^{(1)} \times_3 \widetilde{\boldsymbol{\mathcal{U}}}^{(1)})_{(1)} \times_2 (\boldsymbol{\mathcal{U}}^{(2)} \times_3 \widetilde{\boldsymbol{\mathcal{U}}}^{(2)})_{(1)} \times_3 (\boldsymbol{\mathcal{U}}^{(3)} \times_3 \widetilde{\boldsymbol{\mathcal{U}}}^{(3)})_{(1)} \right\rangle \\
&= \left\langle \boldsymbol{\mathcal{X}}, (\boldsymbol{\mathcal{B}} \otimes \widetilde{\boldsymbol{\mathcal{A}}} \otimes \boldsymbol{\mathcal{A}}) \times_1 \boldsymbol{U}_{\mathcal{DF}}^{(1)} \times_2 \boldsymbol{U}_{\mathcal{DF}}^{(2)} \times_3 \boldsymbol{U}_{\mathcal{DF}}^{(3)} \right\rangle
\end{aligned}
$$

where $\widetilde{\boldsymbol{u}}_{i_j} = \mathrm{vec}(\widetilde{\boldsymbol{\mathcal{U}}}^{(1)}(i_j,:,:)) \in \mathbb{R}^{\widetilde{l}_j \widetilde{p}_j}$ and $\boldsymbol{U}_{\mathcal{DF}}^{(j)} = (\boldsymbol{\mathcal{U}}^{(j)} \times_3 \widetilde{\boldsymbol{\mathcal{U}}}^{(j)})_{(1)}$, for $1 \leq i_j \leq p_j$ and $1 \leq j \leq 3$.

### A.3 THEORETICAL RESULTS AND TECHNICAL PROOFS

Let $\boldsymbol{\mathcal{Z}}^i = \boldsymbol{\mathcal{X}}^i \times_1 \boldsymbol{U}_{\mathcal{F}}^{(1)\prime} \times_2 \boldsymbol{U}_{\mathcal{F}}^{(2)\prime} \times_3 \cdots \times_N \boldsymbol{U}_{\mathcal{F}}^{(N)\prime} \in \mathbb{R}^{l_1 p_1 \times l_2 p_2 \times \cdots \times l_N p_N}$. The trained weights have the form of $\widehat{\boldsymbol{\mathcal{W}}} = \sum_{k=1}^{K} \widehat{\boldsymbol{\mathcal{B}}}_k \otimes \widehat{\boldsymbol{\mathcal{A}}}_k$, where

$$
\{\widehat{\boldsymbol{\mathcal{B}}}_k, \widehat{\boldsymbol{\mathcal{A}}}_k\}_{1 \leq k \leq K} = \underset{\boldsymbol{\mathcal{B}}_k, \boldsymbol{\mathcal{A}}_k, 1 \leq k \leq K}{\arg\min} \frac{1}{n} \sum_{i=1}^{n} \left( y^i - \sum_{k=1}^{K} \langle \boldsymbol{\mathcal{Z}}^i, \boldsymbol{\mathcal{B}}_k \otimes \boldsymbol{\mathcal{A}}_k \rangle \right)^2 .
$$

Given a test sample $(\boldsymbol{\mathcal{X}}, y)$, where $y = \langle \boldsymbol{\mathcal{X}}, \boldsymbol{\mathcal{W}}_X \rangle + \xi$, and $(\boldsymbol{\mathcal{X}}, \xi)$ satisfies Assumptions 1 and 2. The mean-square training and test error are defined as

$$
\mathrm{err}_{\mathrm{train}}(\widehat{\boldsymbol{\mathcal{W}}}) = \sqrt{\frac{1}{n} \sum_{i=1}^{n} \left( y^i - \langle \boldsymbol{\mathcal{X}}^i, \widehat{\boldsymbol{\mathcal{W}}}_X \rangle \right)^2} \text{ and } \mathrm{err}(\widehat{\boldsymbol{\mathcal{W}}}) = \sqrt{\mathbb{E}_{(\boldsymbol{\mathcal{X}},y)} \left( y - \langle \boldsymbol{\mathcal{X}}, \widehat{\boldsymbol{\mathcal{W}}}_X \rangle \right)^2} .
$$

Denote $\kappa_U = C_x C_u$ and $\kappa_L = c_x c_u$. Let $\boldsymbol{\mathcal{W}}^* = \sum_{k=1}^{K} \boldsymbol{\mathcal{B}}_k^* \otimes \boldsymbol{\mathcal{A}}_k^*$ be the true weights, and $\|\boldsymbol{\mathcal{W}}\|_n^2 := n^{-1} \sum_{i=1}^{n} (\langle \boldsymbol{\mathcal{Z}}^i, \boldsymbol{\mathcal{W}} \rangle)^2$ be the empirical norm with respect to $\boldsymbol{\mathcal{W}}$. The model complexity of CNN at equation (1) or (2) is $d_{\mathcal{M}} = K(P + L + 1)$.

**Theorem 1** (Complete Version). *Suppose that Assumptions 1-3 hold and $n \gtrsim (\kappa_U/\kappa_L)^2 d_{\mathcal{M}}$. Then, for some $\delta > 0$,*

$$
(a) \quad \|\widehat{\boldsymbol{\mathcal{W}}} - \boldsymbol{\mathcal{W}}^*\|_{\mathrm{F}} \leq \frac{16\sigma\sqrt{\alpha_{\mathrm{RSM}}}}{\alpha_{\mathrm{RSC}}} \left( \sqrt{\frac{d_{\mathcal{M}}}{n}} + \sqrt{\frac{\delta}{n}} \right)
$$

$$
(b) \quad \|\widehat{\boldsymbol{\mathcal{W}}} - \boldsymbol{\mathcal{W}}^*\|_n \leq \frac{16\sigma\alpha_{\mathrm{RSM}}}{\alpha_{\mathrm{RSC}}} \left( \sqrt{\frac{d_{\mathcal{M}}}{n}} + \sqrt{\frac{\delta}{n}} \right)
$$

$$
(c) \quad \mathrm{err}^2(\widehat{\boldsymbol{\mathcal{W}}}) \leq \left( \frac{16\sigma\alpha_{\mathrm{RSM}}}{\alpha_{\mathrm{RSC}}} \right)^2 \left( \frac{d_{\mathcal{M}}}{n} + \frac{\delta}{n} \right) + \sigma^2,
$$

*with probability $1 - 4\exp\{-[c_{\mathrm{H}}(0.25\kappa_L/\kappa_U)^2 n - 9d_{\mathcal{M}}]\} - \exp\{-d_{\mathcal{M}} - 8\delta\}$, where $c_{\mathrm{H}}$ is a positive value, $\alpha_{\mathrm{RSC}} = \kappa_L/2$, and $\alpha_{\mathrm{RSM}} = 3\kappa_U/2$.*

It can be seen that the estimation error, $\|\widehat{\boldsymbol{\mathcal{W}}} - \boldsymbol{\mathcal{W}}^*\|_{\mathrm{F}}$, is $O_p(\sqrt{d_{\mathcal{M}}/n})$, and hence the sample complexity is of order $O(d_{\mathcal{M}}/\varepsilon^2)$, to achieve a training error $\varepsilon$. The empirical norm is equivalent to the training error minus the sample standard deviation of noise. And we can see that the training error $\mathrm{err}_{\mathrm{train}}(\widehat{\boldsymbol{\mathcal{W}}}) = O_p(\sqrt{d_{\mathcal{M}}/n}) + \widehat{\sigma}$ and test error $\mathrm{err}(\widehat{\boldsymbol{\mathcal{W}}}) = O_p(\sqrt{d_{\mathcal{M}}/n}) + \sigma$, where $\widehat{\sigma}$ and $\sigma$ represent the sample and population standard deviation of noise, respectively.

Now we proceed to provide a complete version of the sample complexity of compressed CNNs. Let $d_{\mathcal{M}}^{\mathrm{TU}} = \prod_{j=1}^{N+1} R_j + \sum_{i=1}^{N} l_i R_i + R_{N+1} P$ and $d_{\mathcal{M}}^{\mathrm{CP}} = R^{N+1} + R(\sum_{i=1}^{N} l_i + P)$.

**Theorem 2** (Complete Version). *Let $(\widehat{\mathcal{W}}, d_{\mathcal{M}})$ be $(\widehat{\mathcal{W}}_{\mathrm{TU}}, d_{\mathcal{M}}^{\mathrm{TU}})$ for Tucker decomposition, or $(\widehat{\mathcal{W}}_{\mathrm{CP}}, d_{\mathcal{M}}^{\mathrm{CP}})$ for CP decomposition. Suppose that Assumptions 1-3 hold and $n \gtrsim (\kappa_U/\kappa_L)^2 c_N d_{\mathcal{M}}$. Then, for some $\delta > 0$,*

$$(a) \|\widehat{\mathcal{W}} - \mathcal{W}^*\|_{\mathrm{F}} \leq \frac{16\sigma\sqrt{\alpha_{\mathrm{RSM}}}}{\alpha_{\mathrm{RSC}}} \left( \sqrt{\frac{c_N d_{\mathcal{M}}}{n}} + \sqrt{\frac{\delta}{n}} \right),$$

$$(b) \|\widehat{\mathcal{W}} - \mathcal{W}^*\|_n \leq \frac{16\sigma\alpha_{\mathrm{RSM}}}{\alpha_{\mathrm{RSC}}} \left( \sqrt{\frac{c_N d_{\mathcal{M}}}{n}} + \sqrt{\frac{\delta}{n}} \right),$$

$$(c) err^2(\widehat{\mathcal{W}}) \leq \left( \frac{16\sigma\alpha_{\mathrm{RSM}}}{\alpha_{\mathrm{RSC}}} \right)^2 \left( \frac{c_N d_{\mathcal{M}}}{n} + \frac{\delta}{n} \right) + \sigma^2,$$

*with probability $1 - 4\exp\{-[c_{\mathrm{H}}(0.25\kappa_L/\kappa_U)^2 n - 3c_N d_{\mathcal{M}}]\} - \exp\{-d_{\mathcal{M}} - 8\delta\}$, where $c_N = 2^{N+1}\log(N+2)$, $c_{\mathrm{H}}$ is a positive value and $\alpha_{\mathrm{RSC}}, \alpha_{\mathrm{RSM}}$ are defined in Theorem 1.*

Because CP decomposition can be considered as a special case of Tucker decomposition where $R_j = R$, for $1 \leq j \leq N+1$. We only provide the proof for Tucker CNN.

### A.3.1 PROOF OF THEOREM 1

Denote the sets $\widehat{\mathcal{S}}_K = \{\sum_{k=1}^{K} \mathcal{B}_k \otimes \mathcal{A}_k : \mathcal{A}_k \in \mathbb{R}^{l_1 \times l_2 \times \cdots \times l_N} \text{ and } \mathcal{B}_k \in \mathbb{R}^{p_1 \times p_2 \times \cdots \times p_N}\}$ and $\mathcal{S}_K = \{\mathcal{W} \in \widehat{\mathcal{S}}_K : \|\mathcal{W}\|_{\mathrm{F}} = 1\}$. Let $\widehat{\mathbf{\Delta}} = \widehat{\mathcal{W}} - \mathcal{W}^*$, and then

$$\frac{1}{n} \sum_{i=1}^{n} (y^i - \langle \mathcal{Z}^i, \widehat{\mathcal{W}} \rangle)^2 \leq \frac{1}{n} \sum_{i=1}^{n} (y^i - \langle \mathcal{Z}^i, \mathcal{W}^* \rangle)^2,$$

which implies that

$$\|\widehat{\mathbf{\Delta}}\|_n^2 \leq \frac{2}{n} \sum_{i=1}^{n} \xi^i \langle \mathcal{Z}^i, \widehat{\mathbf{\Delta}} \rangle \leq 2\|\widehat{\mathbf{\Delta}}\|_{\mathrm{F}} \sup_{\mathbf{\Delta} \in \mathcal{S}_{2K}} \frac{1}{n} \sum_{i=1}^{n} \xi^i \langle \mathcal{Z}^i, \mathbf{\Delta} \rangle, \tag{11}$$

where $\|\mathbf{\Delta}\|_n^2 := n^{-1} \sum_{i=1}^{n} (\langle \mathcal{Z}^i, \mathbf{\Delta} \rangle)^2$ is the empirical norm with respect to $\mathbf{\Delta}$, and $\widehat{\mathbf{\Delta}} \in \widehat{\mathcal{S}}_{2K}$.

Consider a $\varepsilon$-net $\bar{\mathcal{S}}_{2K}$, with the cardinality of $\mathcal{N}(2K, \varepsilon)$, for the set $\mathcal{S}_{2K}$. For any $\mathbf{\Delta} \in \mathcal{S}_{2K}$, there exists a $\bar{\mathbf{\Delta}}_j \in \bar{\mathcal{S}}_{2K}$ such that $\|\mathbf{\Delta} - \bar{\mathbf{\Delta}}_j\|_{\mathrm{F}} \leq \varepsilon$. Note that $\mathbf{\Delta} - \bar{\mathbf{\Delta}}_j \in \widehat{\mathcal{S}}_{4K}$ and, from Lemma 1(a), we can further find $\mathbf{\Delta}_1, \mathbf{\Delta}_2 \in \widehat{\mathcal{S}}_{2K}$ such that $\langle \mathbf{\Delta}_1, \mathbf{\Delta}_2 \rangle = 0$ and $\mathbf{\Delta} - \bar{\mathbf{\Delta}}_j = \mathbf{\Delta}_1 + \mathbf{\Delta}_2$. It then holds that $\|\mathbf{\Delta}_1\|_{\mathrm{F}} + \|\mathbf{\Delta}_2\|_{\mathrm{F}} \leq \sqrt{2}\|\mathbf{\Delta} - \bar{\mathbf{\Delta}}_j\|_{\mathrm{F}} \leq \sqrt{2}\varepsilon$ since $\|\mathbf{\Delta} - \bar{\mathbf{\Delta}}_j\|_{\mathrm{F}}^2 = \|\mathbf{\Delta}_1\|_{\mathrm{F}}^2 + \|\mathbf{\Delta}_2\|_{\mathrm{F}}^2$. As a result,

$$\frac{1}{n} \sum_{i=1}^{n} \xi^i \langle \mathcal{Z}^i, \mathbf{\Delta} \rangle = \frac{1}{n} \sum_{i=1}^{n} \xi^i \langle \mathcal{Z}^i, \bar{\mathbf{\Delta}}_j \rangle + \frac{1}{n} \sum_{i=1}^{n} \xi^i \langle \mathcal{Z}^i, \mathbf{\Delta}_1 \rangle + \frac{1}{n} \sum_{i=1}^{n} \xi^i \langle \mathcal{Z}^i, \mathbf{\Delta}_2 \rangle$$

$$\leq \max_{1 \leq j \leq \mathcal{N}(2K, \varepsilon)} \frac{1}{n} \sum_{i=1}^{n} \xi^i \langle \mathcal{Z}^i, \bar{\mathbf{\Delta}}_j \rangle + \sqrt{2}\varepsilon \sup_{\mathbf{\Delta} \in \mathcal{S}_{2K}} \frac{1}{n} \sum_{i=1}^{n} \xi^i \langle \mathcal{Z}^i, \mathbf{\Delta} \rangle,$$

which leads to

$$\sup_{\mathbf{\Delta} \in \mathcal{S}_{2K}} \frac{1}{n} \sum_{i=1}^{n} \xi^i \langle \mathcal{Z}^i, \mathbf{\Delta} \rangle \leq (1 - \sqrt{2}\varepsilon)^{-1} \max_{1 \leq j \leq \mathcal{N}(2K, \varepsilon)} \frac{1}{n} \sum_{i=1}^{n} \xi^i \langle \mathcal{Z}^i, \bar{\mathbf{\Delta}}_j \rangle. \tag{12}$$

Note that, from Lemma 1(b), $\log \mathcal{N}(2K, \varepsilon) \leq 2d_{\mathcal{M}} \log(9/\varepsilon)$, where $d_{\mathcal{M}} = K(P + L + 1)$. Let $\varepsilon = (2\sqrt{2})^{-1}$, and then $8 - 2\log(9/\varepsilon) > 1$. As a result, by (12) and Lemma 3,

$$
\mathbb{P}\left[\sup_{\boldsymbol{\Delta} \in \mathcal{S}_{2K}} \frac{1}{n}\sum_{i=1}^{n} \xi^i \langle \boldsymbol{\mathcal{Z}}^i, \boldsymbol{\Delta}\rangle \geq 8\sigma\sqrt{\alpha_{\mathrm{RSM}}}\left(\sqrt{\frac{d_{\mathcal{M}}}{n}} + \sqrt{\frac{\delta}{n}}\right), \quad \sup_{\boldsymbol{\Delta} \in \mathcal{S}_{2K}} \|\boldsymbol{\Delta}\|_n^2 \leq \alpha_{\mathrm{RSM}}\right]
$$

$$
\leq \mathbb{P}\left[\max_{1 \leq j \leq \mathcal{N}(2K, \varepsilon)} \frac{1}{n}\sum_{i=1}^{n} \xi^i \langle \boldsymbol{\mathcal{Z}}^i, \bar{\boldsymbol{\Delta}}_j\rangle \geq 4\sigma\sqrt{\alpha_{\mathrm{RSM}}}\left(\sqrt{\frac{d_{\mathcal{M}}}{n}} + \sqrt{\frac{\delta}{n}}\right), \quad \sup_{\boldsymbol{\Delta} \in \mathcal{S}_{2K}} \|\boldsymbol{\Delta}\|_n^2 \leq \alpha_{\mathrm{RSM}}\right]
$$

$$
\leq \sum_{j=1}^{\mathcal{N}(2K, \varepsilon)} \mathbb{P}\left(\frac{1}{n}\sum_{i=1}^{n} \xi^i \langle \boldsymbol{\mathcal{Z}}^i, \bar{\boldsymbol{\Delta}}_j\rangle \geq 4\sigma\sqrt{\alpha_{\mathrm{RSM}}}\left(\sqrt{\frac{d_{\mathcal{M}}}{n}} + \sqrt{\frac{\delta}{n}}\right), \quad \|\bar{\boldsymbol{\Delta}}_j\|_n^2 \leq \alpha_{\mathrm{RSM}}\right)
$$

$$
\leq \exp\{-d_{\mathcal{M}} - 8\delta\}.
$$

$$(13)$$

Note that

$$
\mathbb{P}\left[\sup_{\boldsymbol{\Delta} \in \mathcal{S}_{2K}} \frac{1}{n}\sum_{i=1}^{n} \xi^i \langle \boldsymbol{\mathcal{Z}}^i, \boldsymbol{\Delta}\rangle \geq 8\sigma\sqrt{\alpha_{\mathrm{RSM}}}\left(\sqrt{\frac{d_{\mathcal{M}}}{n}} + \sqrt{\frac{\delta}{n}}\right)\right]
$$

$$
\leq \mathbb{P}\left[\sup_{\boldsymbol{\Delta} \in \mathcal{S}_{2K}} \frac{1}{n}\sum_{i=1}^{n} \xi^i \langle \boldsymbol{\mathcal{Z}}^i, \boldsymbol{\Delta}\rangle \geq 8\sigma\sqrt{\alpha_{\mathrm{RSM}}}\left(\sqrt{\frac{d_{\mathcal{M}}}{n}} + \sqrt{\frac{\delta}{n}}\right), \quad \sup_{\boldsymbol{\Delta} \in \mathcal{S}_{2K}} \|\boldsymbol{\Delta}\|_n^2 \leq \alpha_{\mathrm{RSM}}\right]
$$

$$
+ \mathbb{P}\left(\sup_{\boldsymbol{\Delta} \in \mathcal{S}_{2K}} \|\boldsymbol{\Delta}\|_n^2 \geq \alpha_{\mathrm{RSM}}\right).
$$

From (13) and Lemma 2, we then have that, with probability $1 - 4\exp\{-[c_{\mathrm{H}}(0.25\kappa_L/\kappa_U)^2 n - 9d_{\mathcal{M}}]\} - \exp\{-d_{\mathcal{M}} - 8\delta\}$,

$$
\sup_{\boldsymbol{\Delta} \in \mathcal{S}_{2K}} \frac{1}{n}\sum_{i=1}^{n} \xi^i \langle \boldsymbol{\mathcal{Z}}^i, \boldsymbol{\Delta}\rangle \leq 8\sigma\sqrt{\alpha_{\mathrm{RSM}}}\left(\sqrt{\frac{d_{\mathcal{M}}}{n}} + \sqrt{\frac{\delta}{n}}\right) \quad \text{and} \quad \|\boldsymbol{\Delta}\|_n^2 \geq \alpha_{\mathrm{RSC}}\|\boldsymbol{\Delta}\|_{\mathrm{F}}^2,
$$

which, together with (11), leads to

$$
\|\widehat{\boldsymbol{\Delta}}\|_{\mathrm{F}} \leq \frac{16\sigma\sqrt{\alpha_{\mathrm{RSM}}}}{\alpha_{\mathrm{RSC}}}\left(\sqrt{\frac{d_{\mathcal{M}}}{n}} + \sqrt{\frac{\delta}{n}}\right) \quad \text{and} \quad \|\widehat{\boldsymbol{\Delta}}\|_n \leq 16\sigma\frac{\alpha_{\mathrm{RSM}}}{\alpha_{\mathrm{RSC}}}\left(\sqrt{\frac{d_{\mathcal{M}}}{n}} + \sqrt{\frac{\delta}{n}}\right).
$$

Finally we prove (c). Given a test sample $(\boldsymbol{\mathcal{X}}, y)$, and let $\boldsymbol{\mathcal{Z}} = \boldsymbol{\mathcal{X}} \times_1 \boldsymbol{U}_{\mathcal{F}}^{(1)\prime} \times_2 \boldsymbol{U}_{\mathcal{F}}^{(2)\prime} \times_3 \cdots \times_N \boldsymbol{U}_{\mathcal{F}}^{(N)\prime}$. It holds that

$$
\mathbb{E}(y - \langle \boldsymbol{\mathcal{X}}, \boldsymbol{\mathcal{W}}_X\rangle)^2 = \mathbb{E}(y - \langle \boldsymbol{\mathcal{Z}}, \boldsymbol{\mathcal{W}}\rangle)^2 = \mathbb{E}(\langle \boldsymbol{\mathcal{Z}}, \boldsymbol{\Delta}\rangle)^2 + \sigma^2,
$$

and

$$
\mathbb{E}(\langle \boldsymbol{\mathcal{Z}}, \boldsymbol{\Delta}\rangle)^2 = \boldsymbol{\Delta}'\mathbb{E}(\bar{z}\bar{z}')\boldsymbol{\Delta} = \boldsymbol{\Delta}'\boldsymbol{U}_G\mathbb{E}(\bar{x}\bar{x}')\boldsymbol{U}_G'\boldsymbol{\Delta} \leq \kappa_U\|\boldsymbol{\Delta}\|_2^2.
$$

As a result, from (a) of this theorem, we have

$$
\mathrm{err}^2(\widehat{\boldsymbol{\mathcal{W}}}) \leq \left(\frac{16\sigma\alpha_{\mathrm{RSM}}}{\alpha_{\mathrm{RSC}}}\right)^2\left(\frac{d_{\mathcal{M}}}{n} + \frac{\delta}{n}\right) + \sigma^2.
$$

This accomplishes the proof.

### A.3.2 PROOF OF THEOREM 2

Denote the sets $\widehat{\mathcal{S}}^{\mathrm{TU}}(R_1, \cdots, R_{N+1}) = \{\sum_{k=1}^{K} \boldsymbol{\mathcal{B}}_k \otimes \boldsymbol{\mathcal{A}}_k : \text{the stacked kernel } \boldsymbol{\mathcal{A}}_{\mathrm{stack}} \in \mathbb{R}^{l_1 \times l_2 \times \cdots \times l_N \times K} \text{ has the ranks of } (R_1, ..., R_N, R_{N+1}) \text{ and } \boldsymbol{\mathcal{B}}_k \in \mathbb{R}^{p_1 \times p_2 \times \cdots \times p_N}\}$, $\mathcal{S}_{2K}^{\mathrm{TU}} = \{\boldsymbol{\mathcal{W}}_1 + \boldsymbol{\mathcal{W}}_2 : \boldsymbol{\mathcal{W}}_1, \boldsymbol{\mathcal{W}}_2 \in \widehat{\mathcal{S}}^{\mathrm{TU}}(R_1, \cdots, R_{N+1})\}$, and $\mathcal{S}_{2K}^{\mathrm{TU}} = \{\boldsymbol{\mathcal{W}} \in \widehat{\mathcal{S}}_{2K}^{\mathrm{TU}} : \|\boldsymbol{\mathcal{W}}\|_{\mathrm{F}} = 1\}$. Note that $\widehat{\boldsymbol{\mathcal{W}}}_{\mathrm{TU}}, \boldsymbol{\mathcal{W}}^* \in \widehat{\mathcal{S}}^{\mathrm{TU}}(R_1, \cdots, R_{N+1})$, and $\widehat{\boldsymbol{\Delta}} = \widehat{\boldsymbol{\mathcal{W}}}_{\mathrm{TU}} - \boldsymbol{\mathcal{W}}^* \in \mathcal{S}_{2K}^{\mathrm{TU}}$.

We first consider a $\varepsilon$-net for $\mathcal{S}_{2K}^{\mathrm{TU}}$. For each $1 \leq k \leq K$, Let $\boldsymbol{b}_k = (b_{k1}, \ldots, b_{kP})' = \mathrm{vec}(\boldsymbol{\mathcal{B}}_k)$, and we can rearrange $\boldsymbol{\mathcal{B}}_k \otimes \boldsymbol{\mathcal{A}}_k$ into the form of $\boldsymbol{\mathcal{A}}_k \circ \boldsymbol{b}_k$, which is a tensor of size $\mathbb{R}^{l_1 \times l_2 \times \cdots \times l_N \times P}$, where $P = p_1 p_2 \cdots p_N$. Denote $\boldsymbol{B} = (b_{kj}) \in \mathbb{R}^{P \times K}$, and it holds that

$$\sum_{k=1}^{K} \boldsymbol{\mathcal{A}}_k \circ \boldsymbol{b}_k = \boldsymbol{\mathcal{A}} \times_{N+1} \boldsymbol{B} = \boldsymbol{\mathcal{G}} \times_1 \boldsymbol{H}^{(1)} \times_2 \boldsymbol{H}^{(2)} \cdots \times_{N+1} \widetilde{\boldsymbol{B}},$$

which is a tensor with the size of $\mathbb{R}^{l_1 \times l_2 \times \cdots \times l_N \times P}$ and the ranks of $(R_1, \ldots, R_N, R_{N+1})$ where $\widetilde{\boldsymbol{B}} = \boldsymbol{B}\boldsymbol{H}^{(N+1)} \in \mathbb{R}^{P \times R_{N+1}}$. Essentially, in this step, we rewrite the model into one with $R_{k+1}$ kernels instead. Specifically, we now have $\boldsymbol{\mathcal{W}} = \sum_{r=1}^{R_{N+1}} \widetilde{\boldsymbol{\mathcal{A}}}_r \otimes \widetilde{\boldsymbol{\mathcal{B}}}_r$, where $\widetilde{\boldsymbol{\mathcal{A}}}_r = \boldsymbol{\mathcal{G}}_r \times_1 \boldsymbol{H}^{(1)} \times_2 \boldsymbol{H}^{(2)} \cdots \times_N \boldsymbol{H}^{(N)}$ with $\boldsymbol{\mathcal{G}}_r = \boldsymbol{\mathcal{G}}(:, :, \cdots, r)$ and the $r$-th column of $\widetilde{\boldsymbol{B}}$ is the vectorization of $\widetilde{\boldsymbol{\mathcal{B}}}_r$ for $1 \leq r \leq R_{N+1}$.

As a result, $\widehat{\mathcal{S}}_{2K}^{\mathrm{TU}}$ consists of tensors with the ranks of $(2R_1, \ldots, 2R_N, 2R_{N+1})$ at most.

Denote $\mathcal{S}_{\mathrm{Tucker}}(r_1, \cdots, r_N, r_{N+1}) = \{\boldsymbol{\mathcal{T}} \in \mathbb{R}^{l_1 \times \cdots \times l_N \times l_{N+1}} : \|\boldsymbol{\mathcal{T}}\|_{\mathrm{F}} = 1, \boldsymbol{\mathcal{T}}$ has the Tucker ranks of $(r_1, \cdots, r_N, r_{N+1})\}$, where $l_{N+1} = P = p_1 p_2 \cdots p_N$. Then the $\varepsilon$-covering number for $\mathcal{S}_{2K}^{\mathrm{TU}}$ satisfies

$$|\mathcal{S}_{2K}^{\mathrm{TU}}| \leq |\mathcal{S}_{\mathrm{Tucker}}(2R_1, \cdots, 2R_N, 2R_{N+1})|.$$

For each $\boldsymbol{\mathcal{T}} \in \mathcal{S}_{\mathrm{Tucker}}(r_1, \cdots, r_N, r_{N+1})$, we have

$$\boldsymbol{\mathcal{T}} = \boldsymbol{\mathcal{G}} \times_1 \boldsymbol{U}^{(1)} \times_2 \cdots \times_{N+1} \boldsymbol{U}^{(N+1)},$$

where $\boldsymbol{\mathcal{G}} \in \mathbb{R}^{r_1 \times \cdots \times r_N \times r_{N+1}}$ with $\|\boldsymbol{\mathcal{G}}\|_{\mathrm{F}} = 1$, and $\boldsymbol{U}^{(i)} \in \mathbb{R}^{l_i \times r_i}$ with $1 \leq i \leq N+1$ are orthonormal matrices. We now construct an $\varepsilon$-net for $\mathcal{S}_{\mathrm{Tucker}}(r_1, \cdots, r_N, r_{N+1})$ by covering the sets of $\boldsymbol{\mathcal{G}}$ and all $\boldsymbol{U}^{(i)}$s, and the proof hinges on the covering number of low-multilinear-rank tensors in Wang et al. (2019). Treating $\boldsymbol{\mathcal{G}}$ as $\prod_{j=1}^{N+1} r_j$-dimensional vector with $\|\boldsymbol{\mathcal{G}}\|_{\mathrm{F}} = 1$, we can find an $\varepsilon/(N+2)$-net for it, denoted by $\bar{G}$, with the cardinality of $|\bar{G}| \leq (3(N+2)/\varepsilon)^{\prod_{j=1}^{N+1} r_j}$.

Next, let $O_{n,r} = \{\boldsymbol{U} \in \mathbb{R}^{n \times r} : \boldsymbol{U}^\top \boldsymbol{U} = I_r\}$. To cover $O_{n,r}$, it is beneficial to use the $\|\cdot\|_{1,2}$ norm, defined as

$$\|\boldsymbol{X}\|_{1,2} = \max_i \|\boldsymbol{X}_i\|_2,$$

where $\boldsymbol{X}_i$ denotes the $i^{\mathrm{th}}$ column of $\boldsymbol{X}$. Let $Q_{n,r} = \{\boldsymbol{X} \in \mathbb{R}^{n \times r} : \|\boldsymbol{X}\|_{1,2} \leq 1\}$. One can easily check that $O_{n,r} \subset Q_{n,r}$, and then an $\varepsilon/(N+2)$-net $\bar{O}_{n,r}$ for $O_{n,r}$ has the cardinality of $|\bar{O}_{n,r}| \leq (3(N+2)/\varepsilon)^{nr}$. Denote $\bar{\mathcal{S}}_{\mathrm{Tucker}}(r_1, \cdots, r_N, r_{N+1}) = \{\boldsymbol{\mathcal{G}} \times_1 \boldsymbol{U}^{(1)} \times_2 \cdots \times_{N+1} \boldsymbol{U}^{(N+1)} : \boldsymbol{\mathcal{G}} \in \bar{G}, \boldsymbol{U}^{(i)} \in \bar{O}_{l_i, r_i}, 1 \leq i \leq N+1\}$. By a similar argument presented in Lemma A.1 of Wang et al. (2019), we can show that $\bar{\mathcal{S}}_{\mathrm{Tucker}}(r_1, \cdots, r_N, r_{N+1})$ is a $\varepsilon$-net for the set $\mathcal{S}_{\mathrm{Tucker}}(r_1, \cdots, r_N, r_{N+1})$ with the cardinality of

$$\left(\frac{3N+6}{\varepsilon}\right)^{\prod_{j=1}^{N+1} r_j + \sum_{j=1}^{N+1} l_j r_j}.$$

where $l_{N+1} = P$. Thus, the $\varepsilon$-covering number of $\mathcal{S}_{2K}^{\mathrm{TU}}$ is

$$\mathcal{N}^{\mathrm{TU}}(\varepsilon) = |\mathcal{S}_{\mathrm{Tucker}}(2R_1, \cdots, 2R_N, 2R_{N+1})| \leq \left(\frac{3N+6}{\varepsilon}\right)^{2^{N+1} \prod_{j=1}^{N+1} R_j + 2\sum_{i=1}^{N} l_i R_i + 2R_{N+1} P}.$$

Let $\varepsilon = 1/2$. It then holds that $\log(3(N+2)/\varepsilon) < 3\log(N+2)$ and $\log \mathcal{N}^{\mathrm{TU}}(\varepsilon) \leq 2^{N+1} d_{\mathcal{M}}^{\mathrm{TU}} \log[(3N+6)/\varepsilon]$, where $d_{\mathcal{M}}^{\mathrm{TU}} = \prod_{j=1}^{N+1} R_j + \sum_{i=1}^{N} l_i R_i + R_{N+1} P$. By a method similar to the proof of Lemma 2, we can show that

$$\mathbb{P}\left\{\sup_{\boldsymbol{\Delta} \in \mathcal{S}_{2K}^{\mathrm{TU}}} \|\boldsymbol{\Delta}\|_n^2 \geq \alpha_{\mathrm{RSM}}\right\} \leq 2\exp\left\{-c_{\mathrm{H}} n \left(\frac{\kappa_L}{4\kappa_U}\right)^2 + 3c_N d_{\mathcal{M}}^{\mathrm{TU}}\right\}, \tag{14}$$

and

$$\mathbb{P}\left\{\inf_{\boldsymbol{\Delta} \in \mathcal{S}_{2K}^{\mathrm{TU}}} \|\boldsymbol{\Delta}\|_n^2 \leq \alpha_{\mathrm{RSC}}\right\} \leq 2\exp\left\{-c_{\mathrm{H}} n \left(\frac{\kappa_L}{4\kappa_U}\right)^2 + 3c_N d_{\mathcal{M}}^{\mathrm{TU}}\right\}. \tag{15}$$

where $c_N = 2^{N+1} \log(N + 2)$. Moreover, similar to equation 13, by applying Lemma 3, we can show that

$$
\begin{aligned}
&\mathbb{P}\left[\sup_{\boldsymbol{\Delta}\in\mathcal{S}_{2K}^{\mathrm{TU}}} \frac{1}{n}\sum_{i=1}^{n} \xi^i \langle \boldsymbol{\mathcal{Z}}^i, \boldsymbol{\Delta}\rangle \geq 8\sigma\sqrt{\alpha_{\mathrm{RSM}}}\left(\sqrt{\frac{c_N d_{\mathcal{M}}^{\mathrm{TU}}}{n}} + \sqrt{\frac{\delta}{n}}\right), \quad \sup_{\boldsymbol{\Delta}\in\mathcal{S}_{2K}^{\mathrm{TU}}} \|\boldsymbol{\Delta}\|_n^2 \leq \alpha_{\mathrm{RSM}}\right] \\
&\leq \mathbb{P}\left[\max_{1\leq j\leq \mathcal{N}^{\mathrm{TU}}(\varepsilon)} \frac{1}{n}\sum_{i=1}^{n} \xi^i \langle \boldsymbol{\mathcal{Z}}^i, \bar{\boldsymbol{\Delta}}_j\rangle \geq 4\sigma\sqrt{\alpha_{\mathrm{RSM}}}\left(\sqrt{\frac{c_N d_{\mathcal{M}}^{\mathrm{TU}}}{n}} + \sqrt{\frac{\delta}{n}}\right), \quad \sup_{\boldsymbol{\Delta}\in\mathcal{S}_{2K}^{\mathrm{TU}}} \|\boldsymbol{\Delta}\|_n^2 \leq \alpha_{\mathrm{RSM}}\right] \\
&\leq \sum_{j=1}^{\mathcal{N}^{\mathrm{TU}}(\varepsilon)} \mathbb{P}\left(\frac{1}{n}\sum_{i=1}^{n} \xi^i \langle \boldsymbol{\mathcal{Z}}^i, \bar{\boldsymbol{\Delta}}_j\rangle \geq 4\sigma\sqrt{\alpha_{\mathrm{RSM}}}\left(\sqrt{\frac{c_N d_{\mathcal{M}}^{\mathrm{TU}}}{n}} + \sqrt{\frac{\delta}{n}}\right), \quad \|\bar{\boldsymbol{\Delta}}_j\|_n^2 \leq \alpha_{\mathrm{RSM}}\right) \\
&\leq \exp\{-d_{\mathcal{M}}^{\mathrm{TU}} - 8\delta\},
\end{aligned}
\tag{16}
$$

where $\varepsilon = (2\sqrt{2})^{-1}$ and $2^{N+4}\log(N+2) - 2^{N+1}\log(3(N+1)/\varepsilon) > 1$. By a method similar to the proof of Theorem 1, we can show that, with probability $1 - 4\exp\{-[c_{\mathrm{H}}(0.25\kappa_L/\kappa_U)^2 n - 3c_N \log(N+2)d_{\mathcal{M}}^{\mathrm{TU}}]\} - \exp\{-d_{\mathcal{M}}^{\mathrm{TU}} - 8\delta\}$,

$$
\sup_{\boldsymbol{\Delta}\in\mathcal{S}_{2K}^{\mathrm{TU}}} \frac{1}{n}\sum_{i=1}^{n} \xi^i \langle \boldsymbol{\mathcal{Z}}^i, \boldsymbol{\Delta}\rangle \leq 8\sigma\sqrt{\alpha_{\mathrm{RSM}}}\left(\sqrt{\frac{c_N d_{\mathcal{M}}^{\mathrm{TU}}}{n}} + \sqrt{\frac{\delta}{n}}\right) \quad \text{and} \quad \|\boldsymbol{\Delta}\|_n^2 \geq \alpha_{\mathrm{RSC}}\|\boldsymbol{\Delta}\|_{\mathrm{F}}^2,
$$

which, together with equation 11, leads to

$$
\|\widehat{\boldsymbol{\Delta}}\|_{\mathrm{F}} \leq \frac{16\sigma\sqrt{\alpha_{\mathrm{RSM}}}}{\alpha_{\mathrm{RSC}}}\left(\sqrt{\frac{c_N d_{\mathcal{M}}^{\mathrm{TU}}}{n}} + \sqrt{\frac{\delta}{n}}\right),
$$

$$
\|\widehat{\boldsymbol{\Delta}}\|_n \leq 16\sigma\frac{\alpha_{\mathrm{RSM}}}{\alpha_{\mathrm{RSC}}}\left(\sqrt{\frac{c_N d_{\mathcal{M}}^{\mathrm{TU}}}{n}} + \sqrt{\frac{\delta}{n}}\right),
$$

$$
\mathrm{err}^2(\widehat{\boldsymbol{\mathcal{W}}}_{\mathrm{TU}}) \leq \left(\frac{16\sigma\alpha_{\mathrm{RSM}}}{\alpha_{\mathrm{RSC}}}\right)^2\left(\frac{c_N d_{\mathcal{M}}^{\mathrm{TU}}}{n} + \frac{\delta}{n}\right) + \sigma^2.
$$

We accomplished the proof.

### A.3.3    PROOFS OF COROLLARY 1

When the kernel tensor $\boldsymbol{\mathcal{A}}$ has a Tucker decomposition form of $\boldsymbol{\mathcal{A}} = \boldsymbol{\mathcal{G}} \times_1 \boldsymbol{H}^{(1)} \times_2 \boldsymbol{H}^{(2)} \times_3 \cdots \times_{N+1} \boldsymbol{H}^{(N+1)}$, and the multilinear rank is $(R_1, R_2, \cdots, R_{N+1})$. Let $\boldsymbol{\mathcal{H}} = \boldsymbol{\mathcal{G}} \times_1 \boldsymbol{H}^{(1)} \times_2 \boldsymbol{H}^{(2)} \times_3 \cdots \times_N \boldsymbol{H}^{(N)}$, which is a tensor of size $\mathbb{R}^{l_1 \times \cdots \times l_N \times R_{N+1}}$. The mode-$(N+1)$ unfolding of $\boldsymbol{\mathcal{H}}$ is a matrix with $R_{N+1}$ row vectors, each of size $\mathbb{R}^L$, and we denote them by $\boldsymbol{g}_1, \cdots, \boldsymbol{g}_{R_{N+1}}$. Fold $\boldsymbol{g}_r$s back into tensors, i.e. $\boldsymbol{g}_r = \mathrm{vec}(\widetilde{\boldsymbol{\mathcal{H}}}_r)$, where $\widetilde{\boldsymbol{\mathcal{H}}}_r \in \mathbb{R}^{l_1 \times \cdots \times l_N}$ and $1 \leq r \leq R_{N+1}$. Moreover, let $\boldsymbol{H}^{(N+1)} = (\boldsymbol{h}_1^{(N+1)}, \cdots, \boldsymbol{h}_K^{(N+1)})'$, where $\boldsymbol{h}_k^{(N+1)}$ is a vector of size $\mathbb{R}^{R_{N+1}}$ and we denote its $r^{\mathrm{th}}$ entry as $h_{k,r}^{(N+1)}$, for $1 \leq r \leq R_{N+1}$. It can be verified that, for each $1 \leq k \leq K$,

$$
\boldsymbol{\mathcal{A}}_k = \boldsymbol{\mathcal{G}} \times_1 \boldsymbol{H}^{(1)} \times_2 \boldsymbol{H}^{(2)} \cdots \times_N \boldsymbol{H}^{(N)} \bar{\times}_{N+1} \boldsymbol{h}_k^{(N+1)} = \boldsymbol{\mathcal{H}} \bar{\times}_{N+1} \boldsymbol{h}_k^{(N+1)},
$$

and hence,

$$
\sum_{k=1}^{K} \boldsymbol{\mathcal{B}}_k \otimes \boldsymbol{\mathcal{A}}_k = \sum_{r=1}^{R_{N+1}} \sum_{k=1}^{K} \boldsymbol{\mathcal{B}}_k \otimes \left(\widetilde{\boldsymbol{\mathcal{H}}}_r \cdot h_{k,r}^{(N+1)}\right) = \sum_{r=1}^{R_{N+1}} \left(\sum_{k=1}^{K} h_{k,r}^{(N+1)} \boldsymbol{\mathcal{B}}_k\right) \otimes \widetilde{\boldsymbol{\mathcal{H}}}_r.
$$

By letting $\widetilde{\boldsymbol{\mathcal{B}}}_r = \sum_{k=1}^{K} h_{k,r}^{(N+1)} \boldsymbol{\mathcal{B}}_k$ and $\widetilde{\boldsymbol{\mathcal{A}}}_r = \widetilde{\boldsymbol{\mathcal{H}}}_r$ for $1 \leq r \leq R_{N+1}$, we can reformulate the model into

$$
y^i = \langle \sum_{r=1}^{R_{N+1}} \widetilde{\boldsymbol{\mathcal{B}}}_r \otimes \widetilde{\boldsymbol{\mathcal{A}}}_r, \boldsymbol{\mathcal{Z}}^i\rangle + \xi^i,
$$

and the proof of (a) is then accomplished.

Suppose that the kernel tensor $\mathcal{A}$ has a CP decomposition form of $\mathcal{A} = \sum_{r=1}^{R} \alpha_r \boldsymbol{h}_r^{(1)} \circ \boldsymbol{h}_r^{(2)} \circ \cdots \circ \boldsymbol{h}_r^{(N+1)}$, where $\boldsymbol{h}_r^{(j)} \in \mathbb{R}^{l_j}$ and $\|\boldsymbol{h}_r^{(j)}\| = 1$ for all $1 \leq j \leq N$. Moreover, $\boldsymbol{h}_r^{(N+1)} = (h_{r,1}^{(N+1)}, ..., h_{r,K}^{(N+1)})' \in \mathbb{R}^K$ and $\|\boldsymbol{h}_r^{(N+1)}\| = 1$. Note that

$$\mathcal{A}_k = \sum_{r=1}^{R} \alpha_r h_{r,k}^{(N+1)} \boldsymbol{h}_r^{(1)} \circ \boldsymbol{h}_r^{(2)} \circ \cdots \circ \boldsymbol{h}_r^{(N)},$$

for all $1 \leq k \leq K$, and hence

$$\sum_{k=1}^{K} \mathcal{B}_k \otimes \mathcal{A}_k = \sum_{r=1}^{R} \sum_{k=1}^{K} \mathcal{B}_k \otimes \left( \alpha_r h_{r,k}^{(N+1)} \boldsymbol{h}_r^{(1)} \circ \boldsymbol{h}_r^{(2)} \circ \cdots \circ \boldsymbol{h}_r^{(N)} \right)$$

$$= \sum_{r=1}^{R} \left( h_{r,k}^{(N+1)} \mathcal{B}_k \right) \otimes \left( \alpha_r \boldsymbol{h}_r^{(1)} \circ \boldsymbol{h}_r^{(2)} \circ \cdots \circ \boldsymbol{h}_r^{(N)} \right).$$

By letting $\widetilde{\mathcal{B}}_r = h_{r,k}^{(N+1)} \mathcal{B}_k$ and $\widetilde{\mathcal{A}}_r = \alpha_r \boldsymbol{h}_r^{(1)} \circ \boldsymbol{h}_r^{(2)} \circ \cdots \circ \boldsymbol{h}_r^{(N)}$ for all $1 \leq r \leq R$, we can reparameterize the model into

$$y^i = \langle \sum_{r=1}^{R} \widetilde{\mathcal{B}}_r \otimes \widetilde{\mathcal{A}}_r, \boldsymbol{\mathcal{Z}}^i \rangle + \xi^i,$$

and the proof of (b) is then accomplished.

### A.3.4 LEMMAS USED IN THE PROOFS OF THEOREMS

**Lemma 1.** *(Partition and covering number of restricted spaces) Consider* $\widehat{\mathcal{S}}_K = \{\sum_{k=1}^{K} \mathcal{B}_k \otimes \mathcal{A}_k : \mathcal{A}_k \in \mathbb{R}^{l_1 \times l_2 \times \cdots \times l_N}$ *and* $\mathcal{B}_k \in \mathbb{R}^{p_1 \times p_2 \times \cdots \times p_N}\}$ *and* $\mathcal{S}_K = \{\mathcal{W} \in \widehat{\mathcal{S}}_K : \|\mathcal{W}\|_{\mathrm{F}} = 1\}$.

  (a) *For any* $\boldsymbol{\Delta} \in \widehat{\mathcal{S}}_{2K}$, *there exist* $\mathcal{W}_1, \mathcal{W}_2 \in \widehat{\mathcal{S}}_K$ *such that* $\boldsymbol{\Delta} = \mathcal{W}_1 + \mathcal{W}_2$ *and* $\langle \mathcal{W}_1, \mathcal{W}_2 \rangle = 0$.

  (b) *The* $\varepsilon$-*covering number of the set* $\mathcal{S}_K$ *is*

$$\mathcal{N}(K, \varepsilon) \leq (9/\varepsilon)^{K(L+P+1)},$$

  *where* $L = l_1 l_2 \cdots l_N$ *and* $P = p_1 p_2 \cdots p_N$.

*Proof.* For each $K$, we first define a map $T_K : \widehat{\mathcal{S}}_K \to \mathbb{R}^{P \times L}$. For any $\mathcal{W} = \sum_{k=1}^{K} \mathcal{B}_k \otimes \mathcal{A}_k \in \widehat{\mathcal{S}}_K$, we define that $T_K(\mathcal{W}) = \sum_{k=1}^{K} \mathrm{vec}(\mathcal{B}_k) \mathrm{vec}(\mathcal{A}_k)' \in \mathbb{R}^{P \times L}$, which has the rank of at most $K$.

For any $\boldsymbol{\Delta} \in \widehat{\mathcal{S}}_{2K}$, the rank of matrix $T_K(\boldsymbol{\Delta})$ is at most $2K$. As shown in Figure 6, we can split the singular value decomposition (SVD) of $T_K(\boldsymbol{\Delta})$ into two parts, and it can be verified that $T_K(\boldsymbol{\Delta}) = \boldsymbol{U}_1 \boldsymbol{\Lambda}_1 \boldsymbol{V}_1' + \boldsymbol{U}_2 \boldsymbol{\Lambda}_2 \boldsymbol{V}_2'$ and $\langle \boldsymbol{U}_1 \boldsymbol{\Lambda}_1 \boldsymbol{V}_1', \boldsymbol{U}_2 \boldsymbol{\Lambda}_2 \boldsymbol{V}_2' \rangle = 0$. Denote $\boldsymbol{U}_i = (\boldsymbol{u}_1^{(i)}, \boldsymbol{u}_2^{(i)}, \ldots, \boldsymbol{u}_K^{(i)})$ and $\boldsymbol{V}_i \boldsymbol{\Lambda}_i' = (\boldsymbol{v}_1^{(i)}, \boldsymbol{v}_2^{(i)}, \ldots, \boldsymbol{v}_K^{(i)})$ for $i = 1$ and 2. We then fold $\boldsymbol{u}_j^{(i)}$s and $\boldsymbol{v}_j^{(i)}$s into tensors, i.e. $\boldsymbol{u}_k^{(i)} = \mathrm{vec}(\mathcal{B}_k^{(i)})$ and $\boldsymbol{v}_k^{(i)} = \mathrm{vec}(\mathcal{A}_k^{(i)})$ with $1 \leq k \leq K$. Let $\mathcal{W}_1 = \sum_{k=1}^{K} \mathcal{B}_k^{(1)} \otimes \mathcal{A}_k^{(1)}$ and $\mathcal{W}_2 = \sum_{k=1}^{K} \mathcal{B}_k^{(2)} \otimes \mathcal{A}_k^{(2)}$. It then can be verified that $\mathcal{W}_1, \mathcal{W}_2 \in \widehat{\mathcal{S}}_K$, $\boldsymbol{\Delta} = \mathcal{W}_1 + \mathcal{W}_2$ and $\langle \mathcal{W}_1, \mathcal{W}_2 \rangle = 0$. Thus, we accomplish the proof of (a).

Denote by $\mathcal{S}_{\mathrm{matrix}} \subset \mathbb{R}^{P \times L}$ the set of matrices with unit Frobenius norm and rank at most $K$. Note that $T(\mathcal{S}_K) \subset \mathcal{S}_{\mathrm{matrix}}$, while the $\varepsilon$-covering number for $\mathcal{S}_{\mathrm{matrix}}$ is

$$|\mathcal{S}_{\mathrm{matrix}}| \leq (9/\varepsilon)^{K(L+P+1)};$$

see Candes & Plan (2011). This accomplishes the proof of (b).

$\square$

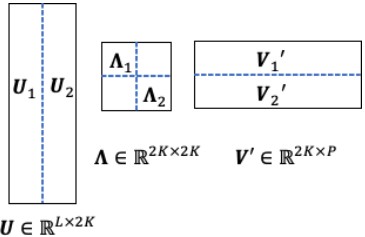

Figure 6: Splitting matrix $T(\mathbf{\Delta})$ based on its singular value decomposition.

**Lemma 2.** *(Restricted strong convexity and smoothness) Suppose that $n \gtrsim (\kappa_U/\kappa_L)^2 d_{\mathcal{M}}$. It holds that, with probability at least $1 - 4\exp\{-[c_{\mathrm{H}}(0.25\kappa_L/\kappa_U)^2 n - 9d_{\mathcal{M}}]\}$,*

$$\alpha_{\mathrm{RSC}} \|\mathbf{\Delta}\|_{\mathrm{F}}^2 \leq \frac{1}{n} \sum_{i=1}^{n} (\langle \mathbf{Z}^i, \mathbf{\Delta} \rangle)^2 \leq \alpha_{\mathrm{RSM}} \|\mathbf{\Delta}\|_{\mathrm{F}}^2 ,$$

*for all $\mathbf{\Delta} \in \widehat{\mathcal{S}}_{2K}$, where $\widehat{\mathcal{S}}_{2K}$ is defined in Lemma 1, $c_{\mathrm{H}}$ is a positive constant from the Hanson-Wright inequality, $\alpha_{\mathrm{RSC}} = \kappa_L/2$ and $\alpha_{\mathrm{RSM}} = 3\kappa_U/2$.*

*Proof.* It is sufficient to show that $\sup_{\mathbf{\Delta} \in \mathcal{S}_{2K}} \|\mathbf{\Delta}\|_n^2 \leq \alpha_{\mathrm{RSM}}$ and $\inf_{\mathbf{\Delta} \in \mathcal{S}_{2K}} \|\mathbf{\Delta}\|_n^2 \geq \alpha_{\mathrm{RSC}}$ hold with a high probability, where $\|\mathbf{\Delta}\|_n^2 := n^{-1} \sum_{i=1}^{n} (\langle \mathbf{Z}^i, \mathbf{\Delta} \rangle)^2$ is the empirical norm, and $\mathcal{S}_{2K}$ is defined in Lemma 1. Without confusion, in this proof, we will also use the notation of $\mathbf{\Delta}$ for its vectorized version, $\mathrm{vec}(\mathbf{\Delta})$. Note that

$$
\begin{aligned}
\|\mathbf{\Delta}\|_n^2 &= \frac{1}{n} \sum_{i=1}^{n} (\langle \mathbf{Z}^i, \mathbf{\Delta} \rangle)^2 = \frac{1}{n} \sum_{i=1}^{n} \mathbf{z}^{i\prime} \mathbf{\Delta}\mathbf{\Delta}' \mathbf{z}^i = \frac{1}{n} \sum_{i=1}^{n} \mathbf{x}^{i\prime} \mathbf{U}_G \mathbf{\Delta}\mathbf{\Delta}' \mathbf{U}_G' \mathbf{x}^i \\
&= \frac{1}{n} \bar{\mathbf{x}}' \Big[ \mathbf{I}_n \otimes (\mathbf{U}_G \mathbf{\Delta}\mathbf{\Delta}' \mathbf{U}_G') \Big] \bar{\mathbf{x}} = \bar{\mathbf{w}}' \mathbf{Q} \bar{\mathbf{w}},
\end{aligned}
\tag{17}
$$

where $\mathbf{Q} = n^{-1} \mathbf{\Sigma}^{\frac{1}{2}} [\mathbf{I}_n \otimes (\mathbf{U}_G \mathbf{\Delta}\mathbf{\Delta}' \mathbf{U}_G')] \mathbf{\Sigma}^{\frac{1}{2}}$, $\bar{\mathbf{w}} \in \mathbb{R}^{nD}$ is a random vector with *i.i.d.* standard normal entries, $\mathbf{z}^i = \mathbf{U}_G' \mathbf{x}^i$, $\mathbf{x}^i = \mathrm{vec}(\mathbf{X}^i)$, $\bar{\mathbf{x}} = (\mathbf{x}^{1\prime}, \mathbf{x}^{2\prime}, \dots, \mathbf{x}^{n\prime})'$, and $\mathbf{\Sigma} = \mathbb{E}(\bar{\mathbf{x}}\bar{\mathbf{x}}') \in \mathbb{R}^{nD \times nD}$.

Denote $\mathbf{w}_G = \mathbf{U}_G \mathbf{\Delta} \in \mathbb{R}^D$ and $\mathcal{S}^{n-1} = \{\mathbf{v} \in \mathbb{R}^n : \|\mathbf{v}\|_2 = 1\}$. Let $\lambda_{\max}(\mathbf{\Sigma}_{ii})$ be the maximum eigenvalue of $\mathbf{\Sigma}_{ii} = \mathbb{E}(\mathrm{vec}(\mathbf{X}^i)\,\mathrm{vec}(\mathbf{X}^i)')$ for $1 \leq i \leq n$, and it holds that $\lambda_{\max}(\mathbf{\Sigma}_{ii}) \leq C_x$ for all $1 \leq i \leq n$. For matrix $\mathbf{Q}$, we have

$$
\mathrm{tr}(\mathbf{Q}) = \frac{1}{n} \mathrm{tr}\left( \mathbf{\Sigma}^{\frac{1}{2}} [\mathbf{I}_n \otimes \mathbf{w}_G][\mathbf{I}_n \otimes \mathbf{w}_G'] \mathbf{\Sigma}^{\frac{1}{2}} \right) = \frac{1}{n} \mathrm{tr}\left( [\mathbf{I}_n \otimes \mathbf{w}_G'] \mathbf{\Sigma} [\mathbf{I}_n \otimes \mathbf{w}_G] \right) = \frac{1}{n} \sum_{i=1}^{n} (\mathbf{w}_G' \mathbf{\Sigma}_{ii} \mathbf{w}_G)
$$

$$
\leq \frac{1}{n} \sum_{i=1}^{n} \left( \sup_{\mathbf{w}_G \in \mathcal{S}^{PL-1}} \frac{\mathbf{w}_G' \mathbf{\Sigma}_{ii} \mathbf{w}_G}{\mathbf{w}_G' \mathbf{w}_G} \right) \left( \sup_{\mathbf{\Delta} \in \mathcal{S}^{PL-1}} \mathbf{\Delta}' \mathbf{U}_G' \mathbf{U}_G \mathbf{\Delta} \right) \leq \frac{1}{n} \sum_{i=1}^{n} \lambda_{\max}(\mathbf{\Sigma}_{ij}) C_u \leq \kappa_U,
\tag{18}
$$

and similarly, we can show that $\mathrm{tr}(\mathbf{Q}) \geq \kappa_L$, where $\kappa_L = c_x c_u$, $\kappa_U = C_x C_u$ and $\kappa_L \leq \kappa_U$. Thus,

$$\kappa_L \leq \mathbb{E} \|\mathbf{\Delta}\|_n^2 = \mathrm{tr}(\mathbf{Q}) \leq \kappa_U. \tag{19}$$

Moreover, denote $\mathbf{Q}_1 = n^{-\frac{1}{2}} \mathbf{\Sigma}^{\frac{1}{2}} [\mathbf{I}_n \otimes \mathbf{w}_G]$ and note that $\mathbf{Q} = \mathbf{Q}_1 \mathbf{Q}_1'$. To bound the operator norm of $\mathbf{Q}$, we have

$$
\begin{aligned}
\|\mathbf{Q}\|_{\mathrm{op}} &\leq \|\mathbf{Q}_1\|_{\mathrm{op}}^2 = \sup_{\mathbf{u} \in \mathbb{S}^{n-1}} \mathbf{u}' \mathbf{Q}_1' \mathbf{Q}_1 \mathbf{u} = \frac{1}{n} \sup_{\mathbf{u} \in \mathbb{S}^{n-1}} \mathbf{u}' [\mathbf{I}_n \otimes \mathbf{w}_G'] \mathbf{\Sigma} [\mathbf{I}_n \otimes \mathbf{w}_G] \mathbf{u} \\
&\leq \frac{1}{n} \left( \sup_{\mathbf{u} \in \mathbb{S}^{n-1}} \frac{\mathbf{u}' [\mathbf{I}_n \otimes \mathbf{w}_G'] \mathbf{\Sigma} [\mathbf{I}_n \otimes \mathbf{w}_G] \mathbf{u}}{\mathbf{u}' [\mathbf{I}_n \otimes \mathbf{w}_G'] [\mathbf{I}_n \otimes \mathbf{w}_G] \mathbf{u}} \right) \left( \sup_{\mathbf{u} \in \mathbb{S}^{n-1}} \mathbf{u}' [\mathbf{I}_n \otimes \mathbf{w}_G' \mathbf{w}_G] \mathbf{u} \right) \\
&\leq \frac{C_x}{n} \|\mathbf{w}_G\|_2^2 \leq \frac{C_x}{n} \|\mathbf{U}_G\|_{\mathrm{op}}^2 \|\mathbf{\Delta}\|_2^2 \leq \frac{\kappa_U}{n}.
\end{aligned}
\tag{20}
$$

Finally we can use (18) and (20) to bound the Frobenius norm of $\boldsymbol{Q}$. By some algebra, for any square matrices $\boldsymbol{A}, \boldsymbol{B} \in \mathbb{R}^{n \times n}$, $\|\boldsymbol{A}\boldsymbol{B}\|_{\mathrm{F}}^2 \leq \|\boldsymbol{A}\|_{\mathrm{op}}^2 \|\boldsymbol{B}\|_{\mathrm{F}}^2$ holds. Hence,

$$\|\boldsymbol{Q}\|_{\mathrm{F}}^2 = \|\boldsymbol{Q}_1 \boldsymbol{Q}_1'\|_{\mathrm{F}}^2 \leq \|\boldsymbol{Q}_1\|_{\mathrm{op}}^2 \|\boldsymbol{Q}_1\|_{\mathrm{F}}^2 = \|\boldsymbol{Q}_1\|_{\mathrm{op}}^2 \operatorname{tr}(\boldsymbol{Q}) \leq \frac{\kappa_U^2}{n}.$$

This, together with (17), (20) and the Hanson-Wright inequality (Vershynin, 2018, Chapter 6), leads to

$$\mathbb{P}\left\{\left|\|\boldsymbol{\Delta}\|_n^2 - \mathbb{E}\|\boldsymbol{\Delta}\|_n^2\right| \geq t\right\} \leq 2\exp\left\{-c_{\mathrm{H}} \min\left(\frac{t}{\|\boldsymbol{Q}\|_{\mathrm{op}}}, \frac{t^2}{\|\boldsymbol{Q}\|_{\mathrm{F}}^2}\right)\right\} \tag{21}$$
$$\leq 2\exp\left\{-c_{\mathrm{H}} n \min\left(t/\kappa_U, (t/\kappa_U)^2\right)\right\},$$

where $c_{\mathrm{H}}$ is a positive constant.

On the other hand,

$$\|\boldsymbol{\Delta}\|_n^2 - \mathbb{E}\|\boldsymbol{\Delta}\|_n^2 = \frac{1}{n}\sum_{i=1}^n \boldsymbol{\Delta}' \boldsymbol{z}^i \boldsymbol{z}^{i\prime} \boldsymbol{\Delta} - \mathbb{E}(\boldsymbol{\Delta}' \boldsymbol{z}^i \boldsymbol{z}^{i\prime} \boldsymbol{\Delta}) = \boldsymbol{\Delta}' \widehat{\boldsymbol{\Gamma}} \boldsymbol{\Delta},$$

where $\widehat{\boldsymbol{\Gamma}} = n^{-1}\sum_{i=1}^n [\boldsymbol{z}^i \boldsymbol{z}^{i\prime} - E(\boldsymbol{z}^i \boldsymbol{z}^{i\prime})]$ is a symmetric matrix. Consider a $\varepsilon$-net $\bar{\mathcal{S}}_{2K}$, with the cardinality of $\mathcal{N}(2K, \varepsilon)$, for the set $\mathcal{S}_{2K}$. For any $\boldsymbol{\Delta} \in \mathcal{S}_{2K}$, there exists a $\bar{\boldsymbol{\Delta}}_j \in \bar{\mathcal{S}}_{2K}$ such that $\|\boldsymbol{\Delta} - \bar{\boldsymbol{\Delta}}_j\|_{\mathrm{F}} \leq \varepsilon$. Note that $\boldsymbol{\Delta} - \bar{\boldsymbol{\Delta}}_j \in \widehat{\mathcal{S}}_{4K}$ and, from Lemma 1, we can further find $\boldsymbol{\Delta}_1, \boldsymbol{\Delta}_2 \in \widehat{\mathcal{S}}_{2K}$ such that $\langle \boldsymbol{\Delta}_1, \boldsymbol{\Delta}_2 \rangle = 0$ and $\boldsymbol{\Delta} - \bar{\boldsymbol{\Delta}}_j = \boldsymbol{\Delta}_1 + \boldsymbol{\Delta}_2$, and it then holds that $\|\boldsymbol{\Delta}_1\|_{\mathrm{F}} + \|\boldsymbol{\Delta}_2\|_{\mathrm{F}} \leq \sqrt{2}\|\boldsymbol{\Delta} - \bar{\boldsymbol{\Delta}}_j\|_{\mathrm{F}} \leq \sqrt{2}\varepsilon$. Moreover, for a general real symmetric matrix $\boldsymbol{A} \in \mathbb{R}^{d \times d}$, $\boldsymbol{u} \in \mathbb{R}^d$ and $\boldsymbol{v} \in \mathbb{R}^d$, $\sup_{\|\boldsymbol{u}\|_2 = \|\boldsymbol{v}\|_2 = 1} \boldsymbol{u}' \boldsymbol{A} \boldsymbol{v} = \sup_{\|\boldsymbol{u}\|_2 = 1} \boldsymbol{u}' \boldsymbol{A} \boldsymbol{u}$. As a result,

$$\boldsymbol{\Delta}' \widehat{\boldsymbol{\Gamma}} \boldsymbol{\Delta} = \bar{\boldsymbol{\Delta}}_j' \widehat{\boldsymbol{\Gamma}} \bar{\boldsymbol{\Delta}}_j + (\boldsymbol{\Delta}_1 + \boldsymbol{\Delta}_2)' \widehat{\boldsymbol{\Gamma}} (\boldsymbol{\Delta}_1 + \boldsymbol{\Delta}_2 + 2\bar{\boldsymbol{\Delta}}_j)$$
$$\leq \max_{1 \leq j \leq \mathcal{N}(2K, \varepsilon)} \bar{\boldsymbol{\Delta}}_j' \widehat{\boldsymbol{\Gamma}} \bar{\boldsymbol{\Delta}}_j + 5\varepsilon \sup_{\boldsymbol{\Delta} \in \mathcal{S}_{2K}} \boldsymbol{\Delta}' \widehat{\boldsymbol{\Gamma}} \boldsymbol{\Delta},$$

where $(\|\boldsymbol{\Delta}_1\|_{\mathrm{F}} + \|\boldsymbol{\Delta}_2\|_{\mathrm{F}})^2 + 2(\|\boldsymbol{\Delta}_1\|_{\mathrm{F}} + \|\boldsymbol{\Delta}_2\|_{\mathrm{F}})\|\bar{\boldsymbol{\Delta}}_j\|_{\mathrm{F}} \leq 2(\varepsilon + \sqrt{2})\varepsilon \leq 5\varepsilon$ as $\varepsilon \leq 1$, and this leads to

$$\sup_{\boldsymbol{\Delta} \in \mathcal{S}_{2K}} \boldsymbol{\Delta}' \widehat{\boldsymbol{\Gamma}} \boldsymbol{\Delta} \leq (1 - 5\varepsilon)^{-1} \max_{1 \leq j \leq \mathcal{N}(2K, \varepsilon)} \bar{\boldsymbol{\Delta}}_j' \widehat{\boldsymbol{\Gamma}} \bar{\boldsymbol{\Delta}}_j. \tag{22}$$

Note that, from Lemma 1(b), $\log \mathcal{N}(2K, \varepsilon) \leq 2d_{\mathcal{M}} \log(9/\varepsilon)$, where $d_{\mathcal{M}} = K(L + P + 1)$. Let $\varepsilon = 1/10$, and then $2\log(9/\varepsilon) < 9$. Combining (21) and (22) and letting $t = \kappa_L/2$, we have

$$\mathbb{P}\left\{\sup_{\boldsymbol{\Delta} \in \mathcal{S}_{2K}} \|\boldsymbol{\Delta}\|_n^2 - \mathbb{E}\|\boldsymbol{\Delta}\|_n^2 \geq \frac{\kappa_L}{2}\right\} \leq 2\exp\left\{-c_{\mathrm{H}} n \left(\frac{\kappa_L}{4\kappa_U}\right)^2 + 9d_{\mathcal{M}}\right\},$$

which, together with (19) and the fact that $\kappa_L \leq \kappa_U$, implies that

$$\mathbb{P}\left\{\sup_{\boldsymbol{\Delta} \in \mathcal{S}_{2K}} \|\boldsymbol{\Delta}\|_n^2 \geq \alpha_{\mathrm{RSM}}\right\} \leq 2\exp\left\{-c_{\mathrm{H}} n \left(\frac{\kappa_L}{4\kappa_U}\right)^2 + 9d_{\mathcal{M}}\right\}, \tag{23}$$

where $\alpha_{\mathrm{RSM}} = 1.5\kappa_U \geq \kappa_U + \kappa_L/2$.

By a method similar to (22), we can also show that

$$\sup_{\boldsymbol{\Delta} \in \mathcal{S}_{2K}} \mathbb{E}\|\boldsymbol{\Delta}\|_n^2 - \|\boldsymbol{\Delta}\|_n^2 \leq (1 - 5\varepsilon)^{-1} \max_{1 \leq j \leq \mathcal{N}(2K, \varepsilon)} \mathbb{E}\|\bar{\boldsymbol{\Delta}}_j\|_n^2 - \|\bar{\boldsymbol{\Delta}}_j\|_n^2,$$

which, together with (21), leads to

$$\mathbb{P}\left\{\sup_{\boldsymbol{\Delta} \in \mathcal{S}_{2K}} \mathbb{E}\|\boldsymbol{\Delta}\|_n^2 - \|\boldsymbol{\Delta}\|_n^2 \geq \frac{\kappa_L}{2}\right\} \leq 2\exp\left\{-c_{\mathrm{H}} n \left(\frac{\kappa_L}{4\kappa_U}\right)^2 + 9d_{\mathcal{M}}\right\},$$

where $t = \kappa_L/2$ and $\varepsilon = 1/10$. As a result,

$$\mathbb{P}\left\{\inf_{\boldsymbol{\Delta} \in \mathcal{S}_{2K}} \|\boldsymbol{\Delta}\|_n^2 \leq \alpha_{\mathrm{RSC}}\right\} \leq 2\exp\left\{-c_{\mathrm{H}} n \left(\frac{\kappa_L}{4\kappa_U}\right)^2 + 9d_{\mathcal{M}}\right\},$$

where $\alpha_{\mathrm{RSC}} = \kappa_L - \kappa_L/2$. This, together with (23), accomplishes the proof. $\qquad\square$

**Lemma 3.** *(Concentration bound for martingale) Let $\{\xi^i, 1 \leq i \leq n\}$ be independent $\sigma^2$-sub-Gaussian random variables with mean zero, and $\{z^i, 1 \leq i \leq n\}$ is another sequence of random variables. Suppose that $\xi^i$ is independent of $\{z^i, z^{i-1}, \ldots, z^1\}$ for all $1 \leq i \leq n$. It then holds that, for any real numbers $\alpha, \beta > 0$,*

$$\mathbb{P}\left[\left\{\frac{1}{n}\sum_{i=1}^{n}\xi^i z^i \geq \alpha\right\} \cap \left\{\frac{1}{n}\sum_{i=1}^{n}(z^i)^2 \leq \beta\right\}\right] \leq \exp\left(-\frac{n\alpha^2}{2\sigma^2\beta}\right).$$

*Proof.* We can prove the lemma by a method similar to Lemma 4.2 in Simchowitz et al. (2018). $\square$

### A.4 CLASSIFICATION PROBLEMS

Starting from Section 2.2, we know that, for each input tensor $\mathcal{X}$, the intermediate scalar output after convolution and pooling has the following form

$$\text{output} = \langle \mathcal{X}, \sum_{k=1}^{K}(\mathcal{B}_k \otimes \mathcal{A}_k) \times_1 U_{\mathcal{F}}^{(1)} \times_2 U_{\mathcal{F}}^{(2)} \times_3 \cdots \times_N U_{\mathcal{F}}^{(N)}\rangle = \langle \mathcal{Z}, \mathcal{W}\rangle,$$

where $\mathcal{W} = \sum_{k=1}^{K}\mathcal{B}_k \otimes \mathcal{A}_k$ and $\mathcal{Z} = \mathcal{X} \times_1 U_{\mathcal{F}}^{(1)\prime} \times_2 U_{\mathcal{F}}^{(2)\prime} \times_3 \cdots \times_N U_{\mathcal{F}}^{(N)\prime}$, $\mathcal{A}_k \in \mathbb{R}^{l_1 \times l_2 \times \cdots \times l_N}$ is the $k$th kernel and $\mathcal{B}_k \in \mathbb{R}^{p_1 \times p_2 \times \cdots \times p_N}$ is the corresponding $k$th fully-connected weight tensor.

Consider a binary classification problem. We have the binary label output $y \in \{0, 1\}$ with

$$p(y|\mathcal{Z}) = \left(\frac{1}{1 + \exp(\langle\mathcal{Z}, \mathcal{W}\rangle)}\right)^{1-y}\left(\frac{\exp(\langle\mathcal{Z}, \mathcal{W}\rangle)}{1 + \exp(\langle\mathcal{Z}, \mathcal{W}\rangle)}\right)^{y} = \exp\left\{y\langle\mathcal{Z}, \mathcal{W}\rangle - \log\left[1 + \exp(\langle\mathcal{Z}, \mathcal{W}\rangle)\right]\right\}.$$

Suppose we have samples $\{\mathcal{Z}^i, y^i\}_{i=1}^n$, we use the negative log-likelihood function as our loss function. It is given, up to a scaling of $n^{-1}$ by

$$\mathcal{L}_n(\mathcal{W}) = -\frac{1}{n}\sum_{i=1}^{n}y^i\langle\mathcal{Z}^i, \mathcal{W}\rangle + \frac{1}{n}\sum_{i=1}^{n}\phi(\langle\mathcal{Z}^i, \mathcal{W}\rangle), \tag{24}$$

where $\phi(z) = \log(1 + e^z)$ and its gradient and Hessian matrix is

$$\nabla\mathcal{L}_n(\mathcal{W}) = \frac{\partial\mathcal{L}_n(\mathcal{W})}{\partial\,\text{vec}(\mathcal{W})} = -\frac{1}{n}\sum_{i=1}^{n}y^i\,\text{vec}(\mathcal{Z}^i) + \frac{1}{n}\sum_{i=1}^{n}\phi'(\langle\mathcal{Z}^i, \mathcal{W}\rangle)\,\text{vec}(\mathcal{Z}^i) \text{ and,} \tag{25}$$

$$H_n(\mathcal{W}) = \frac{\partial^2\mathcal{L}_n(\mathcal{W})}{\partial^2\,\text{vec}(\mathcal{W})} = \frac{1}{n}\sum_{i=1}^{n}\phi''(\langle\mathcal{Z}^i, \mathcal{W}\rangle)\,\text{vec}(\mathcal{Z}^i)\,\text{vec}(\mathcal{Z}^i)', \tag{26}$$

with $\phi'(z) = 1/(1 + e^{-z}) \in (0, 1)$ and $\phi''(z) = e^z/(1 + e^z)^2 = 1/(e^{-z} + 2 + e^z) \in (0, 0.25)$ [because $e^{-z} + e^z \geq 2$]. Since $H_n(\mathcal{W})$ is a positive semi-definite matrix, our loss function in (24) is convex.

Suppose $\widehat{\mathcal{W}}$ is the minimizer to the loss function

$$\widehat{\mathcal{W}} \in \operatorname*{arg\,min}_{\mathcal{W} \in \widehat{\mathcal{S}}_K \cap \mathbb{B}(R)} \mathcal{L}_n(\mathcal{W}),$$

where $\widehat{\mathcal{S}}_K = \{\sum_{k=1}^{K}\mathcal{B}_k \otimes \mathcal{A}_k : \mathcal{A}_k \in \mathbb{R}^{l_1 \times l_2 \times \cdots \times l_N} \text{ and } \mathcal{B}_k \in \mathbb{R}^{p_1 \times p_2 \times \cdots \times p_N}\}$, and $\mathbb{B}(R)$ is a Frobenius ball of some fixed radius $R$ centered at the underlying true parameter.

Similar to Fan et al. (2019), we need to make two additional assumptions to guarantee the locally strong convexity condition.

**Assumption 4** (Classification). *Apart from Assumptions 2-3, we additionally assume that*

*(C1) $\{\text{vec}(\mathcal{X}^i)\}_{i=1}^n$ are i.i.d gaussian vectors with mean zero and variance $\Sigma$, where $\Sigma \leq C_x I$.*

*(C2) the Hessian matrix at the underlying true parameter $\mathcal{W}^*$ is positive definite and there exists some $\kappa_0 > \kappa_U > 0$, such that $H(\mathcal{W}^*) = \mathbb{E}(H_n(\mathcal{W}^*)) \geq \kappa_0 \cdot I$, where $\kappa_U = C_x C_u$;*

*(C3) $\|\mathcal{W}^*\|_F \geq \alpha\sqrt{d_{\mathcal{M}}}$ for some constant $\alpha$, where $d_{\mathcal{M}} = K(P + L + 1)$.*

Notice that we can relax (C1) into Assumption 1, but it will require more techincal details. Also, $\{\text{vec}(\boldsymbol{\mathcal{X}}^i)\}_{i=1}^n$ can be sub-gaussian random vectors instead of gaussian random vectors.

Denote $d_{\mathcal{M}} = K(P + L + 1)$.

**Theorem 3** (Classification: CNN). *Suppose that Assumptions 2&3 and Assumption 4 hold and $n \gtrsim d_{\mathcal{M}}$. Then, for some $\delta > 0$,*

$$\|\widehat{\boldsymbol{\mathcal{W}}} - \boldsymbol{\mathcal{W}}^*\|_{\mathrm{F}} \leq \frac{2\sqrt{\kappa_U}}{\kappa_1}\left(\sqrt{\frac{d_{\mathcal{M}}}{n}} + \sqrt{\frac{\delta}{n}}\right),$$

*with probability $1 - 4\exp\{-0.25cn + 9d_{\mathcal{M}}\} - 2\exp\{-c_{\gamma}d_{\mathcal{M}} - c\delta\}$, where $\delta = O_p(1)$, $c$ and $c_{\gamma}$ are some positive constants.*

Denote $d_{\mathcal{M}}^{\mathrm{TU}} = \prod_{j=1}^{N+1} R_j + \sum_{i=1}^N l_i R_i + R_{N+1}P$ and $d_{\mathcal{M}}^{\mathrm{CP}} = R^{N+1} + R(\sum_{i=1}^N l_i + P)$.

**Corollary 2** (Classification: Compressed CNN). *Let $(\widehat{\boldsymbol{\mathcal{W}}}, d_{\mathcal{M}})$ be $(\widehat{\boldsymbol{\mathcal{W}}}_{\mathrm{TU}}, d_{\mathcal{M}}^{\mathrm{TU}})$ for Tucker decomposition, or $(\widehat{\boldsymbol{\mathcal{W}}}_{\mathrm{CP}}, d_{\mathcal{M}}^{\mathrm{CP}})$ for CP decomposition. Suppose that Assumptions in Theorem 3 hold and $n \gtrsim c_N d_{\mathcal{M}}$. Then, for some $\delta > 0$,*

$$\|\widehat{\boldsymbol{\mathcal{W}}} - \boldsymbol{\mathcal{W}}^*\|_{\mathrm{F}} \leq \frac{2\sqrt{\kappa_U}}{\kappa_1}\left(\sqrt{\frac{3c_N d_{\mathcal{M}}}{n}} + \sqrt{\frac{\delta}{n}}\right),$$

*with probability $1 - 4\exp\{-0.25cn + 3c_N d_{\mathcal{M}}\} - 2\exp\{-c_{\gamma}d_{\mathcal{M}} - c\delta\}$, where $\delta = O_p(1)$, $c$ and $c_{\gamma}$ are some positive constants, and $c_N$ is defined as in Theorem 2.*

We consider a binary classification problem as a simple illustration. In fact, the analysis framework can be easily extended to a multiclass classification problem. Here, we consider a $M$-class classification problem.

Because we need our intermediate output after convolution and pooling to be a vector of length $M$, instead of a scalar, we need to introduce one additional dimension to the fully-connected weight tensor. Hence, we introduce another subscript $m$ to represent the class label. And the set of fully-connected weights is represented as $\{\boldsymbol{\mathcal{B}}_{k,m}\}_{1 \leq k \leq K, 1 \leq m \leq M}$ where each $\boldsymbol{\mathcal{B}}_{k,m}$ is of size $p_1 \times p_2 \times \cdots \times p_N$.

Then, for each input tensor $\boldsymbol{\mathcal{X}}$, our intermediate output is a vector of size $M$, where the $m$th entry is represented by

$$\text{output}_m = \langle \boldsymbol{\mathcal{Z}}, \boldsymbol{\mathcal{W}}_m \rangle,$$

where $\boldsymbol{\mathcal{W}}_m = \sum_{k=1}^K (\boldsymbol{\mathcal{B}}_{k,m} \otimes \boldsymbol{\mathcal{A}}_k)$ is an $N$-th order tensor of size $l_1 p_1 \times l_2 p_2 \times \cdots \times l_N p_N$.

For a $M$-class classification problem, we have the vector label output $\boldsymbol{y} \in (0,1)^M$. Essentially, each entry of $\boldsymbol{y}$ comes from a different binary classification problem, with $M$ problems in total. We can model it as

$$\begin{aligned} p(y_m|\boldsymbol{\mathcal{Z}}) &= \left(\frac{1}{1 + \exp(\langle \boldsymbol{\mathcal{Z}}, \boldsymbol{\mathcal{W}}_m \rangle)}\right)^{1-y_m}\left(\frac{\exp(\langle \boldsymbol{\mathcal{Z}}, \boldsymbol{\mathcal{W}}_m \rangle)}{1 + \exp(\langle \boldsymbol{\mathcal{Z}}, \boldsymbol{\mathcal{W}}_m \rangle)}\right)^{y_m} \\ &= \exp\{y_m\langle \boldsymbol{\mathcal{Z}}, \boldsymbol{\mathcal{W}}_m \rangle - \log[1 + \exp(\langle \boldsymbol{\mathcal{Z}}, \boldsymbol{\mathcal{W}}_m \rangle)]\} \end{aligned}$$

If we stack $\{\boldsymbol{\mathcal{W}}_m\}_{m=1}^M$ into a tensor $\boldsymbol{\mathcal{W}}_{\mathrm{stack}}$, which is an $N+1$-order tensor of size $l_1 p_1 \times l_2 p_2 \times \cdots \times l_N p_N \times M$. And we further introduce some natural basis vectors $\{\boldsymbol{e}_m \in \mathbb{R}^M\}_{m=1}^M$. It can be shown that

$$\langle \boldsymbol{\mathcal{Z}}^i, \boldsymbol{\mathcal{W}}_m \rangle = \langle \underbrace{\boldsymbol{\mathcal{Z}}^i \circ \boldsymbol{e}_m}_{\boldsymbol{\mathcal{Z}}_m^i}, \boldsymbol{\mathcal{W}}_{\mathrm{stack}} \rangle,$$

where $\circ$ is the outer product.

We can then recast this model into one with $nM$ samples $\{\boldsymbol{\mathcal{Z}}_m^i, y_m^i : 1 \leq k \leq K, 1 \leq m \leq M\}$. The corresponding loss function is

$$\mathcal{L}_n(\boldsymbol{\mathcal{W}}_{\mathrm{stack}}) = -\frac{1}{nM}\sum_{m=1}^M \sum_{i=1}^n y^i \langle \boldsymbol{\mathcal{Z}}_m^i, \boldsymbol{\mathcal{W}}_{\mathrm{stack}} \rangle + \frac{1}{nM}\sum_{m=1}^M \sum_{i=1}^n \phi(\langle \boldsymbol{\mathcal{Z}}_m^i, \boldsymbol{\mathcal{W}}_{\mathrm{stack}} \rangle).$$

We can now use the techniques in Theorem 3 to show the following corollaries for multiclass classification problem.

Denote $d_{\mathcal{M}}^{\mathrm{MC}} = K(MP + L + 1)$.

**Corollary 3** (Multiclass Classification: CNN). *Under similar assumptions as in Theorem 3, suppose that $n \gtrsim d_{\mathcal{M}}^{\mathrm{MC}}$. Then, for some $\delta > 0$,*

$$\|\widehat{\mathcal{W}}_{\mathrm{stack}} - \mathcal{W}_{\mathrm{stack}}^*\|_{\mathrm{F}} \le \frac{2\sqrt{\kappa_U}}{\kappa_1} \left( \sqrt{\frac{d_{\mathcal{M}}^{\mathrm{MC}}}{n}} + \sqrt{\frac{\delta}{n}} \right),$$

*with probability $1 - 4\exp\left\{-0.25cn + 9d_{\mathcal{M}}^{\mathrm{MC}}\right\} - 2\exp\left\{-c_\gamma d_{\mathcal{M}}^{\mathrm{MC}} - c\delta\right\}$, where $\delta = O_p(1)$, $c$ and $c_\gamma$ are some positive constants.*

Denote $d_{\mathcal{M}}^{\mathrm{MC-TU}} = \prod_{j=1}^{N+1} R_j + \sum_{i=1}^{N} l_i R_i + R_{N+1} PM$ and $d_{\mathcal{M}}^{\mathrm{MC-CP}} = R^{N+1} + R(\sum_{i=1}^{N} l_i + PM)$.

**Corollary 4** (Multiclass Classification: Compressed CNN). *Let $(\widehat{\mathcal{W}}_{\mathbf{stack}}, d_{\mathcal{M}}^{\mathrm{MC}})$ be $(\widehat{\mathcal{W}}_{\mathrm{stack,TU}}, d_{\mathcal{M}}^{\mathrm{MC-TU}})$ for Tucker decomposition, or $(\widehat{\mathcal{W}}_{\mathrm{stack,CP}}, d_{\mathcal{M}}^{\mathrm{MC-CP}})$ for CP decomposition. Suppose that Assumptions in Theorem 3 hold and $n \gtrsim c_N d_{\mathcal{M}}^{\mathrm{multi}}$. Then, for some $\delta > 0$,*

$$\|\widehat{\mathcal{W}}_{\mathrm{stack}} - \mathcal{W}_{\mathrm{stack}}^*\|_{\mathrm{F}} \le \frac{2\sqrt{\kappa_U}}{\kappa_1} \left( \sqrt{\frac{3c_N d_{\mathcal{M}}^{\mathrm{MC}}}{n}} + \sqrt{\frac{\delta}{n}} \right),$$

*with probability $1 - 4\exp\left\{-0.25cn + 3c_N d_{\mathcal{M}}^{\mathrm{MC}}\right\} - 2\exp\left\{-c_\gamma d_{\mathcal{M}}^{\mathrm{MC}} - c\delta\right\}$, where $\delta = O_p(1)$, $c$ and $c_\gamma$ are some positive constants, and $c_N$ is defined as in Theorem 2.*

*Proof of Theorem 3.* Denote the sets $\widehat{\mathcal{S}}_K = \{\sum_{k=1}^{K} \mathcal{B}_k \otimes \mathcal{A}_k : \mathcal{A}_k \in \mathbb{R}^{l_1 \times l_2 \times \cdots \times l_N}$ and $\mathcal{B}_k \in \mathbb{R}^{p_1 \times p_2 \times \cdots \times p_N}\}$ and $\mathcal{S}_K = \{\mathcal{W} \in \widehat{\mathcal{S}}_K : \|\mathcal{W}\|_{\mathrm{F}} = 1\}$. We further denote $\mathbf{\Delta} = \mathrm{vec}(\mathcal{W} - \mathcal{W}^*)$, where $\mathcal{W}^*$ is the underlying true parameter and $\mathcal{W}, \mathcal{W}^* \in \widehat{\mathcal{S}}_K$, and define the first-order Taylor error

$$\mathcal{E}_n(\mathbf{\Delta}) = \mathcal{L}_n(\mathcal{W}) - \mathcal{L}_n(\mathcal{W}^*) - \langle \nabla \mathcal{L}_n(\mathcal{W}^*), \mathbf{\Delta} \rangle.$$

Suppose $\widehat{\mathcal{W}}$ is the minimizer for the loss function, i.e.,

$$\widehat{\mathcal{W}} = \arg\min_{\mathcal{W} \in \widehat{\mathcal{S}}_K} \mathcal{L}_n(\mathcal{W}).$$

Denote $\widehat{\mathbf{\Delta}} = \mathrm{vec}(\widehat{\mathcal{W}} - \mathcal{W}^*)$. We then have

$$\mathcal{L}_n(\widehat{\mathcal{W}}) - \mathcal{L}_n(\mathcal{W}^*) \le 0, \text{ which can be rearranged into } \mathcal{E}_n(\widehat{\mathbf{\Delta}}) \le - \left\langle \nabla \mathcal{L}_n(\mathcal{W}^*), \widehat{\mathbf{\Delta}} \right\rangle.$$

Then, for some $\widetilde{\mathcal{W}}$ between $\widehat{\mathcal{W}}$ and $\mathcal{W}^*$,

$$\frac{1}{2}\widehat{\mathbf{\Delta}}' \mathbf{H}_n(\widetilde{\mathcal{W}})\widehat{\mathbf{\Delta}} \le \left| \left\langle \nabla \mathcal{L}_n(\mathcal{W}^*), \widehat{\mathbf{\Delta}} \right\rangle \right|,$$

which leads to

$$\|\widehat{\mathbf{\Delta}}\|_{\mathrm{F}}^2 \sup_{\mathbf{\Delta} \in \mathcal{S}_{2K}} \mathbf{\Delta}' \mathbf{H}_n(\widetilde{\mathcal{W}})\mathbf{\Delta} \le 2\|\widehat{\mathbf{\Delta}}\|_{\mathrm{F}} \sup_{\mathbf{\Delta} \in \mathcal{S}_{2K}} |\langle \nabla \mathcal{L}_n(\mathcal{W}^*), \mathbf{\Delta} \rangle|. \tag{27}$$

From Lemma 4 and Lemma 5, when $n \gtrsim d_{\mathcal{M}}$, we obtain that, for some $\delta > 0$,

$$\|\widehat{\mathcal{W}} - \mathcal{W}^*\|_{\mathrm{F}} \le \frac{2\sqrt{\kappa_U}}{\kappa_1} \left( \sqrt{\frac{d_{\mathcal{M}}}{n}} + \sqrt{\frac{\delta}{n}} \right),$$

with probability $1 - 4\exp\left\{-0.25cn + 9d_{\mathcal{M}}\right\} - 2\exp\left\{-c_\gamma d_{\mathcal{M}} - c\delta\right\}$.

$\square$

Now we proof several lemmas to be used in Theorem 3. For simplicity of notation, denote $\boldsymbol{x}^i = \mathrm{vec}(\boldsymbol{\mathcal{X}}^i)$ and $\boldsymbol{z}^i = \mathrm{vec}(\boldsymbol{\mathcal{Z}}^i)$. It holds that $\boldsymbol{z}^i = \boldsymbol{U}_G \boldsymbol{x}^i$, for $1 \leq i \leq n$.

**Lemma 4** (Deviation bound). *Under Assumption 4(C3), suppose that $n \gtrsim d_{\mathcal{M}}$, then, for some $\delta > 0$,*

$$\mathbb{P}\left\{\sup_{\boldsymbol{\Delta} \in \mathcal{S}_{2K}} |\langle \nabla \mathcal{L}_n(\boldsymbol{\mathcal{W}}^*), \boldsymbol{\Delta}\rangle| \geq 0.5\sqrt{\kappa_U}\left(\sqrt{\frac{d_{\mathcal{M}}}{n}} + \sqrt{\frac{\delta}{n}}\right)\right\} \leq 2\exp\left\{-c_\gamma d_{\mathcal{M}}\right\},$$

*where $\delta = O_p(1)$, $d_{\mathcal{M}} = K(P + L + 1)$, $\kappa_U = C_x C_u$ and $c_\gamma$ is some positive constant.*

*Proof.* Let $\eta^i = \langle \boldsymbol{\mathcal{Z}}^i, \boldsymbol{\mathcal{W}}^* \rangle$, and from (25),

$$\langle \nabla \mathcal{L}_n(\boldsymbol{\mathcal{W}}^*), \boldsymbol{\Delta}\rangle = \frac{1}{n}\sum_{i=1}^n [\phi'(\eta^i) - y^i]\langle \boldsymbol{z}^i, \boldsymbol{\Delta}\rangle,$$

and we can observe that

$$\mathbb{E}\{[\phi'(\eta^i) - y^i]\langle \boldsymbol{z}^i, \boldsymbol{\Delta}\rangle\} = \mathbb{E}\{\mathbb{E}[\phi'(\eta^i) - y|z]\langle \boldsymbol{z}^i, \boldsymbol{\Delta}\rangle\} = 0 = \mathbb{E}[\phi'(\eta^i) - y^i]\mathbb{E}[\langle \boldsymbol{z}^i, \boldsymbol{\Delta}\rangle].$$

It implies (i) $\mathbb{E}\langle \nabla \mathcal{L}_n(\boldsymbol{\mathcal{W}}^*), \boldsymbol{\Delta}\rangle = 0$, (ii) the independence between $\phi'(\eta^i) - y^i$ and $\langle \boldsymbol{z}^i, \boldsymbol{\Delta}\rangle$. And from Lemma 6, the independence leads to $\|[\phi'(\eta^i) - y^i]\langle \boldsymbol{z}^i, \boldsymbol{\Delta}\rangle\|_{\psi_1} \leq \|\phi'(\eta^i) - y^i\|_{\psi_2}\|\langle \boldsymbol{z}^i, \boldsymbol{\Delta}\rangle\|_{\psi_2}$. Denote $\kappa_U = C_x C_u$. For any fixed $\boldsymbol{\Delta}$ such that $\|\boldsymbol{\Delta}\|_2 = 1$,

$$\|\langle \boldsymbol{z}^i, \boldsymbol{\Delta}\rangle\|_{\psi_2} = \|\langle \boldsymbol{w}^i, \boldsymbol{\Sigma}^{1/2}\boldsymbol{U}_G\boldsymbol{\Delta}\rangle\|_{\psi_2} \leq \sqrt{\kappa_U}.$$

This, together with $\|\phi'(\eta^i) - y^i\|_{\psi_2} \leq 0.25$ in Lemma 7, gives us

$$\|[\phi'(\eta^i) - y^i]\langle \boldsymbol{z}^i, \boldsymbol{\Delta}\rangle\|_{\psi_1} \leq 0.25\sqrt{\kappa_U}.$$

Then, we can use the Beinstein-type inequality, namely Corollary 5.17 in (Vershynin, 2010) to derive that, for any fixed $\boldsymbol{\Delta}$ with unit $l_2$-norm,

$$\mathbb{P}\{|\langle \nabla \mathcal{L}_n(\boldsymbol{\mathcal{W}}^*), \boldsymbol{\Delta}\rangle| \geq t\} = \mathbb{P}\left\{\frac{1}{n}\left|\sum_{i=1}^n [\phi'(\eta^i) - y^i]\langle \boldsymbol{z}^i, \boldsymbol{\Delta}\rangle\right| \geq t\right\}$$
$$\leq 2\exp\left\{-cn\min\left(\frac{4t}{\sqrt{\kappa_U}}, \frac{16t^2}{\kappa_U}\right)\right\}. \tag{28}$$

Consider a $\varepsilon$-net $\bar{\mathcal{S}}_{2K}$, with the cardinality of $\mathcal{N}(2K, \varepsilon)$, for the set $\mathcal{S}_{2K}$. For any $\boldsymbol{\Delta} \in \mathcal{S}_{2K}$, there exists a $\bar{\boldsymbol{\Delta}}_j \in \bar{\mathcal{S}}_{2K}$ such that $\|\boldsymbol{\Delta} - \bar{\boldsymbol{\Delta}}_j\|_F \leq \varepsilon$. Note that $\boldsymbol{\Delta} - \bar{\boldsymbol{\Delta}}_j \in \widehat{\mathcal{S}}_{4K}$ and, from Lemma 1(a), we can further find $\boldsymbol{\Delta}_1, \boldsymbol{\Delta}_2 \in \widehat{\mathcal{S}}_{2K}$ such that $\langle \boldsymbol{\Delta}_1, \boldsymbol{\Delta}_2\rangle = 0$ and $\boldsymbol{\Delta} - \bar{\boldsymbol{\Delta}}_j = \boldsymbol{\Delta}_1 + \boldsymbol{\Delta}_2$. It then holds that $\|\boldsymbol{\Delta}_1\|_F + \|\boldsymbol{\Delta}_2\|_F \leq \sqrt{2}\|\boldsymbol{\Delta} - \bar{\boldsymbol{\Delta}}_j\|_F \leq \sqrt{2}\varepsilon$ since $\|\boldsymbol{\Delta} - \bar{\boldsymbol{\Delta}}_j\|_F^2 = \|\boldsymbol{\Delta}_1\|_F^2 + \|\boldsymbol{\Delta}_2\|_F^2$. As a result,

$$|\langle \nabla \mathcal{L}_n(\boldsymbol{\mathcal{W}}^*), \boldsymbol{\Delta}\rangle| = |\langle \nabla \mathcal{L}_n(\boldsymbol{\mathcal{W}}^*), \bar{\boldsymbol{\Delta}}_j\rangle| + |\langle \nabla \mathcal{L}_n(\boldsymbol{\mathcal{W}}^*), \boldsymbol{\Delta}_1\rangle| + |\langle \nabla \mathcal{L}_n(\boldsymbol{\mathcal{W}}^*), \boldsymbol{\Delta}_2\rangle|$$
$$\leq \max_{1 \leq j \leq \mathcal{N}(2K, \varepsilon)} |\langle \nabla \mathcal{L}_n(\boldsymbol{\mathcal{W}}^*), \bar{\boldsymbol{\Delta}}_j\rangle| + \sqrt{2}\varepsilon \sup_{\boldsymbol{\Delta} \in \mathcal{S}_{2K}} |\langle \nabla \mathcal{L}_n(\boldsymbol{\mathcal{W}}^*), \boldsymbol{\Delta}\rangle|,$$

which leads to

$$\sup_{\boldsymbol{\Delta} \in \mathcal{S}_{2K}} |\langle \nabla \mathcal{L}_n(\boldsymbol{\mathcal{W}}^*), \boldsymbol{\Delta}\rangle| \leq (1 - \sqrt{2}\varepsilon)^{-1} \max_{1 \leq j \leq \mathcal{N}(2K, \varepsilon)} |\langle \nabla \mathcal{L}_n(\boldsymbol{\mathcal{W}}^*), \bar{\boldsymbol{\Delta}}_j\rangle|.$$

Note that, from Lemma 1(b), $\log \mathcal{N}(2K, \varepsilon) \leq 2d_{\mathcal{M}}\log(9/\varepsilon)$, where $d_{\mathcal{M}} = K(P + L + 1)$. Let $\varepsilon = (2\sqrt{2})^{-1}$ and then $2\log(9/\varepsilon) < 7$. With (28), we can show that

$$\mathbb{P}\left\{\sup_{\boldsymbol{\Delta} \in \mathcal{S}_{2K}} |\langle \nabla \mathcal{L}_n(\boldsymbol{\mathcal{W}}^*), \boldsymbol{\Delta}\rangle| \geq 2t\right\} \leq 2\exp\left\{-cn\min\left(\frac{4t}{\sqrt{\kappa_U}}, \frac{16t^2}{\kappa_U}\right) + 7d_{\mathcal{M}}\right\}.$$

Take $t = 0.25\sqrt{\kappa_U}(\sqrt{d_{\mathcal{M}}/n} + \sqrt{\delta/n})$ where $\delta = O_p(1)$, and there exists some $\gamma$ such that $\sqrt{d_{\mathcal{M}}/n} \leq \gamma$ holds. We can finally show that

$$\mathbb{P}\left\{\sup_{\boldsymbol{\Delta} \in \mathcal{S}_{2K}} |\langle \nabla \mathcal{L}_n(\boldsymbol{\mathcal{W}}^*), \boldsymbol{\Delta}\rangle| \geq 0.5\sqrt{\kappa_U}\left(\sqrt{\frac{d_{\mathcal{M}}}{n}} + \sqrt{\frac{\delta}{n}}\right)\right\} \leq 2\exp\left\{-c_\gamma d_{\mathcal{M}} - c\delta\right\},$$

where $c_\gamma$ is some positive constant related to $\gamma$. $\square$

**Lemma 5** (LRSC). *Suppose that $n \gtrsim d_{\mathcal{M}}$, under Assumptions 4, there exists some constant $R > 0$, such that for any $\mathcal{W} \in \widehat{\mathcal{S}}_{2K}$ satisfying $\|\mathcal{W} - \mathcal{W}^*\|_{\mathrm{F}} \leq R$,*

$$\inf_{\boldsymbol{\Delta} \in \mathcal{S}_{2K}} \boldsymbol{\Delta}' \boldsymbol{H}_n(\mathcal{W}) \boldsymbol{\Delta} \geq \frac{\widetilde{\kappa}_1}{2}$$

*holds with probability*

$$1 - 4 \exp\left\{-0.25cn + 9d_{\mathcal{M}}\right\},$$

*where $\widetilde{\kappa}_1 = \kappa_1 - \kappa_U$. And $\kappa_1$ is defined in Lemma 8.*

*Proof.* We divide this proof into two parts.

1. RSC of $\mathcal{L}_n(\mathcal{W})$ at $\mathcal{W} = \mathcal{W}^*$.

We first show that, for all $\boldsymbol{\Delta} \in \mathcal{S}_{2K}$, the following holds with probability at least $1 - 2 \exp\left\{-0.25c(\kappa_L/\kappa_U)^2 n + 9d_{\mathcal{M}}\right\}$,

$$\boldsymbol{\Delta}' \boldsymbol{H}_n(\mathcal{W}^*) \boldsymbol{\Delta} \geq \widetilde{\kappa},$$

where $\widetilde{\kappa} = \kappa_0 - \kappa_L > 0$.

Let $\eta^i = \langle \mathcal{Z}^i, \mathcal{W}^* \rangle$ and denote $\widetilde{\boldsymbol{z}}^i = \sqrt{\phi''(\eta^i)} \boldsymbol{z}^i$ and we can see that

$$\boldsymbol{H}_n(\mathcal{W}^*) = \frac{1}{n} \sum_{i=1}^n \widetilde{\boldsymbol{z}}^i \widetilde{\boldsymbol{z}}^{i\prime} \text{ and, } \quad \boldsymbol{H}(\mathcal{W}^*) = \mathbb{E} \boldsymbol{H}_n(\mathcal{W}^*).$$

Denote $\boldsymbol{x}^i = \mathrm{vec}(\mathcal{X}^i)$. Here, for simplicity, we assume $\{\boldsymbol{x}^i\}_{i=1}^n$ to be independent gaussian vectors with mean zero and covariance matrix $\boldsymbol{\Sigma}$, where $c_x \boldsymbol{I} \leq \boldsymbol{\Sigma} \leq C_x \boldsymbol{I}$ for some $0 < c_x < C_x$. We will also use the notation of $\boldsymbol{\Delta}$ for its vectorized version, $\mathrm{vec}(\boldsymbol{\Delta})$, and we consider $\boldsymbol{\Delta}$ with unit $l_2$-norm.

Since $\|\langle \boldsymbol{\Delta}, \widetilde{\boldsymbol{z}}^i \rangle\|_{\psi_2} = \|\langle \boldsymbol{\Delta}, \sqrt{\phi''(\eta^i)} \boldsymbol{U}_G' \boldsymbol{\Sigma}^{1/2} \boldsymbol{w}^i \rangle\|_{\psi_2} \leq 0.25\sqrt{\kappa_U}$, where $\boldsymbol{w}^i$ is a standard gaussian vector and $\kappa_U = C_x C_u$, we can show that

$$\|\langle \boldsymbol{\Delta}, \widetilde{\boldsymbol{z}}^i \rangle^2 - \mathbb{E}\left[\langle \boldsymbol{\Delta}, \widetilde{\boldsymbol{z}} \rangle^2\right]\|_{\psi_1} \leq 2\|\langle \boldsymbol{\Delta}, \widetilde{\boldsymbol{z}}^i \rangle^2\|_{\psi_1} \leq 4\|\langle \boldsymbol{\Delta}, \widetilde{\boldsymbol{z}}^i \rangle\|_{\psi_2}^2 \leq \kappa_U,$$

where the first inequality comes from Remark 5.18 in (Vershynin, 2010) and second inequality comes from Lemma 5.14 in (Vershynin, 2010).

And hence, by the Beinstein-type inequality in Corollary 5.17 in (Vershynin, 2010), for any fixed $\boldsymbol{\Delta}$ such that $\|\boldsymbol{\Delta}\|_2 = 1$, we have

$$\mathbb{P}\left\{|\boldsymbol{\Delta}'(\boldsymbol{H}_n(\mathcal{W}^*) - \boldsymbol{H}(\mathcal{W}^*))\boldsymbol{\Delta}| \geq t\right\} = \mathbb{P}\left\{\frac{1}{n}\left|\sum_{i=1}^n \left\{\langle \boldsymbol{\Delta}, \widetilde{\boldsymbol{z}}^i \rangle^2 - \mathbb{E}\left[\langle \boldsymbol{\Delta}, \widetilde{\boldsymbol{z}} \rangle^2\right]\right\}\right| \geq t\right\}$$

$$\leq 2 \exp\left\{-cn \min\left(\frac{t}{\kappa_U}, \frac{t^2}{\kappa_U^2}\right)\right\}.$$

With similar covering number argument as presented in Lemma 2 in our paper, we can show that,

$$\mathbb{P}\left\{\sup_{\boldsymbol{\Delta} \in \mathcal{S}_{2K}} |\boldsymbol{\Delta}'(\boldsymbol{H}_n(\mathcal{W}^*) - \boldsymbol{H}(\mathcal{W}^*))\boldsymbol{\Delta}| \geq 2t\right\} \leq 2 \exp\left\{-cn \min\left(\frac{t}{\kappa_U}, \frac{t^2}{\kappa_U^2}\right) + 9d_{\mathcal{M}}\right\},$$

where $d_{\mathcal{M}} = K(P + L + 1)$.

Llet $t = 0.5\kappa_U$. By Assumption 4(C1), we can obtain that, when $n \gtrsim d_{\mathcal{M}}$,

$$\mathbb{P}\left\{\inf_{\boldsymbol{\Delta} \in \mathcal{S}_{2K}} \boldsymbol{\Delta}' \boldsymbol{H}_n(\mathcal{W}^*) \boldsymbol{\Delta} \leq \widetilde{\kappa}\right\} \leq 2 \exp\left\{-0.25cn + 9d_{\mathcal{M}}\right\},$$

where $\widetilde{\kappa} = \kappa_0 - \kappa_U > 0$.

2. LRSC of $\mathcal{L}_n(\mathcal{W})$ around $\mathcal{W}^*$.

Define the event $A = \{|\langle \mathcal{W}^*, \mathcal{Z}^i \rangle| > \tau \sup_{\boldsymbol{\Delta} \in \mathcal{S}_{2K}} |\langle \boldsymbol{\Delta}, \boldsymbol{z}^i \rangle|\}$ and construct the functions

$$\boldsymbol{h}_n(\mathcal{W}) = \frac{1}{n} \sum_{i=1}^n \phi''(\langle \mathcal{Z}^i, \mathcal{W} \rangle) \mathbb{I}_A \boldsymbol{z}^i \boldsymbol{z}^{i\prime} \text{ and, } \quad \boldsymbol{h}(\mathcal{W}) = \mathbb{E} \boldsymbol{h}_n(\mathcal{W}),$$

where $\tau$ is some positive constant to be selected according to Lemma 8. Since the difference between $\boldsymbol{h}_n(\cdot)$ and $\boldsymbol{H}_n(\cdot)$ is the indicator function, it holds that $\boldsymbol{H}_n(\cdot) \geq \boldsymbol{h}_n(\cdot)$.

We will finish the proof of LRSC in two steps. Firstly, we show that, with high probability, $\boldsymbol{h}_n(\boldsymbol{\mathcal{W}}^*)$ is positive definite on the restricted set $\mathcal{S}_{2K}$. Secondly, we bound the difference between $\boldsymbol{\Delta}' \boldsymbol{h}_n(\boldsymbol{\mathcal{W}}) \boldsymbol{\Delta}$ and $\boldsymbol{\Delta}' \boldsymbol{h}_n(\boldsymbol{\mathcal{W}}^*) \boldsymbol{\Delta}$, and hence show that $\boldsymbol{h}_n(\boldsymbol{\mathcal{W}})$ is locally positive definite around $\boldsymbol{\mathcal{W}}^*$. This naturally leads to the LRSC of $\mathcal{L}_n(\boldsymbol{\mathcal{W}})$ around $\boldsymbol{\mathcal{W}}^*$.

From Lemma 8, we can select $\tau$, such that $\boldsymbol{h}(\boldsymbol{\mathcal{W}}^*) \geq \kappa_1 \boldsymbol{I}$. Following similar arguments as in the first part, we can show that for all $\boldsymbol{\Delta} \in \mathcal{S}_{2K}$, the following holds with probability at least $1 - 2 \exp\{-0.25cn + 9d_{\mathcal{M}}\}$,

$$\boldsymbol{\Delta}' \boldsymbol{h}_n(\boldsymbol{\mathcal{W}}^*) \boldsymbol{\Delta} \geq \widetilde{\kappa}_1, \tag{29}$$

where $\widetilde{\kappa}_1 = \kappa_1 - \kappa_U > 0$.

In the meanwhile, for any $\boldsymbol{\mathcal{W}} \in \widehat{\mathcal{S}}_K$ such that $\|\boldsymbol{\mathcal{W}} - \boldsymbol{\mathcal{W}}^*\|_{\mathrm{F}} \leq R$, where $R$ can be specified later to satisfy some conditions,

$$\begin{aligned}
|\boldsymbol{h}_n(\boldsymbol{\mathcal{W}}) - \boldsymbol{h}_n(\boldsymbol{\mathcal{W}}^*)| &= \left| \frac{1}{n} \sum_{i=1}^n \phi''(\langle \boldsymbol{\mathcal{Z}}^i, \boldsymbol{\mathcal{W}} \rangle) \mathbb{I}_A \boldsymbol{z}^i \boldsymbol{z}^{i\prime} - \frac{1}{n} \sum_{i=1}^n \phi''(\langle \boldsymbol{\mathcal{Z}}^i, \boldsymbol{\mathcal{W}}^* \rangle) \mathbb{I}_A \boldsymbol{z}^i \boldsymbol{z}^{i\prime} \right| \\
&\leq \frac{1}{n} \sum_{i=1}^n \left| \phi''(\langle \boldsymbol{\mathcal{Z}}^i, \boldsymbol{\mathcal{W}} \rangle) - \phi''(\langle \boldsymbol{\mathcal{Z}}^i, \boldsymbol{\mathcal{W}}^* \rangle) \right| \mathbb{I}_A \boldsymbol{z}^i \boldsymbol{z}^{i\prime} \\
&= \frac{1}{n} \sum_{i=1}^n \left| \phi'''(\langle \boldsymbol{\mathcal{Z}}^i, \bar{\boldsymbol{\mathcal{W}}} \rangle) \langle \boldsymbol{\mathcal{Z}}^i, \boldsymbol{\mathcal{W}} - \boldsymbol{\mathcal{W}}^* \rangle \right| \mathbb{I}_A \boldsymbol{z}^i \boldsymbol{z}^{i\prime},
\end{aligned}$$

where $\bar{\boldsymbol{\mathcal{W}}}$ lies between $\boldsymbol{\mathcal{W}}$ and $\boldsymbol{\mathcal{W}}^*$, and $\phi'''(z) = e^z(1 - e^z)/(1 + e^z)^3$. Given the event $A$ holds, choose $R < \tau$, we can lower bound the term,

$$|\langle \boldsymbol{\mathcal{Z}}^i, \bar{\boldsymbol{\mathcal{W}}} \rangle| \geq |\langle \boldsymbol{\mathcal{Z}}^i, \boldsymbol{\mathcal{W}}^* \rangle| - \sup_{\boldsymbol{\Delta} \in \widehat{\mathcal{S}}_{2K}, \|\boldsymbol{\Delta}\|_{\mathrm{F}} \leq R} |\langle \boldsymbol{\mathcal{Z}}^i, \boldsymbol{\Delta} \rangle| \leq (\tau - R) \sup_{\boldsymbol{\Delta} \in \mathcal{S}_{2K}} |\langle \boldsymbol{\mathcal{Z}}^i, \boldsymbol{\Delta} \rangle|.$$

Notice that, for all $z \in \mathbb{R}$, the third order derivative of the function $\phi(z)$ is upper bounded as $|\phi'''(z)| \leq 1/|z|$. This relationship helps us further bound the term,

$$\begin{aligned}
\left| \phi'''(\langle \boldsymbol{\mathcal{Z}}^i, \bar{\boldsymbol{\mathcal{W}}} \rangle) \langle \boldsymbol{\mathcal{Z}}^i, \boldsymbol{\mathcal{W}} - \boldsymbol{\mathcal{W}}^* \rangle \right| &\leq \frac{|\langle \boldsymbol{\mathcal{Z}}^i, \boldsymbol{\mathcal{W}} - \boldsymbol{\mathcal{W}}^* \rangle|}{|\langle \boldsymbol{\mathcal{Z}}^i, \bar{\boldsymbol{\mathcal{W}}} \rangle|} \leq \frac{\sup_{\boldsymbol{\Delta} \in \widehat{\mathcal{S}}_{2K}, \|\boldsymbol{\Delta}\|_{\mathrm{F}} \leq R} |\langle \boldsymbol{\mathcal{Z}}^i, \boldsymbol{\Delta} \rangle|}{(\tau - R) \sup_{\boldsymbol{\Delta} \in \mathcal{S}_{2K}} |\langle \boldsymbol{\mathcal{Z}}^i, \boldsymbol{\Delta} \rangle|} \\
&\leq \frac{R \sup_{\boldsymbol{\Delta} \in \mathcal{S}_{2K}} |\langle \boldsymbol{\mathcal{Z}}^i, \boldsymbol{\Delta} \rangle|}{(\tau - R) \sup_{\boldsymbol{\Delta} \in \mathcal{S}_{2K}} |\langle \boldsymbol{\mathcal{Z}}^i, \boldsymbol{\Delta} \rangle|} = \frac{R}{\tau - R}.
\end{aligned}$$

Hence, we can show that

$$\mathbb{P}\left\{ \sup_{\boldsymbol{\Delta} \in \mathcal{S}_{2K}} |\boldsymbol{\Delta}'[\boldsymbol{h}_n(\boldsymbol{\mathcal{W}}) - \boldsymbol{h}_n(\boldsymbol{\mathcal{W}}^*)] \boldsymbol{\Delta}| \geq t \right\} \leq \mathbb{P}\left\{ \frac{R}{\tau - R} \sup_{\boldsymbol{\Delta} \in \mathcal{S}_{2K}} \frac{1}{n} \sum_{i=1}^n \boldsymbol{\Delta}' \boldsymbol{z}^i \boldsymbol{z}^{i\prime} \boldsymbol{\Delta} \geq t \right\}.$$

By setting $t = \alpha_{\mathrm{RSM}} R/(\tau - R)$, where $\alpha_{\mathrm{RSM}} = 3\kappa_U/2$, we can use the equation (16) in Lemma 2 to obtain, as long as $n \gtrsim d_{\mathcal{M}}$,

$$\mathbb{P}\left\{ \sup_{\boldsymbol{\Delta} \in \mathcal{S}_{2K}} |\boldsymbol{\Delta}'[\boldsymbol{h}_n(\boldsymbol{\mathcal{W}}) - \boldsymbol{h}_n(\boldsymbol{\mathcal{W}}^*)] \boldsymbol{\Delta}| \geq \frac{\alpha_{\mathrm{RSM}} R}{\tau - R} \right\} \leq 2 \exp\{-c_H n + 9d_{\mathcal{M}}\}.$$

By rearranging terms, this is equivalent to

$$\mathbb{P}\left\{ \underbrace{\inf_{\boldsymbol{\Delta} \in \mathcal{S}_{2K}} \boldsymbol{\Delta}' \boldsymbol{h}_n(\boldsymbol{\mathcal{W}}) \boldsymbol{\Delta} \leq \sup_{\boldsymbol{\Delta} \in \mathcal{S}_{2K}} \boldsymbol{\Delta}' \boldsymbol{h}_n(\boldsymbol{\mathcal{W}}^*) \boldsymbol{\Delta} - \frac{\alpha_{\mathrm{RSM}} R}{\tau - R}}_{\text{denoted as the event } B_1} \right\} \leq 2 \exp\{-c_H n + 9d_{\mathcal{M}}\}.$$

If we define the event $B_2 = \{\inf_{\boldsymbol{\Delta} \in \mathcal{S}_{2K}} \boldsymbol{\Delta}' \boldsymbol{h}_n(\boldsymbol{\mathcal{W}}^*) \boldsymbol{\Delta} \leq \widetilde{\kappa}_1\}$ and denote its complementary event by $B_2^c$, and from (29), we know that $\mathbb{P}(B_2) \leq 2 \exp\{-0.25cn + 9d_{\mathcal{M}}\}$. It can be seen that

$$\mathbb{P}\left( \left\{ \inf_{\boldsymbol{\Delta} \in \mathcal{S}_{2K}} \boldsymbol{\Delta}' \boldsymbol{h}_n(\boldsymbol{\mathcal{W}}) \boldsymbol{\Delta} \leq \widetilde{\kappa}_1 - \frac{\alpha_{\mathrm{RSM}} R}{\tau - R} \right\} \cap B_2^c \right) \leq \mathbb{P}(B_1 \cap B_2^c) \leq \mathbb{P}(B_1)$$

So, if we choose $R$ to be sufficiently small, such that $\alpha_{\text{RSM}} R/(\tau - R) \leq \widetilde{\kappa}_1/2$, it holds that,

$$\mathbb{P}\left\{\inf_{\boldsymbol{\Delta} \in \mathcal{S}_{2K}} \boldsymbol{\Delta}' \boldsymbol{h}_n(\boldsymbol{\mathcal{W}})\boldsymbol{\Delta} \leq \frac{\widetilde{\kappa}_1}{2}\right\} \leq \mathbb{P}\left(\left\{\inf_{\boldsymbol{\Delta} \in \mathcal{S}_{2K}} \boldsymbol{\Delta}' \boldsymbol{h}_n(\boldsymbol{\mathcal{W}})\boldsymbol{\Delta} \leq \widetilde{\kappa}_1 - \frac{\alpha_{\text{RSM}} R}{\tau - R}\right\} \cap B_2^c\right) + \mathbb{P}(B_2).$$

This, together with $\boldsymbol{H}_n(\cdot) \geq \boldsymbol{h}_n(\cdot)$, leads us to conclude that, when $n \gtrsim d_{\mathcal{M}}$, there exists some $R > 0$, such that for any $\boldsymbol{\mathcal{W}} \in \widehat{\mathcal{S}}_{2K}$ satisfying $\|\boldsymbol{\mathcal{W}} - \boldsymbol{\mathcal{W}}^*\|_{\text{F}} \leq R$,

$$\inf_{\boldsymbol{\Delta} \in \mathcal{S}_{2K}} \boldsymbol{\Delta}' \boldsymbol{H}_n(\boldsymbol{\mathcal{W}})\boldsymbol{\Delta} \geq \frac{\widetilde{\kappa}_1}{2},$$

holds with probability

$$1 - 2\exp\left\{-0.25 c_H n + 9 d_{\mathcal{M}}\right\} - 2\exp\left\{-0.25 c n + 9 d_{\mathcal{M}}\right\}.$$

We accomplished our proof of the LRSC of $\mathcal{L}_n(\boldsymbol{\mathcal{W}})$ around $\boldsymbol{\mathcal{W}}^*$. $\qquad\square$

**Lemma 6.** *For two sub-gaussian random variables, $X$ and $Y$, when $\mathbb{E}(XY) = \mathbb{E}X\mathbb{E}Y$, i.e. $X$ is independent to $Y$, it holds that*

$$\|XY\|_{\psi_1} \leq \|X\|_{\psi_2}\|Y\|_{\psi_2}.$$

**Lemma 7.**

$$\|\phi'(\eta^i) - y^i\|_{\psi_2} \leq 0.25.$$

*Proof.* Firstly, we observe that

$$
\begin{aligned}
\mathbb{E}\left(\exp\{\lambda[\phi'(\eta^i) - y^i]\}|z^i\right) &= \exp\{\lambda\phi'(\eta^i)\}\exp\{-\phi(\eta^i)\} + \{\lambda[\phi'(\eta^i) - 1]\}\exp\{\eta^i - \phi(\eta^i)\} \\
&= \exp\{\lambda\phi'(\eta^i) - \phi(\eta^i)\}[1 + \exp\{\eta^i - \lambda\}] \\
&= \exp\{\phi(\eta^i - \lambda) - \phi(\eta^i) + \lambda\phi'(\eta^i)\} \\
&= \exp\{0.5\lambda^2\phi''(\eta^*)\} \leq \exp\{0.125\lambda^2\}.
\end{aligned}
$$

It then holds that $\mathbb{E}\left(\exp\{\lambda[\phi'(\eta^i) - y^i]\}\right) = \mathbb{E}\{\mathbb{E}\left(\exp\{\lambda[\phi'(\eta^i) - y^i]\}|z^i\right)\} \leq \exp\{0.125\lambda^2\}$, and this implies that $\|\phi'(\eta^i) - y^i\|_{\psi_2} \leq 0.25$.

$\qquad\square$

**Lemma 8.** *Under Assumption 4, there exists a universal constant $\tau > 0$ such that $\boldsymbol{h}(\boldsymbol{\mathcal{W}}^*) \geq \kappa_1 \boldsymbol{I}$, where $\kappa_1$ is a positive constant.*

*Proof.* We first show that for any $p_0 \in (0, 1)$, there exists a constant $\tau$ such that

$$\mathbb{P}(|\langle \boldsymbol{\mathcal{W}}^*, \boldsymbol{\mathcal{Z}}^i\rangle| > \tau \sup_{\boldsymbol{\Delta} \in \mathcal{S}_{2K}} |\langle \boldsymbol{\Delta}, z^i\rangle|) \geq p_0.$$

We would separately show that

$$\mathbb{P}(\underbrace{|\langle \boldsymbol{\mathcal{W}}^*, \boldsymbol{\mathcal{Z}}^i\rangle| > c_1\sqrt{d_{\mathcal{M}}}}_{\text{denoted by event } D_1}) \geq \frac{p_0 + 1}{2} \text{ and, } \mathbb{P}(\underbrace{\sup_{\boldsymbol{\Delta} \in \mathcal{S}_{2K}} |\langle \boldsymbol{\Delta}, z^i\rangle| \leq c_2\sqrt{d_{\mathcal{M}}}}_{\text{denoted by event } D_2}) \geq \frac{p_0 + 1}{2},$$

for some positive constants $c_1$ and $c_2$. Using the relationship $\mathbb{P}(D_1 \cap D_2) = P(D_1) + P(D_2) - P(D_1^c \cup D_2^c) \geq P(D_1) + P(D_2) - 1$, it follows naturally that

$$\mathbb{P}(|\langle \boldsymbol{\mathcal{W}}^*, \boldsymbol{\mathcal{Z}}^i\rangle| > \tau \sup_{\boldsymbol{\Delta} \in \mathcal{S}_{2K}} |\langle \boldsymbol{\Delta}, z^i\rangle|) \geq \frac{p_0 + 1}{2} + \frac{p_0 + 1}{2} - 1 = p_0, \tag{30}$$

where $\tau = c_1/c_2$.

Since $x^i$ is a gaussian vector with mean zero and covariance $\boldsymbol{\Sigma}$, $z^i = \boldsymbol{U}_G'x^i$ is a zero-mean gaussian vector with covariance given by $\boldsymbol{U}_G'\boldsymbol{\Sigma}\boldsymbol{U}_G$, and $\langle \boldsymbol{\mathcal{W}}^*, \boldsymbol{\mathcal{Z}}^i\rangle = \langle \text{vec}(\boldsymbol{\mathcal{W}}^*), z^i\rangle$ also follows a normal distribution with mean zero and variance (also its sub-Gaussian norm) upper bounded by $\kappa_U\|\boldsymbol{\mathcal{W}}^*\|_{\text{F}}^2$, where $\kappa_U = C_x C_u$.

Since from Assumption 4(C2), $\|\boldsymbol{\mathcal{W}}^*\|_{\mathrm{F}}^2 \geq \alpha\sqrt{d_{\mathcal{M}}}$, we can take $c_1$ to be sufficiently small such that

$$\mathbb{P}(D_1) = \mathbb{P}\left(\frac{|\langle \boldsymbol{\mathcal{W}}^*, \boldsymbol{\mathcal{Z}}^i\rangle|}{\kappa_U \|\boldsymbol{\mathcal{W}}^*\|_{\mathrm{F}}} > \frac{c_1}{\kappa_U \alpha}\right) \geq \mathbb{P}(|x| > \frac{c_1}{\kappa_U \alpha}) \geq \frac{p_0 + 1}{2},$$

where $x$ is a gaussian variable with variance upper bounded by 1.

Then, we can also observe that, for any fixed $\boldsymbol{\Delta} \in \mathcal{S}_{2K}$, $\langle \boldsymbol{\Delta}, \boldsymbol{z}^i\rangle$ is a gaussian variable with zero mean and variance upper bounded by $\kappa_U$. We can use the concentration inequality for gaussian random variable to establish that

$$\mathbb{P}\left(|\langle \boldsymbol{\Delta}, \boldsymbol{z}^i\rangle| \geq t\right) \leq 2\exp(-\frac{t^2}{\kappa_U}),$$

for all $t \in \mathbb{R}$. We can further use the union bound to show that

$$\mathbb{P}\left(\sup_{\boldsymbol{\Delta} \in \mathcal{S}_{2K}} |\langle \boldsymbol{\Delta}, \boldsymbol{z}^i\rangle| \geq t\right) \leq 2\exp(-\frac{t^2}{\kappa_U} + 7d_{\mathcal{M}}).$$

Let $t = c_2\sqrt{d_{\mathcal{M}}}$ for some positive constant $c_2 > \sqrt{7\kappa_U}$. We can choose $c_2$ large enough such that

$$\mathbb{P}\left(\sup_{\boldsymbol{\Delta} \in \mathcal{S}_{2K}} |\langle \boldsymbol{\Delta}, \boldsymbol{z}^i\rangle| \geq c_2\sqrt{d_{\mathcal{M}}}\right) \leq \frac{1 - p_0}{2}.$$

The probability at (30) is hence shown.

Now, we will take a look at the matrix $\boldsymbol{h}(\boldsymbol{\mathcal{W}}^*)$ and show that it is positive definite. Same as in Lemma 5, we denote the event $A = \{|\langle \boldsymbol{\mathcal{W}}^*, \boldsymbol{\mathcal{Z}}^i\rangle| > \tau \sup_{\boldsymbol{\Delta} \in \mathcal{S}_{2K}} |\langle \boldsymbol{\Delta}, \boldsymbol{z}^i\rangle|\}$, and its complement by $A^c$, then

$$\boldsymbol{\Delta}'\boldsymbol{h}(\boldsymbol{\mathcal{W}}^*)\boldsymbol{\Delta} = \frac{1}{n}\mathbb{E}\left\{\sum_{i=1}^n \phi''(\langle \boldsymbol{\mathcal{Z}}^i, \boldsymbol{\mathcal{W}}\rangle)\mathbb{I}_A \boldsymbol{\Delta}'\boldsymbol{z}^i\boldsymbol{z}^{i\prime}\boldsymbol{\Delta}\right\}$$

$$= \frac{1}{n}\mathbb{E}\left\{\sum_{i=1}^n \phi''(\langle \boldsymbol{\mathcal{Z}}^i, \boldsymbol{\mathcal{W}}\rangle)\boldsymbol{\Delta}'\boldsymbol{z}^i\boldsymbol{z}^{i\prime}\boldsymbol{\Delta}\right\} - \frac{1}{n}\mathbb{E}\left\{\sum_{i=1}^n \phi''(\langle \boldsymbol{\mathcal{Z}}^i, \boldsymbol{\mathcal{W}}\rangle)\mathbb{I}_{A^c}\boldsymbol{\Delta}'\boldsymbol{z}^i\boldsymbol{z}^{i\prime}\boldsymbol{\Delta}\right\}$$

$$= \boldsymbol{\Delta}'\boldsymbol{H}(\boldsymbol{\mathcal{W}}^*)\boldsymbol{\Delta} - \frac{1}{n}\mathbb{E}\left\{\sum_{i=1}^n \phi''(\langle \boldsymbol{\mathcal{Z}}^i, \boldsymbol{\mathcal{W}}\rangle)\mathbb{I}_{A^c}\boldsymbol{\Delta}'\boldsymbol{z}^i\boldsymbol{z}^{i\prime}\boldsymbol{\Delta}\right\}$$

$$\overset{(i)}{\geq} \kappa_0 - \frac{1}{n}\sqrt{\mathbb{E}\left\{\sum_{i=1}^n [\phi''(\langle \boldsymbol{\mathcal{Z}}^i, \boldsymbol{\mathcal{W}}\rangle)]^2 \langle \boldsymbol{\Delta}, \boldsymbol{z}^i\rangle^4\right\}}\sqrt{\mathbb{E}\left\{\sum_{i=1}^n \mathbb{I}_{A^c}\right\}}$$

$$\overset{(ii)}{\geq} \kappa_0 - \frac{\sqrt{3\kappa_U}}{4}(1 - p_0),$$

where (i) follows from Assumption 4(C1), And since $\langle \boldsymbol{\Delta}, \boldsymbol{z}^i\rangle$ is a gaussian variable with mean zero and variance bounded by $\kappa_U$, its fourth moment is bounded by $3\kappa_U$. Also, by $\phi''(z) \in (0, 0.25)$ for all $z \in \mathbb{R}$, (ii) can be shown.

Here, we can take $p_0$ to be small enough such that $0.25\sqrt{3\kappa_U}(1 - p_0) \leq 0.5\kappa_0$ holds. It follows that

$$\boldsymbol{\Delta}'\boldsymbol{h}(\boldsymbol{\mathcal{W}}^*)\boldsymbol{\Delta} \geq \kappa_1,$$

with $\kappa_1 = 0.5\kappa_0$. We hence accomplished the proof of the lemma. $\qquad\square$

## A.5 ADDITIONAL EXPERIMENTS

### A.5.1 MORE EXPERIMENTS FOR THEORETICAL RESULTS

We first provide some additional implementation details for Section 4.1 in the paper. We consider two types of inputs for verification of the sample complexity in Theorem 1. For the time-dependent inputs, we generate a sequence of vectors $\{\boldsymbol{x}^i\}$ with size $\mathbb{R}^d$ from a stationary VAR(1) process, where $d = 245$ or 192. The VAR(1) process has a transition matrix with spectral norm less than

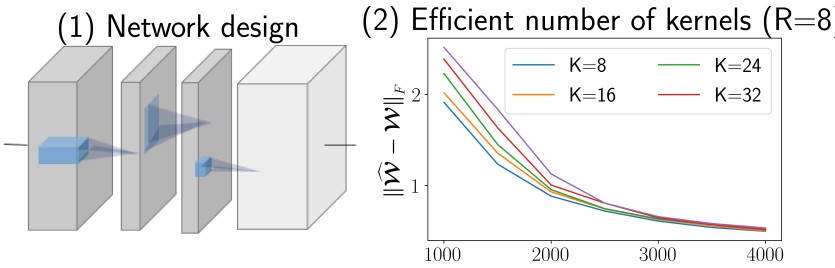

Figure 7: Additional experiments for efficient number of kernels.

1 to ensure stationarity, and a white noise sequence which follows multivariate standard normal distribution. The first 50 simulated data points in the sequence are discarded to alleviate the influence of initialization, and the length of the resulting sequence is equal to the training sample size plus the testing sample size, i.e. $n + 200$. We then reshape $\{\boldsymbol{x}^i\}$ into a sequence of tensors of shape $(7, 5, 7)$ or $(8, 8, 3)$, which are the time-dependent inputs for experiments under S1/S2 or S3/S4.

Next, we conduct extra experiments for efficient number of kernels for a CP block design. This study uses $32 \times 32$ inputs with 16 channels, and we set stride to 1 and pooling sizes to $(5, 5)$. We generate the orthonormal factor matrices $\{\boldsymbol{H}^{(j)}, 1 \leq j \leq 4\}$, where $\boldsymbol{H}^{(1)}$ is of size $\mathbb{R}^{8 \times R}$, $\boldsymbol{H}^{(2)}$ is of size $\mathbb{R}^{8 \times R}$, $\boldsymbol{H}^{(3)}$ is of size $\mathbb{R}^{16 \times R}$ and $\boldsymbol{H}^{(4)}$ is of size $\mathbb{R}^{K \times R}$ where $R = 8$ and $K \in \{8, 16, 24, 32\}$. If we denote the orthonormal column vectors of $\boldsymbol{H}^{(j)}$ by $\boldsymbol{h}_r^{(j)}$ where $1 \leq r \leq R$ and $1 \leq j \leq 4$, the stacked kernel tensor $\mathcal{A}$ can then be generated as $\mathcal{A} = \sum_{r=1}^{R} \boldsymbol{h}_r^{(1)} \circ \boldsymbol{h}_r^{(2)} \circ \boldsymbol{h}_r^{(3)} \circ \boldsymbol{h}_r^{(4)}$. And it can be seen that $\mathcal{A}$ has a CP rank of $R = 8$. We split the stacked kernel tensor $\mathcal{A}$ along the kernel dimension to obtain $\{\mathcal{A}_k, 1 \leq k \leq K\}$ and generate the corresponding fully-connected weight tensors $\{\mathcal{B}_k, 1 \leq k \leq K\}$ with standard normal entries. The parameter tensor $\mathcal{W}$ is hence obtained, and we further normalize it to have unit Frobenius norm to ensure the comparability of estimation errors between different $K$s. The block structure in Figure 7(1) is employed to train the network, and it is equivalent to the bottleneck structure with a CP decomposition on $\mathcal{A}$; see Kossaifi et al. (2020b). We can see that as $K$ increases, there is more redundancy in the network parameters. Fifty training sets are generated for each training size, and we stop training when the target function drops by less than $10^{-5}$. From Figure 7(2), the estimation error increases as $K$ is larger, and the difference is more pronounced when training size is small.

### A.5.2 ABLATION STUDIES ON MORE BLOCK DESIGNS

In this section, we present the results from the studies of $K/R$ ratios on two popular networks structures, namely, ResNeXt (Xie et al., 2017) and Shufflenet v2 (Ma et al., 2018). To start with, we will show the bottleneck block structure and its respective $K/R$ ratio in ResNeXt and Shufflenet.

In Figure 8, we represent the bottleneck block structure of ResNeXt consisting of 3 layers: a $1 \times 1$ convolution layer, a $3 \times 3$ group convolution layer followed by another $1 \times 1$ convolution layer. During group convolution phase, there are $g$ parallel paths, each has a bottleneck size of $r$, which

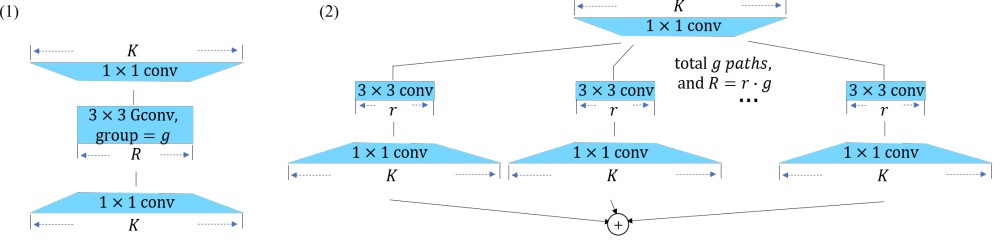

Figure 8: Equivalent ResNext building blocks. Both (1) and (2) represent a block of ResNeXt with cardinality $g$, with $R = r \cdot g$. Here, the expansion ratio $t = K/R$, and takes values of 1,2 and 4.

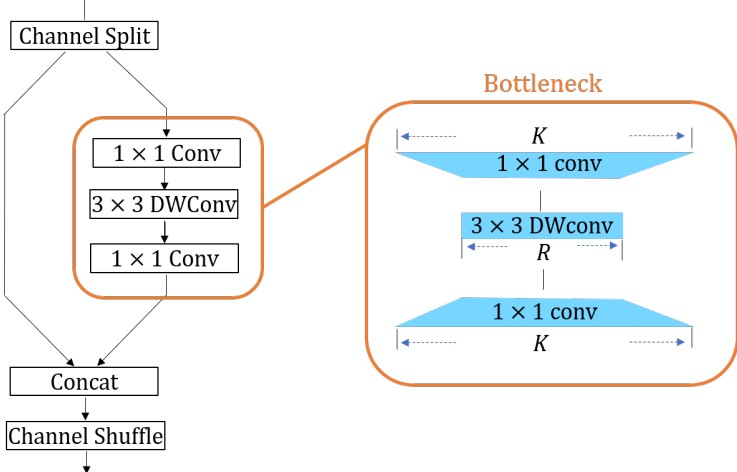

Figure 9: A basic block of Shufflenet v2, with its bottleneck structure depict explicitly. Here, the expansion ratio $t = K/R$, and takes values of 1,2 and 4.

Table 3: Test accuracy(%) of ResNeXt and Shufflenet on SVNH.

| $t$ | ResNeXt | #FLOPs | #Params | Shufflenet | #FLOPs | #Params |
|---|---|---|---|---|---|---|
| 1 | 96.20 | 0.05GMac | 1.45M | 95.93 | 0.05GMac | 1.33M |
| 2 | 96.25 | 0.07GMac | 2.13M | 96.08 | 0.08GMac | 1.67M |
| 4 | 96.20 | 0.14GMac | 4.44M | 96.03 | 0.16GMac | 3.84M |

is also the rank of the output channel in each individual path; see also Figure 1 in Xie et al. (2017). Since the output from each path is aggregated via summation, the rank of the output channel of the entire bottleneck block is equal to $R = r \cdot g$. Subsequently, we can define the expansion ratio of this block to be $t = K/R$.

In Figure 9, we represent the bottleneck block structure of Shufflenet. It consists of 3 layers: a $1 \times 1$ convolution layer, a $3 \times 3$ depthwise convolution layer followed by another $1 \times 1$ convolution layer. So, the only difference between this block structure and that in ResNet (He et al., 2016) is that the full $3 \times 3$ convolution is replaced by the depthwise separable convolution. It follows that the expansion ratio is $t = K/R$.

We conducted the experiments on the SVNH dataset and take $t = 1, 2, 4$, since $t = 2$ or $t = 4$ in particular, is commonly used in practice. We followed the design of ResNeXt-50 (see Table 1 in Xie et al. (2017)) and Shufflenet v2-50 (see Appendix Table 2 in Ma et al. (2018)), only be removing the three convolution block in "conv5" to avoid overfitting. And we set $R = 24, 58, 116, 232$ as the bottleneck rank for "conv1"-"conv4", respectively. And $K = t \cdot R$ is taken according to different values of $t$. The implementations are similar to before. Following the practice in He et al. (2016), we adopt batch normalization(BN) (Ioffe & Szegedy, 2015) right after each convolution and before the ReLU activation. For ResNeXt, the cardinality of the group convolution is set to be 32, and we initialize the weights as in He et al. (2015). We use stochastic gradient descent with weight decay $10^{-4}$, momentum 0.9 and mini-batch size 128. The learning rate starts from 0.1 and is divided by 10 for every 100 epochs. We stop training after 350 epochs, since the training accuracy hardly changes. The training accuracy is approximately 96%. We set seeds 1-5 and report the worst case scenario of the test accuracy in Table 3.

We can see that, when $t = 1$, the test accuracy is comparable to when $t = 2$ or 4, and the number of parameters are reduced by a lot. In fact, the ratio of number of parameters at a single bottleneck block for $t = 1, 2$ and 4 is roughly $1 : 2 : 4$, for both networks. This implies that the number of parameters will increase dramatically when we have a deeper CNN, for instance, ResNeXt-101. Hence, it is always recommended to keep $t = 1$ to achieve more parameter efficiency without sacrificing the test accuracy.

