# OpenReview forum: "Rethinking Compressed Convolution Neural Network from a Statistical Perspective"
_ICLR.cc/2021/Conference — Reject_

### Official Review · AnonReviewer1 · 2020-10-25

**Rating:** 5
**Confidence:** 3

**Review:**

This paper formulates CNNs with high-order inputs into an explicit Tucker model, and provides sample complexity analysis to CNNs as well as compressed CNNs via tensor decomposition. Experiments support their theoretical results. Sample complexity analysis of CNNs and compressed CNNs is an interesting topic. This paper is well written and is easy to follow.

Cons:

1. Technical novelty is limited. It has been well understood that CNNs can be formulated as a tensor model, see Lebedev et al. (2015); Hayashi et al. (2019);  Kossaifi et al. (2019). By assuming a realization model, the sample complexity analysis of CNNs and compressed CNNs was transferred to the sample complexity analysis of tensor regression model and low-rank tensor regression model. The latter analysis is not new given a rich literature on this topic, see e.g., Bahadori et al. (2014); Yu and Liu (2016); Kossaifi et al. (2020).

Lebedev, V., et al. "Speeding-up convolutional neural networks using fine-tuned CP-decomposition." 3rd International Conference on Learning Representations, ICLR 2015-Conference Track Proceedings. 2015.

Hayashi, Kohei, et al. "Exploring Unexplored Tensor Network Decompositions for Convolutional Neural Networks." Advances in Neural Information Processing Systems. 2019.

Kossaifi, Jean, et al. "T-net: Parametrizing fully convolutional nets with a single high-order tensor." Proceedings of the IEEE Conference on Computer Vision and Pattern Recognition. 2019.

Bahadori, Mohammad Taha, Qi Rose Yu, and Yan Liu. "Fast multivariate spatio-temporal analysis via low rank tensor learning." Advances in neural information processing systems. 2014.

Yu, Rose, and Yan Liu. "Learning from multiway data: Simple and efficient tensor regression." International Conference on Machine Learning. 2016.

Kossaifi, Jean, et al. "Tensor regression networks." Journal of Machine Learning Research 21.123 (2020): 1-21.


2. The sample complexity analysis is for the global minimizer from a non-convex optimization, see (5).  It would be more interesting to study the sample complexity analysis for the estimator from some polynomial algorithm.

---

> ### Author Response · Authors · 2020-11-22
> **Response to Reviewer 1 (1/2)**
>
> Thank you for your comment!
>
> Following the suggestions from you and another reviewer, we have reorganized the whole Section 1 "Introduction" to better motivate our work. Specifically,
>
> a) we have made more comparisons between our work and other tensor methods for CNN compression;
>
> b) we have better motivated the method for statistical sample complexity, which is  model-based, rather than algorithm-based.
>
> __Con 1__
>
> > 1. Technical novelty is limited. It has been well understood that CNNs can be formulated as a tensor model, see Lebedev et al. (2015); Hayashi et al. (2019); Kossaifi et al. (2019). By assuming a realization model, the sample complexity analysis of CNNs and compressed CNNs was transferred to the sample complexity analysis of tensor regression model and low-rank tensor regression model. The latter analysis is not new given a rich literature on this topic, see e.g., Bahadori et al. (2014); Yu and Liu (2016); Kossaifi et al. (2020).
>
> Tensor methods have been widely adopted to compress or decompose CNN layers [1]-[8]. However, all of these work simply summarize the weights into tensors, and then apply CP, Tucker or Tensor-Train decomposition to the summarized weight tensors. They do not explicitly account for the interaction between weights and inputs and, to our best knowledge, the whole CNN model has never been depicted in a tensor regression form before.
>
> In fact, such a formulation is only meaningful when considering a high-order convolution (in our case, a general $N$ dimensional convolution), while most of the existing work ([1]-[7]) focuses on 2D convolution. Kossaifi et al. [8] considered using CP to factorize high-order convolution kernels. However, they focused on empirical studies on the factorized layers, instead of formulating the CNN as a whole. So, establishing the tensor regression form for a high-order CNN can be considered as one of our contributions.
>
> By building this connection, more tensor regression based methodology can hopefully be introduced into the theoretical investigation of CNNs.
>
> Moreover, as discussed at the end of Section 2.3 (Page 5), though we assume a simple regression model at equation (5), our sample complexity analysis can be extended to classification problems as well. We have developed one whole new section *"A.4 Classification problems"* to the Appendix on pages 24-31. In the section, we have provided some theorem and corollaries for both binary and multiclass classification problems, and provide a complete proof of the new theorem.
>
> Now, we discuss in details how this paper relates and differs from the references you provided.
>
> __Relation to [1],[2]\&[4]__
>
> It is common in literature to apply tensor decomposition to CNNs layer-by-layer. On the one hand, when tensor decomposition is applied to the convolution layers, it corresponds to various block designs in [1]\&[2], which we focus on in our theoretical study (Theorem 2). On the other hand, when tensor decomposition is applied to the fully-connected layer instead, it corresponds to the  tensor regression layer in [4].
>
> Parameter efficiency in [1],[2]\&[4] was heuristically justified by methods, such as FLOPs counting, naive parameter counting, the amount of space savings and/or empirical running time. However, there is still lack of a theoretical study to understand the mechanism of how tensor decomposition can compress CNNs. This paper attempts to fill this gap from statistical perspectives.
>
> In our paper, we focus on studying the low-rank structure on convolution kernel $\mathcal{A}$ (corresponding to various block structures). In fact, our CNN formulation allows us to easily extend the theoretical analysis to the fully-connected weights $\mathcal{B}$ with low rank structure. And we can hence establish the sample complexity analysis for the CNNs with tensor regression layer [4], which we leave as future work; see Section 5.
>
> __Relation to [3]__
>
> Our formulation is similar to [3] in that we also parametrize the network weights into a single tensor. While [3] assumes the tensor to have a heuristic Tucker form, we replicate the layer-by-layer operations of CNNs and show that the summarized tensor has a "nested doll" structure. In other words, the low-rank structure of the previous layer is nested within that of the current layer. [3] also uses the heuristic naive parameter counting (compression ratio) method to measure the efficiency.

---

> > ### Author Response · Authors · 2020-11-22
> > **Response to Reviewer 1 (2/2)**
> >
> > __Relation to [9]\&[10]__
> >
> > Both [9]\&[10] and our paper are related to a hot topic in the literature, which is to bound the estimation error in the form of
> > $\|\mathcal{\widehat{W}}^t - \mathcal{W}^\*\| \leq \|\mathcal{\widehat{W}}^t - \mathcal{\widehat{W}}\| + \|\mathcal{\widehat{W}} - \mathcal{W}^\*\|$,
> > where  $\mathcal{W}^*$ is the true underlying weight, $\mathcal{\widehat{W}}^t$ is the optimizer from the $t$th iteration step and $\mathcal{\widehat{W}}$ is the global optimizer.  The key difference is that, [9]\&[10] target at bounding $\|\mathcal{\widehat{W}}^t-\mathcal{\widehat{W}}\|$, while our paper targets at bounding $\|\mathcal{\widehat{W}}-\mathcal{W}^*\|$.
> >
> > As a result, the bounds developed in [9]\&[10] are related to a certain algorithm, while our theoretical results are independent of any training algorithms. We are more interested in the network architecture itself. Specifically speaking, as discussed in Subsection 1.1, we conduct statistical sample complexity analysis, to theoretically explain how much compressibility is achieved in a compressed network architecture.
> >
> > Since CNN architectures can be easily designed in Tensorflow and its training algorithms are well studied, we can directly apply them to train our model and hence obtain the trained weights for $\mathcal{A}_k$ and $\mathcal{B}_k$. And then, we can use the relationship $\mathcal{W} = \sum_\{k=1}^K\mathcal{A}_k\otimes\mathcal{B}_k$ to recover the composite weight $\mathcal{W}$.
> >
> > However, as both bounds are of equal importance in understanding the behavior of the estimation error, it will be interesting to investigate $\|\mathcal{\widehat{W}}^t-\mathcal{\widehat{W}}\|$ in our future work.
> >
> > __Con 2__
> >
> > > 2. The sample complexity analysis is for the global minimizer from a non-convex optimization, see (5). It would be more interesting to study the sample complexity analysis for the estimator from some polynomial algorithm.
> >
> > Yes, you are right in that, although the global minimizer exists theoretically, it cannot be obtained empirically, since the objective function at (6) [(5) in the previous draft] on page 5 is nonconvex with respect to $\mathcal{B}_k$s and $\mathcal{A}_k$s.
> >
> > However, as explained in our newly added Subsection 1.1, this paper attempts to conduct a statistical sample complexity analysis, to theoretically explain how much compressibility is achieved in a compressed network architecture. We are more interested in the trained model itself, and the possible optimization errors may be ignored when comparing two CNN architectures. To better clarify our problem, we have introduced a prediction error $\mathcal{E}(\mathcal{\widehat{W}})=\sqrt{E_{\textrm{x}}|F_{\textrm{CNN}}(\textrm{x}, \mathcal{\widehat{W}})- F_{\textrm{CNN}}(\textrm{x}, \mathcal{W}^*)|^2}$, and in Theorems 1\&2, we also provide the bound for $\mathcal{E}(\mathcal{\widehat{W}})$.
> >
> > Moreover, our analysis framework is closely related to [11]. And they actually do not consider the computational complexity or algorithm convergence.
> >
> > [1] Lebedev, V., et al. "Speeding-up convolutional neural networks using fine-tuned CP-decomposition." 3rd International Conference on Learning Representations. 2015.
> >
> > [2] Hayashi, Kohei, et al. "Exploring Unexplored Tensor Network Decompositions for Convolutional Neural Networks." Advances in Neural Information Processing Systems. 2019.
> >
> > [3] Kossaifi, Jean, et al. "T-net: Parametrizing fully convolutional nets with a single high-order tensor." Proceedings of the IEEE Conference on Computer Vision and Pattern Recognition. 2019.
> >
> > [4] Kossaifi, Jean, et al. "Tensor regression networks." Journal of Machine Learning Research 21.123 (2020): 1-21.
> >
> > [5] Astrid, Marcella, and Seung-Ik Lee. "Cp-decomposition with tensor power method for convolutional neural networks compression." *2017 IEEE International Conference on Big Data and Smart Computing (BigComp)*. IEEE, 2017.
> >
> > [6] Yong-Deok Kim, et al. "Compression of deep convolutional neural networks for fast and low power mobile applications.“ In International Conference on Learning Representations, 2016.
> >
> > [7] Kossaifi, Jean, et al. "Tensor contraction layers for parsimonious deep nets." *Proceedings of the IEEE Conference on Computer Vision and Pattern Recognition Workshops*. 2017.
> >
> > [8] Kossaifi, Jean, et al. "Factorized Higher-Order CNNs with an Application to Spatio-Temporal Emotion Estimation." *Proceedings of the IEEE/CVF Conference on Computer Vision and Pattern Recognition*. 2020.
> >
> > [9] Bahadori, Mohammad Taha, Qi Rose Yu, and Yan Liu. "Fast multivariate spatio-temporal analysis via low rank tensor learning." Advances in neural information processing systems. 2014.
> >
> > [10] Yu, Rose, and Yan Liu. "Learning from multiway data: Simple and efficient tensor regression." International Conference on Machine Learning. 2016.
> >
> > [11] Du, Simon S., et al. "How many samples are needed to estimate a convolutional neural network?." *Advances in Neural Information Processing Systems*. 2018.

---

### Official Review · AnonReviewer4 · 2020-10-25
**Interesting theoretical results on CNN but some important aspects not analyzed nor discussed**

**Rating:** 5
**Confidence:** 4

**Review:**

This paper provides theoretical analysis of the estimating power of CNN (3 and 5 layers). By formulating the problem using tensors, the authors showed that the estimating error of the learned CNN weights with respect to the true weights is of the order $\sqrt{d/n}$ where $d$ measures model complexity and $n$ is the training sample size. In addition, the authors considered low rank approximation to the convolution tensor through CP and Tucker decompositions, and they derived convergence result for the CNN weights in this case. The authors then applied these results to analyze different block designs through numerical experiments and ablation studies.

The writing is generally clear. The use of tensors give a very compact representation of an CNN, however tensor notations and indices can quickly become complicated in higher dimensions. The convergence results are a nice contribution to the growing literature on neural networks' theoretical analysis. In particular, the analysis of various tensor decompositions help guide the design of blocks in CNNs.

Although the results are interesting, I feel that important aspects of CNNs were not analyzed nor discussed in this paper. Here are some of my comments and questions.

One of the main uses of CNNs is to perform classification where there is a softmax layer after the fully connected layer. Hence model (2) does not reflect what is done in practice and $\boldsymbol{y}$ should be a vector of probabilities. In fact I think the authors did classification on CIFAR-10 and SVHN in the numerical experiments (although it is not mentioned) using residual networks and not the model in (2). Also, why do you add sub-Gaussian errors to CNN outputs as in (2)?

In the experiments, the authors used ReLU instead of linear activation, and there is batch normalization between convolution layers. In addition, there are residual connections in the ResNet (block) architecture. All these factors that affect convergence rates were not considered in the theoretical analysis. Although I understand that these issues were omitted to simplify the proofs, I feel that there should be consistency between the theory and implementation parts. Hence Table 2 and Figure 3 do not necessarily provide empirical evidence to the theoretical results derived, as these networks have different architectures than the ones considered in Section 2. Consequently, the results hold true for a special type of CNN and it is not clear to me whether these results will still hold for CNNs used in practice (with non-linear activation, batch norm, max pooling etc.), or when the true weight tensor does not have the same tensor product structure as the learned weight tensor.

Equation (5) seems to assume that you can optimize the weights for the full connected layer and the convolution kernel perfectly. However in the numerical experiments and in practice, stochastic gradient descent with momentum is used for training and this will contribute an error term to the right hand side of Theorem 1.

In Theorem 1, can you give more explanation as to the meaning of model complexity $d_{\mathcal{M}}$? Is this the effective number of parameters? Also what are $P$ and $L$ here? In addition, is $d_{\mathcal{M}}$ always larger than $\delta$? As there is no restriction on $\delta$, it is possible that $\delta$ is much larger than $d_{\mathcal{M}}$. The same comment also applies to Theorem 2.

Some other comments:
1.  In (2), state that $1\leq i\leq n$
2.  In the paragraph after Assumption 1, what do you mean by $\lambda_{\mathrm{min}}(f_X(\theta))$ as $f_X(\theta)$ is not a matrix.
3.  On Page 5 the 2nd line after the first display, mode-$3$ multiplication between two tensors does not seem to be introduced in the notations

---

> ### Author Response · Authors · 2020-11-21
> **Response to Reviewer 2 (1/5)**
>
> Thank you for your constructive comments and suggestions!
>
> Firstly, we would like to bring to your attention that we have updated our draft. Based on your comment, we have added one whole new section *"A.4 Classification problems"* to the Appendix on pages 24-31. In the section, we provide some theorem and corollaries for both binary and multiclass classification problems. Since our theoretical analysis is model-based, and changes in model assumption will require different proof techniques, we have also provided a complete proof for the new theorem.
>
> Also, the numbering of some equations has changed, and in our response below, we will use the numbering of the newest version of the draft. For instance, the equation (2) in the old version is the equation (3) in the new version.
>
> __Comment 1__
>
> > One of the main uses of CNNs is to perform classification where there is a softmax layer after the fully connected layer. Hence model (2) does not reflect what is done in practice and y should be a vector of probabilities. In fact I think the authors did classification on CIFAR-10 and SVHN in the numerical experiments (although it is not mentioned) using residual networks and not the model in (2). Also, why do you add sub-Gaussian errors to CNN outputs as in (2)?
>
> We consider a simple regression problem for our theoretical analysis, and hence our formulation at (3) [equation (2),originally]. We assume our CNN Model at (3) has an additive error $\xi$, which follows a sub-Gaussian distribution, akin to [1]. For a regression problem, the additive error assumption is very natural and corresponds to many frequently used loss functions, including the mean square loss considered in our paper. The sub-Gaussian distribution include normal distribution and many other normal-like distributions. This setting is considered as the most fundamental case, but we can indeed extend our theoretical framework to classification problems.
>
> Due to the page limit, we organized the classification problem as a whole new section into our Appendix, since the regression setting (which allows for both discrete and continuous output) is more general than classification setting (which allows for only discrete label output).
>
> Specifically, consider the settings in our Appendix A.4, we present the following theoretical results, together with some proofs.
>
> - For binary classification problems,
>
> __Theorem 3 (Classification: CNN)__
> > Under some technical assumptions, suppose that $n\gtrsim d_\mathcal{M}$, where $d_\mathcal{M} = K(P+L+1)$. Then, for some $\delta>0$,
> $$
>  \|\mathcal{\widehat{W}} - \mathcal{W}^*\|_\mathrm{F} \leq \frac{2\sqrt{\kappa_U}}{\kappa_1}\left(\sqrt{\frac{d_\mathcal{M}}{n}}+\sqrt{\frac{\delta}{n}}\right),
> $$
> with probability $1-\exp(-0.25cn + 9d_\mathcal{M})-2\exp(-c_\gamma d_\mathcal{M}-c\delta)$, where $\delta = O_p(1)$, $c$ and $c_\gamma$ are some positive constants.
>
> Denote $d_\mathcal{M}^\mathrm{TU} = \prod_{j=1}^{N+1}R_j+\sum_{i=1}^{N}l_iR_i+R_{N+1}P$ and $d_\mathcal{M}^\mathrm{CP}=R^{N+1}+R(\sum_{i=1}^{N}l_i+P)$.
>
> __Corollary 2 (Classification: Compressed CNN)__
> > Let $(\mathcal{\widehat{W}},d_{\mathcal{M}})$ be $(\mathcal{\widehat{W}}_{\mathrm{TU}} , d_\mathcal{M}^\mathrm{TU})$ for Tucker decomposition, or $(\mathcal{\widehat{W}}_\mathrm{CP} , d_\mathcal{M}^\mathrm{CP})$ for CP decomposition. Under some technical assumptions, suppose that $n\gtrsim c_Nd_\mathcal{M}$. Then, for some $\delta>0$,
> $$
> \|\mathcal{\widehat{W}} - \mathcal{W}^*\|_\mathrm{F} \leq \frac{2\sqrt{\kappa_U}}{\kappa_1}\left(\sqrt{\frac{3c_Nd_\mathcal{M}}{n}}+\sqrt{\frac{\delta}{n}}\right),
> $$
> with probability $1-4\exp(-0.25cn + 3c_Nd_\mathcal{M})-2\exp(-c_\gamma d_\mathcal{M}-c\delta)$, where $\delta = O_p(1)$, $c$ and $c_\gamma$ are some positive constants, and $c_N$ is defined as in Theorem 2.

---

> > ### Author Response · Authors · 2020-11-21
> > **Response to Reviewer 2 (2/5)**
> >
> > - For multiclass classification problems, suppose we have $M$ classes of labels in total.
> >
> > Because in this case, the predicted output is a vector of length $M$ instead of a scalar, we need to introduce one additional dimension to the fully-connected weight tensor. Subsequently, we use another subscript $m$ to represent the class label. And the fully-connected weights are represented as a set of $\mathcal{B}_\{k,m}$s, for $1\leq k\leq K,1\leq m\leq M$, where each $\mathcal{B}_\{k,m}$ is of size $p_1\times p_2\times\cdots\times p_N$.
> >
> > Then, for each input tensor $\mathcal{X}$, our predicted output is a vector of length $M$, where the $m$th entry is represented by
> >
> > $$
> > \text{output}_m = \langle \mathcal{Z}, \mathcal{W}_m \rangle,
> > $$
> >
> > where $\mathcal{W}_m = \sum_\{k=1}^K(\mathcal{B}_\{k,m}\otimes\mathcal{A}_k)$ is an $N$-th order tensor of size $l_1p_1\times l_2p_2\times\cdots\times l_Np_N$.  We can stack the set of $\mathcal{W}_m$s into $\mathcal{W}_\mathrm{stack}$, which is an $N+1$-order tensor of size $l_1p_1\times l_2p_2\times\cdots\times l_Np_N\times M$. Then, we can show the following theoretical results for $\mathcal{W}_\mathrm{stack}$.
> >
> > __Corollary 3 (Multiclass Classification: CNN)__
> >
> > > Under some technical assumptions, suppose that $n\gtrsim d_\mathcal{M}^\mathrm{MC}$, where $d_\mathcal{M}^\mathrm{MC} = K(MP+L+1)$. Then, for some $\delta>0$,
> > $$
> > \|\mathcal{\widehat{W}}_\mathrm{stack} - \mathcal{W}_\mathrm{stack}^*\|_\mathrm{F} \leq \frac{2\sqrt{\kappa_U}}{\kappa_1}\left(\sqrt{\frac{d_\mathcal{M}^\mathrm{MC}}{n}}+\sqrt{\frac{\delta}{n}}\right),
> > $$
> > with probability $1-4\exp(-0.25cn + 9d_\mathcal{M}^\mathrm{MC})-2\exp(-c_\gamma d_\mathcal{M}^\mathrm{MC}-c\delta)$, where $\delta = O_p(1)$, $c$ and $c_\gamma$ are some positive constants.
> >
> > Denote $d_\mathcal{M}^\mathrm{MC-TU} = \prod_{j=1}^{N+1}R_j+\sum_{i=1}^{N}l_iR_i+R_{N+1}MP$ and $d_\mathcal{M}^\mathrm{MC-CP}=R^{N+1}+R(\sum_{i=1}^{N}l_i+MP)$.
> >
> > __Corollary 4 (Multiclass Classification: Compressed CNN)__
> >
> > > Let $(\mathcal{\widehat{W}}_\mathrm{stack},d_\mathcal{M})$ be $(\mathcal{\widehat{W}}_\mathrm{stack,TU},d_\mathcal{M}^\mathrm{MC-TU})$ for Tucker decomposition, or $(\mathcal{\widehat{W}}_\mathrm{stack,CP},d_\mathcal{M}^\mathrm{MC-CP})$ for CP decomposition.
> > Under some technical assumptions, suppose that $n\gtrsim c_Nd_\mathcal{M}$. Then, for some $\delta>0$,
> > $$
> > \|\mathcal{\widehat{W}}_\mathrm{stack} - \mathcal{W}_\mathrm{stack}^*\|_\mathrm{F} \leq \frac{2\sqrt{\kappa_U}}{\kappa_1}\left(\sqrt{\frac{3c_Nd_\mathcal{M}^\mathrm{MC}}{n}}+\sqrt{\frac{\delta}{n}}\right),
> > $$
> > with probability $1-4\exp(-0.25cn + 3c_Nd_\mathcal{M}^\mathrm{MC})-2\exp(-c_\gamma d_\mathcal{M}^\mathrm{MC}-c\delta)$, where $\delta = O_p(1)$, $c$ and $c_\gamma$ are some positive constants, and $c_N$ is defined as in Theorem 2.
> >
> > So similar observations can be drawn.

---

> > > ### Author Response · Authors · 2020-11-21
> > > **Response to Reviewer 2 (3/5)**
> > >
> > > __Comment 2__
> > >
> > > > In the experiments, the authors used ReLU instead of linear activation, and there is batch normalization between convolution layers. In addition, there are residual connections in the ResNet (block) architecture. All these factors that affect convergence rates were not considered in the theoretical analysis. Although I understand that these issues were omitted to simplify the proofs, I feel that there should be consistency between the theory and implementation parts. Hence Table 2 and Figure 3 do not necessarily provide empirical evidence to the theoretical results derived, as these networks have different architectures than the ones considered in Section 2. Consequently, the results hold true for a special type of CNN and it is not clear to me whether these results will still hold for CNNs used in practice (with non-linear activation, batch norm, max pooling etc.), or when the true weight tensor does not have the same tensor product structure as the learned weight tensor.
> > >
> > > For theoretical analysis, there is a gap between generalization and specialization.  We aim at providing a result that is more general, so we adopt vanilla CNNs  and compressed CNNs to set up our basic framework. And at the end of Section 3, we point out that the ablation studies are conducted to see whether our finding (under linear activations) can also apply to a more realistic setting (with nonlinearity and other possible features).
> > >
> > > Our ablation studies try to adjust the $K/R$ ratio on the residual bottleneck block in ResNet [2], and empirical results showed that when $K/R=1$, the test accuracy is comparable to $K/R=4,8,16$ and the number of parameters is reduced by a lot. This is in accordance with our theoretical finding in Corollary 1. And it hints that our finding can indeed provide guidance to network designs in real life. To provide stronger empirical support for this point, we have also conducted more ablation studies on recent state-of-the-art models, Shufflenet [3] and ResNeXt [4]. We conducted the experiments on the SVNH dataset and take $t = 1,2,4$, since $t=2$ or $4$ is commonly used in practice. The experiment results are presented in the following table, where we set seeds 1-5 and reported the worst case scenario of the test accuracy.
> > >
> > > (Updated !)
> > > --------------------------------------------------------------------------
> > >   | t=K/R | ResNeXt | #FLOPs   | Params | Shufflenet | #FLOPs   | Params |
> > >   | ----- | ------- | -------- | ------ | ---------- | -------- | ------ |
> > >   | 1     | 96.20   | 0.05GMac | 1.45M  | 95.93      | 0.05GMac | 1.33M  |
> > >   | 2     | 96.25   | 0.07GMac | 2.13M  | 96.08      | 0.08GMac | 1.67M  |
> > >   | 4     | 96.20   | 0.14GMac | 4.44M  | 96.03      | 0.16GMac | 3.84M  |
> > > --------------------------------
> > > We can see that, when $t=1$, the test accuracy is comparable to when $t=2$ or 4, and the number of parameters are reduced by a lot. Hence, it is consistent with our finding in Corollary 1.
> > >
> > > However, as we are starting from a vanilla model, it is always possible to add other features to it. This leads to more interesting future works. For instance, if we add the shortcut connection from a vector input $\textbf{x}$ to the intermediate output after convolution $\textbf{x}_c$, and denote the aggregated intermediate output to be $\textbf{x}_c^\prime$. To ensure $\textbf{x}_c$ and $\textbf{x}_c^\prime$ have equal dimensions, we perform a linear projection $\textbf{W}_s$ by the shortcut connection to match the dimensions:
> > > $$
> > > \textbf{x}_c^\prime = \textbf{x}_c + \textbf{W}_s\textbf{x}
> > > $$
> > >  similar to the equation (2) in [6].
> > >
> > >  We conjecture that, the predicted outcome of a single kernel case, will be of the form
> > > $$
> > > \widehat{y} = \left\langle\textbf{x},\widetilde{\textbf{W}}_s^\prime\textbf{b} + \textbf{U}^\{(1)}_\mathcal{F}(\textbf{b}\otimes\textbf{a})\right\rangle,
> > > $$
> > > where $\widetilde{\textbf{W}}_s\in\mathbb{R}^{p_1\times d_1}$ , and each of its rows is a linear combination of the rows of $\textbf{W}_s$, the shortcut transformation matrix. A general 3-layer $N$ dimensional CNN will have a similar expression. The complexity will increase greatly for a multilayer CNN.
> > >
> > > Nonlinearity and max-pooling are possible features to add as well. However, the non-differentiability and possible non-convexity will bring some technical challenges and new proof techniques will need to be developed.

---

> > > > ### Author Response · Authors · 2020-11-21
> > > > **Response to Reviewer 2 (4/5)**
> > > >
> > > > __Comment 3__
> > > >
> > > > > Equation (5) seems to assume that you can optimize the weights for the full connected layer and the convolution kernel perfectly. However in the numerical experiments and in practice, stochastic gradient descent with momentum is used for training and this will contribute an error term to the right hand side of Theorem 1.
> > > >
> > > > We agree with you that the weights in the model cannot be optimized perfectly, and a specific optimization algorithm will further introduce an optimization error. Actually the objective function at (6) on page 5 is nonconvex with respect to $\mathcal{B}_k$s and $\mathcal{A}_k$s, and the trained weights are not even a global minimizer.
> > > >
> > > > However, as explained in our newly added Subsection 1.1, this paper attempts to conduct a statistical sample complexity analysis, to theoretically explain how much compressibility is achieved in a compressed network architecture. we are more interested in the trained model itself, and the possible optimization errors may be ignored when comparing two CNNs. To better clarify our problem, we have introduced a prediction error $\mathcal{E}(\mathcal{\widehat{W}})=\sqrt{E_{\textrm{x}}|F_{\textrm{CNN}}(\textrm{x}, \mathcal{\widehat{W}})- F_{\textrm{CNN}}(\textrm{x}, \mathcal{W}^*)|^2}$, and in Theorem 1, we also provide the bound for $\mathcal{E}(\mathcal{\widehat{W}})$.
> > > >
> > > > Moreover, our analysis framework is closely related to [1]. And they actually do not consider the computational complexity or algorithm convergence.
> > > >
> > > > __Comment 4__
> > > >
> > > > > In Theorem 1, can you give more explanation as to the meaning of model complexity $d_\mathcal{M}$? Is this the effective number of parameters? Also what are P and L here? In addition, is $d_\mathcal{M}$ always larger than $\delta$? As there is no restriction on $\delta$, it is possible that $\delta$ is much larger than $d_\mathcal{M}$. The same comment also applies to Theorem 2.
> > > >
> > > > Thank you so much for your careful reading!  $L=l_1l_2\cdots l_N$ represents the size of each convolution kernel $\mathcal{A}_k$, and $P = p_1p_2\cdots p_N$ represents the size of each corresponding fully-connected weights $\mathcal{B}_k$. We have now added the notations before Theorem 1 for clarification.
> > > >
> > > > Yes, you are right. Intuitively, you can understand $d_\mathcal{M}$ as the effective number of parameters. More rigorously, the sample complexity analysis is to investigate how many samples are needed to guarantee a given tolerance on the prediction error $\mathcal{E}(\mathcal{\widehat{W}})$. For instance, in Theorem 1, to achieve a fixed prediction error of $\varepsilon$, it requires that the number of samples is order  $O(d_\mathcal{M}/\varepsilon^2)$. So, you could also intuitively view it as the degree of freedom in the system.
> > > >
> > > > In fact, $\delta=O_p(1)$. Since $d_\mathcal{M}$ will grow with $K$, $P$ and $L$ , $d_\mathcal{M}$ will always be the dominating term. And $\delta$ will not affect the order of our bound. The reason for including $\delta$ is that, for our theoretical analysis, we do not require $n\to \infty$. In other words, the upper bounds in Theorems 1\&2 hold, even when $n$ is finite. So, we include $\delta$ only to make our probability statement more rigorous. We have added "$\delta = O_p(1)$" into our Theorems to clarify this point.

---

> > > > > ### Author Response · Authors · 2020-11-21
> > > > > **Response to Reviewer 2 (5/5)**
> > > > >
> > > > > __Comment 5__
> > > > >
> > > > > > Some other comments:
> > > > > >
> > > > > > 1. In (2), state that $1\leq i\leq n$
> > > > > > 2. In the paragraph after Assumption 1, what do you mean by $\lambda_{min}(f_X(\theta))$ as $f_X(\theta)$ is not a matrix.
> > > > > > 3. On Page 5 the 2nd line after the first display, mode-3 multiplication between two tensors does not seem to be introduced in the notations
> > > > >
> > > > > For 1, we have added this statement back. Thanks you for the reminder!
> > > > >
> > > > > For 2, in fact, $f_X(\theta)$ represents the spectral density matrix of a multivariate time series. Specifically, consider a $p$-dimensional stationary time series process $\textbf{x}^t$, $t\in\mathbb{Z}$ with autocovariance function given by $\Gamma_X(h) = \text{Cov}(\textbf{x}^t,\textbf{x}^\{t+h})$, $t,h\in\mathbb{Z}$. The spectral density density (matrix) function is given by $f_X(\theta) = {2\pi}^{-1}\sum_\{l=-\infty}^\infty\Gamma_X(l)e^\{-il\theta}$, where $\theta\in[-\pi,\pi]$. Due to the page limit, we may not be able to include the above definition into our paper, so we refer the readers to [5] for more detailed explanations.
> > > > >
> > > > > For 3, we would like to thank you again for your thorough reading! We have added the explanation for the notation in Section 2.4, below equation (7).
> > > > >
> > > > >
> > > > >
> > > > > [1] Du, Simon S., et al. "How many samples are needed to estimate a convolutional neural network?." *Advances in Neural Information Processing Systems*. 2018.
> > > > >
> > > > > [2] He, Kaiming, et al. "Deep residual learning for image recognition." *Proceedings of the IEEE conference on computer vision and pattern recognition*. 2016.
> > > > >
> > > > > [3] Ma, Ningning, et al. "Shufflenet v2: Practical guidelines for efficient cnn architecture design." Proceedings of the European conference on computer vision (ECCV). 2018.
> > > > >
> > > > > [4] Xie, Saining, et al. "Aggregated residual transformations for deep neural networks." *Proceedings of the IEEE conference on computer vision and pattern recognition*. 2017.
> > > > >
> > > > > [5] Basu, Sumanta, and George Michailidis. "Regularized estimation in sparse high-dimensional time series models." *The Annals of Statistics* 43.4 (2015): 1535-1567.

---

### Official Review · AnonReviewer3 · 2020-10-29
**A theoretical analysis for higher-order CNNs using tensor methods**

**Rating:** 6
**Confidence:** 3

**Review:**

Summary:
This paper formulated higher-order CNNs into a Tucker form and provides sample complexity analysis to higher-order CNNs and compressed designs of CNNs via tensor analysis. It uses then theoretically analyzes the efficiency of four block designs from ResNet, MobileNetV1, and MobileNetV2. The paper also conducts numerical experiments to verify its theoretical results and provide some empirical studies to show that increasing the expansive ratio of a bottleneck

Pros:
1.This work provides a theoretical analysis for higher-order CNNs via analyzing its Tucker formulation using tensor methods.
2.The proposed theoretical analysis can be applied to compressed designs of CNNs.
3.This work also provides numerical experiments to verify its theoretical claims.

Cons:
1.This work lacks comparisons with many important and relevant works (e.g. [1-5]), which also formulate CNNs or higher-order CNNs using various tensor decomposition forms. Many of the works also provide theoretical analysis (e.g. generalization bound in [5]) for the proposed formulations. It would be nice for the authors to show how their formulation is different from the existing ones and what is novel about the proposed formulation. Since [3] and [4] both have formulations of higher-order CNNs/CNNs using Tucker decomposition, it seems to me that the current formulation proposed in this work lacks novelty.

2.The current presentation of this work could be much more improved via a) providing more intuitions for its theoretical analysis, b) putting more connections between the theoretical analysis and empirical experiments, and c) adopting better notations (e.g. definitions of tensor operations rather than using elementwise notations) to make the theoretical analysis cleaner and clearer.

3.The finding discovered in this work via its theoretical analysis lacks sufficient experimental supports: the finding is only shown using one particular architecture design and the room for potential improvements is very limited. For example, in Table 2, the test accuracy of even the smallest model is already very close to the state-of-the-art results on these datasets, which leaves very limited room for potential improvements of test accuracies by simply increasing the expansion ratio.

4.As mentioned above, because a) the proposed formulation of higher-order CNNs lack proper comparisons with existing works and has limited novelty and b) the finding from theoretical analysis lack sufficient experimental supports, the contribution of this paper is limited and it would be nice for the authors to provide more justifications for its novelty and better designs of experiments to convey the message.

[1] Kossaifi, Jean, et al. "Tensor contraction layers for parsimonious deep nets." Proceedings of the IEEE Conference on Computer Vision and Pattern Recognition Workshops. 2017.

[2] Kossaifi, Jean, et al. "T-net: Parametrizing fully convolutional nets with a single high-order tensor." Proceedings of the IEEE Conference on Computer Vision and Pattern Recognition. 2019.

[3] Su, Jiahao, et al. "Tensorial neural networks: Generalization of neural networks and application to model compression." arXiv preprint arXiv:1805.10352 (2018).

[4] Kossaifi, Jean, et al. "Tensor regression networks." Journal of Machine Learning Research 21.123 (2020): 1-21.

[5] Li, Jingling, et al. "Understanding Generalization in Deep Learning via Tensor Methods." arXiv preprint arXiv:2001.05070 (2020).

---

> ### Author Response · Authors · 2020-11-20
> **Response to Reviewer 3 (1/2)**
>
> Thank you for your detailed comments and suggestions!
>
> __Updates based on your comments__
> Based on your suggestions, we have reorganized the whole Section 1 "Introduction" by including more discussions about (i) relevant works in tensor methods for CNN compression and, (ii) how the literature on generalization error differs from our sample complexity analysis in both goal and methodology. We also added extra discussions in Section 5 "Conclusion and Discussion". We plan to include more experiment studies on the $K/R$ ratio in more recent state-of-the-art networks in Section 4 "Experiments".
>
> __Con 1__
> >  This work lacks comparisons with many important and relevant works (e.g. [1-5]), which also formulate CNNs or higher-order CNNs using various tensor decomposition forms. Many of the works also provide theoretical analysis (e.g. generalization bound in [5]) for the proposed formulations. It would be nice for the authors to show how their formulation is different from the existing ones and what is novel about the proposed formulation. Since [3] and [4] both have formulations of higher-order CNNs/CNNs using Tucker decomposition, it seems to me that the current formulation proposed in this work lacks novelty.
>
> Briefly speaking, our formulation summarizes all weights of a high-order CNN into a single tensor, akin to [2], but our summarized tensor has an explicit "nested doll" structure to account for the interactions between weights across layers; see for instance, equations (4)&(7). While the literature on generalization error aims to understand why deep neural networks (including CNNs) can generalize well via various regularization methods, we aim to theoretically explain how much compressibility is achieved in a compressed CNN architecture and whether there remains model redundancy to be reduced. For this purpose, we formulate the networks exactly and conduct statistical sample complexity analysis.
>
> __Relation to [1]&[4]__: It is common in literature to apply tensor decomposition to CNNs layer-by-layer. On the one hand, when tensor decomposition is applied to the convolution layers, it corresponds to various block designs, which we focus on in our theoretical study in Theorem 2. On the other hand, when tensor decomposition is applied to the fully-connected layer instead, it corresponds to the tensor contraction in [1] or tensor regression layer in [4]. In fact, our CNN formulation allows us to easily extend the theoretical analysis to the fully-connected weights $\mathcal{B}$ with low rank structure. And we can hence establish the sample complexity analysis for the CNNs with tensor regression layer, which we leave as future work; see Section 5.
>
> __Relation to [2]__: Our formulation is similar to [2] in that we also parametrize the network weights into a single tensor. While [2] assumes the tensor to have a heuristic Tucker form, we replicate the layer-by-layer operations of CNNs and show that the summarized tensor has a "nested doll" structure. In other words, the low-rank structure of the previous layer is nested within that of the current layer.
>
> __Relation to [3]__: Both our paper and [3] target towards tensor structured inputs. However, each layer of Tensorial Neural Network in [3] still conducts a 2D convolution along two selected dimensions of the input (see Fig 4(d) in [3]). Meanwhile, we allow for a multidimensional (high-order) convolution along all input dimensions, which helps to explore the data structure more efficiently.
>
> __Relation to [5]__: We include a subsection 1.1 to make the comparison between our theoretical approach and the generalization bound. In short, most studies on generalization bound, including [5], aim to understand the generalization ability of DNNs.  Hence, they do not require an explicit network architecture but requires some form of regularization. Li [5] proposed to use tensor decomposition as their regularization technique. But their study is essentially still model-agnostic, since the ranks of the CP layers depend solely on trained weights. In conclusion, their approach is not suitable for explaining fixed network architectures with pre-designed low-rank layers (regardless of training).
>
> An additional comment is that the parameter efficiency in [1]-[4] was heuristically justified by methods, such as FLOPs counting, naive parameter counting and/or empirical running time. However, there is still lack of a theoretical study to understand the mechanism of how tensor decomposition can compress CNNs. Our paper attempts to fill this gap from statistical perspectives.

---

> > ### Author Response · Authors · 2020-11-20
> > **Response to Reviewer 3 (2/2)**
> >
> > __Con 2__
> > > The current presentation of this work could be much more improved via a) providing more intuitions for its theoretical analysis, b) putting more connections between the theoretical analysis and empirical experiments, and c) adopting better notations (e.g. definitions of tensor operations rather than using elementwise notations) to make the theoretical analysis cleaner and clearer.
> >
> > Thank you for this constructive comments! Accordingly, we updated our draft by (a) providing more explanation about statistical sample complexity in Section 1 "Introduction", and (b) adding more details in Section 3 to show how results from Theorem 2 and Corollary 1 can inspire us to conduct empirical experiments.
> >
> > Specifically, in Section 1, we define the root-mean-square prediction error $\mathcal{E}(\mathcal{\widehat{W}})$ and the sample complexity analysis is to investigate how many samples are needed to guarantee a given tolerance on the prediction error. It can also be used to detect possible model redundancy in a compressed network. Theorem 1\& 2 analyze the sample complexity for a basic CNN and a compressed CNN. Comparing the two, we can see that, when the input dimension $N$ is large, tensor decomposition to the convolution layer can indeed reduce a large number of parameters in the network.
> >
> > We further observe that, if we assume the output channels dimension of size $K$ to have a rank of $R$, the sample complexity in Theorem 2 then contains the parameter $R$ instead of $K$. This differs from the naive parameter counting. We try to find the rationale behind this counter-intuitive observation, and hence come up with Corollary 1 as the explanation. Essentially, Corollary 1 states that, if we impose low rank along the output channels dimension, the compressed CNNs with linear activation will have unnecessary model redundancy. We are curious to whether such findings can be extended to realistic settings. So, we carry out the ablation studies on widely used residual bottleneck structure introduced in ResNet[6].
> >
> > Some elementwise notations are used to help understand the derivation of our model. The notations may perhaps be easier to follow if we explain the formulation using a special case of $N=3$ before extending to a general $N$ dimensional case. Thank you for your suggestion, we will think about it more carefully.
> >
> > __Con 3__
> > > The finding discovered in this work via its theoretical analysis lacks sufficient experimental supports: the finding is only shown using one particular architecture design and the room for potential improvements is very limited. For example, in Table 2, the test accuracy of even the smallest model is already very close to the state-of-the-art results on these datasets, which leaves very limited room for potential improvements of test accuracies by simply increasing the expansion ratio.
> >
> > Thank you for your feedback. Firstly, our ablation studies aims to show that, for the bottleneck block structure, we can choose $K/R=1$ over $K/R>1$, which is commonly used for the original ResNet bottleneck. This is because, for $K/R = 1,4,8,16$, the accuracies are comparable while the number of parameter doubles or even quadruples when $K/R$ is large.
> >
> > As the bottleneck block structure is adopted in many other state-of-the-art networks, such as Shufflenet[7] or ResNeXt[8], it is highly likely  that the findings from our ablation studies can be applied to them as well. So the implications can be far-reaching.
> >  Following your suggestions, more ablation studies are conducted on Shufflenet and ResNeXt with $t = K/R = 1,2,4$. We set seeds 1-5 and reported the worst case scenario of the test accuracy on the SVNH dataset.
> >
> > (Updated!)
> > --------------------------------------------------------------------------
> >    | t=K/R | ResNeXt | #FLOPs   | Params | Shufflenet | #FLOPs   | Params |
> >    | ----- | ------- | -------- | ------ | ---------- | -------- | ------ |
> >    | 1     | 96.20   | 0.05GMac | 1.45M  | 95.93      | 0.05GMac | 1.33M  |
> >    | 2     | 96.25   | 0.07GMac | 2.13M  | 96.08      | 0.08GMac | 1.67M  |
> >    | 4     | 96.20   | 0.14GMac | 4.44M  | 96.03      | 0.16GMac | 3.84M  |
> > --------------------------------
> >
> > We can see that, when $t=1$, the test accuracy is comparable to when $t=2$ or 4, and the number of parameters are reduced by a lot.  Hence, it is always recommended to keep $t=1$ to achieve more parameter efficiency without sacrificing the test accuracy.
> >
> > [1] - [5] Same as the provided references.
> >
> > [6] He, Kaiming, et al. "Deep residual learning for image recognition." *Proceedings of the IEEE conference on computer vision and pattern recognition*. 2016.
> >
> > [7] Ma, Ningning, et al. "Shufflenet v2: Practical guidelines for efficient cnn architecture design." Proceedings of the European conference on computer vision (ECCV). 2018.
> >
> > [8] Xie, Saining, et al. "Aggregated residual transformations for deep neural networks." *Proceedings of the IEEE conference on computer vision and pattern recognition*. 2017.

---

### Author Response · Authors · 2020-11-14
**Thanks to all the reviewers for their valuable comments, we will respond to all the suggestions and improve upon our work**

We thank all the reviewers for their constructive comments and suggestions! We would like to first address the general interest in the novelty of our work, and then address detailed questions from each reviewers individually in the next few days.

Notably, our main goal for empirical experiments is not to introduce brand new network structures, but to slightly modify some existing block structures, such as the residual bottleneck block or the depthwise separable block. The modification on the $K/R$-ratio is guided by our theoretical analysis. Since these block structures are widely adopted in more recent state-of-the-art models, such as ResNeXt [1] and Shufflenet [2], our finding on the $K/R$-ratio can appeal to a larger audience.

Our CNN formulation shows that, under linear activations, it is possible to formulate all parameters in a multilayer CNN into a single high-order weight tensor $\mathcal{W}_X$, which has a special "nested doll" structure to account for interactions across layers; see for instance, equations (4)&(7). And, one can freely impose low-rank structure on the convolution kernels $\mathcal{A}$ in each layer or the fully-connected weights $\mathcal{B}$.

Based on our formulation, we conduct theoretical studies to understand the mechanism of how tensor decomposition can compress CNNs via statistical sample complexity analysis. We define the root-mean-square prediction error, and the sample complexity analysis aims to investigate how many samples are needed to guarantee a given tolerance on the prediction error. It can also be used to detect the model redundancy.

There are two main insights from our theoretical studies:

- Comparing the sample complexity in Theorem 1 and 2, we see that the sample complexity of the compressed CNN depends on $\sum_{i=1}^{N}l_i$ instead of $\prod_{i=1}^{N}l_i$. When the input dimension $N$ is large, compressed CNN can indeed reduce a large number of parameters.
- Corollary 1 states that, if we impose low rank $R$ along the output channels dimension of size $K$ $(K>R)$, the compressed CNNs with linear activation will have unnecessary model redundancy.

Though our theoretical framework works under linear activations, we believe the implications can be extended to real applications. We hence perform minor modifications to ResNet bottleneck structure by requiring the expansion ratio $t = K/R = 1$, and show that the revised network maintain comparable performance against the original ResNet with $t = 4,8,16$, while using much less parameters. The empirical results show that our theoretical finding under simpler assumptions, can be effective to guide the design modification of highly complicated network designs with added residual shortcuts, etc.

To provide further empirical support for our findings, we are currently conducting ablation studies on more recent state-of-the-art network designs such as ResNeXt and ShuffleNet, and will present the results near the end of the review period.

Following the suggestions from the reviewers, in these few days, we will update our paper constantly.

1) We will add more detailed discussions about the connections and comparisons between our work and other tensor compressed CNNs in the "Introduction" section.

2) We will add an additional subsection below the "Introduction" section to address the difference between our theoretical analysis and the generalization error bound. In the subsection, we will also show that we focus on statistical sample complexity and is hence different from the studies on algorithm convergence.

3) We will add more intuition about Corollary 1, and how it inspires our ablations studies.

4) We are conducting more ablation studies and will present the results.

We would also like to welcome any further discussions or suggestions!



-------------------------

*References*

[1] Xie, Saining, et al. "Aggregated residual transformations for deep neural networks." *Proceedings of the IEEE conference on computer vision and pattern recognition*. 2017.

[2] Ma, Ningning, et al. "Shufflenet v2: Practical guidelines for efficient cnn architecture design." Proceedings of the European conference on computer vision (ECCV). 2018.

[3] Kossaifi, Jean, et al. "T-net: Parametrizing fully convolutional nets with a single high-order tensor." *Proceedings of the IEEE Conference on Computer Vision and Pattern Recognition*. 2019.

---

### Author Response · Authors · 2020-11-23
**Summary of updates**

Based on the constructive suggestions and feedbacks from the reviewers, we have updated our draft!

__Summary of main updates__

 __1.__ The whole Section 1 "Introduction" was reorganized, following the helpful advices from reviewer 1&3.



   *Summary*: Most work on tensor methods for CNN compression use heuristic approaches to evaluate parameter efficiency in networks, and we aimed to provide a more theoretical based approach. We adopted the statistical sample complexity analysis, which is related to our defined root-mean square prediction error $\mathcal{E}(\mathcal{W})$. It can also be used to detect model redundancy. Our theoretical analysis is different from the study of generalization bound in both purpose and methodology. And our derived bounds are independent of any training algorithm.



 __2.__ We added one whole new section *"A.4 Classification problems"* to the Appendix on pages 24-31, following the insightful comments from reviewer 2.



   *Summary*: We included one Theorem and two corollaries to show that our analysis can be extended to binary and multiclass classification problems.



 __3.__ We added additional ablation studies of the $K/R$ ratio on more state-of-the-art networks, namely ResNeXt [1] and Shufflenet v2 [2]. The details for network designs and implementations are presented in Appendix 5.2 on pages 32-33.



   *Summary\&results of experiment*

   We conducted the experiment on the SVNH dataset. The experiment results are presented in the following table, where we set seeds 1-5 and reported the worst case scenario of the test accuracy.

--------------------------------
   | t=K/R | ResNeXt | #FLOPs   | Params | Shufflenet | #FLOPs   | Params |
   | ----- | ------- | -------- | ------ | ---------- | -------- | ------ |
   | 1     | 96.20   | 0.05GMac | 1.45M  | 95.93      | 0.05GMac | 1.33M  |
   | 2     | 96.25   | 0.07GMac | 2.13M  | 96.08      | 0.08GMac | 1.67M  |
   | 4     | 96.20   | 0.14GMac | 4.44M  | 96.03      | 0.16GMac | 3.84M  |
------------------------------

We can see that, when $t=1$, the test accuracy is comparable to when $t=2$ or 4, and the number of parameters are reduced by a lot. In fact, the ratio of number of parameters at a single bottleneck block for $t=1, 2$ and 4 is roughly $1:2:4$ for both networks. This implies that the number of parameters will increase dramatically when we have a deeper CNN, for instance, ResNeXt-101. Hence, it is always recommended to keep $t=1$ to achieve more parameter efficiency without sacrificing the test accuracy.

---

### Decision · Program_Chairs · 2021-01-07
**Final Decision**

**Decision:**

Reject

**Comment:**

This paper presents a theoretical analysis of CNN compression using tensor methods. None of the three reviewers have strong opinion; there scores are 5, 6, and 5.

The attempt to understand the mechanism of how tensor decomposition compresses CNNs is meaningful and interesting. However, the main contribution of this work is not sufficiently distinct compared to the existing approaches and the analysis and proof is conduected only for simplified models as mentioned by reviewers. The practical benefit of this paper is not clear and the experimental validation is weak because only a small number of model architectures were tested on a few small datasets.

This is a borderline paper. However, this paper needs to extend its contribution by performing more comprehensive analysis for general CNNs.